# Functional Adjoint Sampler: Scalable Sampling on Infinite Dimensional Spaces

**Byoungwoo Park** [1]   **Juho Lee** [1]   **Guan-Horng Liu** [2]

## Abstract

Learning-based methods for sampling from the Gibbs distribution in finite-dimensional spaces have progressed quickly, yet theory and algorithmic design for infinite-dimensional function spaces remain limited. This gap persists despite their strong potential for sampling the paths of conditional diffusion processes, enabling efficient simulation of trajectories of diffusion processes that respect rare events or boundary constraints. In this work, we present the adjoint sampler for infinite-dimensional function spaces, a stochastic optimal control-based diffusion sampler that operates in function space and targets Gibbs-type distributions on infinite-dimensional Hilbert spaces. Our Functional Adjoint Sampler (FAS) generalizes Adjoint Sampling (Havens et al., 2025) to Hilbert spaces based on a SOC theory called stochastic maximum principle, yielding a simple and scalable matching-type objective for a functional representation. We show that FAS achieves superior performance on transition path sampling across synthetic potential and real molecular systems, including Alanine Dipeptide and Chignolin. Codes are available at https://github.com/bw-park/FAS.

## 1. Introduction

The study of constructing Markov diffusion processes under certain conditions—such as rare events and prescribed terminal distributions—has long been a central theme in probability and statistical physics (Doob, 2001; Chetrite and Touchette, 2015). With the recent explosive interests in generative modeling (Song et al., 2021; Lipman et al., 2023), the framework has emerged in many other machine learning applications, including data assimilation (Chopin et al., 2023; Park et al., 2025), population modeling (Liu

et al., 2022; Chen et al., 2023), molecule dynamics simulation (Plainer et al., 2023; Holdijk et al., 2022), and—of particular focus of this work—sampling from the Gibbs distribution (Tzen and Raginsky, 2019).

Specifically, we are interested in generating samples from a Gibbs distribution $\pi(\mathbf{x}) \propto e^{-U(\mathbf{x})}$ given its unnormalized energy function $U(\mathbf{x})$. Modern computational methods—known as *diffusion samplers*—achieve this by learning diffusion processes whose terminal distribution matches the law of $\pi$. Among these, adjoint-based diffusion samplers (Havens et al., 2025; Liu et al., 2026; Choi et al., 2026; Guo et al., 2026) form a distinctive family by reformulating the sampling problem as a stochastic optimal control problem (SOC) (Fleming and Soner, 1993; Kappen, 2005) and, from which, employing Adjoint Matching (AM) (Domingo-Enrich et al., 2025). For instance, Adjoint Sampling (AS) (Havens et al., 2025) learns the drift of a *memoryless*[1] stochastic differential equation with an adjoint-matching objective that depends solely on on-policy samples and a simulation-free target. This yields a highly scalable method compared to vanilla SOC-based methods that rely on simulation-based targets (Zhang and Chen, 2022), as well as those that require importance-weighted samples (Phillips et al., 2024; Akhound-Sadegh et al., 2024).

Despite their practical successes, all of the aforementioned adjoint samplers generate samples only in finite-dimensional state spaces, *e.g.*, $\mathbf{x} \in \mathbb{R}^d$. This essentially precludes their applications to sampling Gibbs distributions defined on more general *trajectory* spaces—a family of distributions that appears prevalently in computational chemistry, with a representative example known as the *transition path sampling* (TPS) (Bolhuis et al., 2002; Vanden-Eijnden et al., 2010). At its core, TPS aims to identify rare reactive paths by sampling ensembles of dynamical trajectories between metastable states driven by stochastic fluctuations or external forcing (Dürr and Bach, 1978). Studying these transitions enables accurate estimations of reaction rates and mechanisms that explain how molecules transition between metastable states, benefits various applications such as protein folding (Dobson, 2003; Noé et al., 2009), enzyme catalysis (Basner and Schwartz, 2005), drug design (Tiwary et al.,

---

[1]KAIST [2]FAIR at Meta. Correspondence to: Byoungwoo Park <bw.park@kaist.ac.kr>, Guan-Horng Liu <ghliu@meta.com>.

*Proceedings of the 43$^{rd}$ International Conference on Machine Learning*, Seoul, South Korea. PMLR 306, 2026. Copyright 2026 by the author(s).

[1]Practically, AS (Havens et al., 2025) fulfills the memoryless condition with a fixed initial condition.

2015) and nucleation (Sosso et al., 2016).

As TPS aims to generate a Gibbs distribution of trajectories following specific energy structures, it is nature to consider samplers formulated directly in these *infinite-dimensional function spaces*. Indeed, trajectories of $\mathbb{R}^d$-valued diffusion processes are commonly viewed as random functions (Hairer et al., 2009) where the associated probability laws typically reside on function spaces such as $\mathcal{H} := L^2(\mathbf{L}, \mathbb{R}^d)$, where $\mathbf{L} \subseteq \mathbb{R}$ denotes a time interval. In this case, classical sampling methods based on Langevin-type infinite-dimensional stochastic differential equations have been investigated (Stuart et al., 2004; Beskos et al., 2008), despite remaining primarily of theoretical interest rather than practical utility. On the other hand, existing diffusion samplers for TPS (Holdijk et al., 2022; Seong et al., 2025) remains confined to finite dimensional state spaces because they rely on reference dynamics driven by local energy gradients, which preclude a scalable AM objective.

In this paper, we introduce **Functional Adjoint Sampler** (FAS), a scalable matching algorithm that extends AS to infinite-dimensional function spaces. Formally, we recast sampling from Gibbs-type distributions in their infinite-dimensional formulation and build diffusion samplers in function spaces. Leveraging infinite-dimensional SOC, we cast the non-equilibrium sampling (NES) as a function-space generalization of (Tzen and Raginsky, 2019; Zhang and Chen, 2022). Furthermore, under standard regularity, we demonstrate that Adjoint Matching (AM) can also be extended to function spaces by applying the *Stochastic Maximum Principle* (Bensoussan, 1983; Du and Meng, 2013), providing a simple and scalable training objective.

In practice, we take a particular interest in the problem of TPS, our method generates conditional paths across diverse settings, from synthetic potential to molecular systems such as Alanine Dipeptide and Chignolin. Specifically, FAS operates directly in infinite dimensional function spaces by choosing a basis that enforces the prescribed boundary condition (*i.e.*, Dirichlet boundary) of the simulated path, thereby yielding perfect transition between metastable states without extra projection or penalty terms. This principled formulation ensures well posedness of the conditional diffusion and supports stable discretizations, while outperforming prior works that rely on finite-dimensional dynamics.

We summarize our contributions as follows:

- We introduces a function-space diffusion sampler that draws samples from Gibbs-type distribution on infinite-dimensional path spaces. Taking an infinite-dimensional Ornstein–Uhlenbeck semigroup as the reference dynamics, We casts NES as a infinite-dimensional SOC problem.

- Based on the SOC theory called *Stochastic Maximum*

*Principle*, we propose FAS, a generalization of the matching-based training objective of AS (Havens et al., 2025) to Hilbert spaces and yielding a scalable training algorithm suitable for functional representations.

- We demonstrate superior performance of FAS on TPS, outperforming prior finite-dimensional methods across synthetic potential and challenging real molecular systems including Alanine Dipeptide and Chignolin, while maintaining perfect transition-hit rates to a target states.

**Notation.** Throughout the paper, we consider a real and separable Hilbert space $\mathcal{H}$, equipped with the norm $\|\cdot\|_{\mathcal{H}}$ and inner product $\langle \cdot, \cdot \rangle_{\mathcal{H}}$. We denote $\mathbb{P}^{(\cdot)}$ the path measure on the space of all continuous mappings $\Omega = C([0,T], \mathcal{H})$ and $\mathbf{X}^{(\cdot)} \in \mathcal{H}$ the stochastic processes associated with this path measure. We denote the evaluated value of a (random) function $\mathbf{X}[\cdot] : \mathbb{R}^n \to \mathbb{R}^d$ at a point $u \in \mathbb{R}^n$ from domain space $\mathbb{R}^n$ as $\mathbf{X}[u] \in \mathbb{R}^d$, which will be omitted unless necessary for clarity. For a function $\mathcal{V} : [0,T] \times \mathcal{H} \to \mathbb{R}$, we define $D_{\mathbf{x}}\mathcal{V}, D_{\mathbf{xx}}\mathcal{V}$ as the first and second order Fréchet derivatives with respect to the variable $\mathbf{x} \in \mathcal{H}$, respectively, and $\partial_t \mathcal{V}$ as the derivative with respect to the time variable $t \in [0,T]$. The Hermitian adjoint will be denoted as $\dagger$.

## 2. Preliminaries

Here, we provide a concise overview of the necessary probability concepts for studying diffusion samplers formulated in infinite-dimensional state spaces. Additional review can be found in Appendix A.

**Gaussian measure in Hilbert spaces** Let $(\Omega, \mathcal{F}, \mathbb{Q})$ be a probability space and $(\mathcal{H}, \mathcal{B}(\mathcal{H}))$ be a measurable state space, and let $\mathbf{X} : \Omega \to \mathcal{H}$ be an $\mathcal{H}$-valued random variable inducing the push-forward measure $\nu := \mathbf{X}_{\#}\mathbb{Q}$. Then, the measure $\nu$ is called Gaussian when every $\mathbb{R}$-valued random variable $\langle u, \mathbf{X} \rangle_{\mathcal{H}}$ is Gaussian for all $u \in \mathcal{H}$. In this case, there is a unique mean function $\mathbf{m} \in \mathcal{H}$ satisfying $\langle \mathbf{m}, u \rangle_{\mathcal{H}} = \mathbb{E}_{\nu}[\langle \mathbf{X}, u \rangle_{\mathcal{H}}]$ and unique non-negative self-adjoint covariance operator $Q : \mathcal{H} \to \mathcal{H}$ defined by $\langle u, Qv \rangle_{\mathcal{H}} = \mathbb{E}_{\nu}[\langle \mathbf{X} - \mathbf{m}, u \rangle_{\mathcal{H}} \langle \mathbf{X} - \mathbf{m}, v \rangle_{\mathcal{H}}]$ for all $u, v \in \mathcal{H}$, where it is trace class operator *i.e.*, $\mathrm{Tr}(Q) < \infty$. We write $\nu = \mathcal{N}(\mathbf{m}, Q)$ and say that $\mathbf{X}$ is centered when $\mathbf{m} = 0$. Assume now that $\mathbf{X}$ is centred with law $\mathcal{N}(0, Q)$. Then, there exists an eigen-system $\{(\lambda^{(k)}, \phi^{(k)}) \in \mathbb{R} \times \mathcal{H} : k \in \mathbb{N}\}$ such that $Q\phi^{(k)} = \lambda^{(k)}\phi^{(k)}$ and $\mathrm{Tr}(Q) = \sum_{k=1}^{\infty} \langle Q\phi^{(k)}, \phi^{(k)} \rangle_{\mathcal{H}} = \sum_{k=1}^{\infty} \lambda^{(k)} < \infty$.

$Q$-**Wiener process and Cameron-Martin space** In infinite-dimensional spaces, we distinguish two Gaussian driving noises. Let $\mathcal{U}$ be a separable Hilbert space with orthonormal basis $\{\phi^{(k)}\}_{k \geq 1}$. A *cylindrical Wiener process* is the natural extension of $\mathbb{R}$-valued Wiener process, formally

represented by the series $\mathbf{W}_t = \sum_{k=1}^{\infty} \mathbf{W}_t^{(k)} \phi^{(k)}$, where $\{\mathbf{W}_t^{(k)}\}_{k \geq 1}$ are independent Brownian motions. It satisfies independent increments $\mathbf{W}_{t+\Delta_t} - \mathbf{W}_t \sim \mathcal{N}(0, \Delta_t \mathbf{I}_{\mathcal{U}})$ but the covariance $\mathbf{I}_{\mathcal{U}}$ is *not* trace class when $\dim \mathcal{U} = \infty$. In other words, $\mathcal{N}(0, \Delta_t \mathbf{I}_{\mathcal{U}})$ is not supported in $\mathcal{U}$ itself, thereby $\mathbf{W}_t$ is *not* an $\mathcal{U}$-valued random variable. Here, we embed $\mathcal{U}$ into another Hilbert space $\mathcal{H}$ and define an Hilbert–Schmidt operator $Q^{1/2} : \mathcal{U} \to \mathcal{H}$ such that $Q = Q^{1/2}(Q^{1/2})^{\dagger}$ is trace class on $\mathcal{H}$. Then, we can introduce the following $\mathcal{H}$-valued covariated noise process:

$$\mathbf{W}_t^Q := Q^{1/2}\mathbf{W}_t = \sum_{k=1}^{\infty} \sqrt{\lambda^{(k)}}\mathbf{W}_t^{(k)}\phi^{(k)} \in \mathcal{H}, \quad (1)$$

where $\lambda^{(k)}$ is an eigen-value of $Q$. Consequently, we obtain, $\mathbf{W}_{t+\Delta_t}^Q - \mathbf{W}_t^Q \sim \mathcal{N}(0, \Delta_t Q)$ with trace class operator $Q$. We call $\mathbf{W}_t^Q$ as $Q$-*Wiener process*, which is now $\mathcal{H}$-valued random variable. To formalize the set of admissible control directions associated with $\mathbf{W}_t^Q$, we introduce the *Cameron-Martin Space* $\mathcal{H}_0 \subseteq \mathcal{H}$ equipped with inner product:

$$\langle u, v \rangle_{\mathcal{H}_0} = \langle Q^{-1/2}u, Q^{-1/2}v \rangle_{\mathcal{H}}, \quad \forall u, v \in \mathcal{H}_0. \quad (2)$$

**Infinite dimensional stochastic differential equations**
With the infinite dimensional Wiener processes defined in (1), we introduce the infinite-dimensional stochastic differential equations as follows (Da Prato and Zabczyk, 2014):

$$d\mathbf{X}_t = \mathcal{A}\mathbf{X}_t dt + \sigma_t d\mathbf{W}_t^Q, \quad \mathbf{X}_0 = \mathbf{x}_0 \in \mathcal{H}, \quad (3)$$

where $\mathcal{A} : \mathcal{H} \to \mathcal{H}$ is a linear operator and $\sigma_t : \mathbb{R}_{>0} \to \mathbb{R}_{>0}$ is a noise schedule. Under the mild conditions in Assumption B.1, the dynamics in (3) generates an Ornstein–Uhlenbeck $C_0$-semigroup (Da Prato and Zabczyk, 2002) and admit the unique *mild* solution. The resulting stochastic process $\{\mathbf{X}_t\}_{t \geq 0}$ is then an $\mathcal{H}$-valued Gaussian process with trace-class covariance operator:

$$\mathbf{X}_t \sim \nu_t^{\mathbf{x}_0} := \mathcal{N}(e^{t\mathcal{A}}\mathbf{x}_0, Q_t), \quad (4)$$

where $Q_t = \int_0^t e^{(t-s)\mathcal{A}}\sigma_s^2 Q e^{(t-s)\mathcal{A}^{\dagger}} ds$. Finally, as $t \to \infty$, the law of $\mathbf{X}_\infty$ converges to the Gaussian invariant law

$$\nu_\infty = \mathcal{N}(0, Q_\infty), \text{ with } Q_\infty = -\tfrac{1}{2}\sigma_\infty^2 \mathcal{A}^{-1}Q. \quad (5)$$

## 3. Functional Adjoint Sampler

In this section, we present a scalable diffusion sampler that operates directly in infinite-dimensional Hilbert spaces, targeting a *distribution of functions*. Full proofs and necessary derivations are provided in Appendix B.

### 3.1. Non-Equilibrium Sampling in Function Spaces

Sampling from a Gibbs distribution is a fundamental task. The goal is to draw samples from a target distribution $\pi$ that

is known only through an unnormalized energy $U : \mathcal{H} \to \mathbb{R}$:

$$d\pi(\mathbf{x}) = \frac{1}{\mathcal{Z}}e^{-U(\mathbf{x})}d\nu(\mathbf{x}), \quad \mathcal{Z} = \int_{\mathcal{H}} e^{-U(\mathbf{x})}d\nu(\mathbf{x}), \quad (6)$$

where $\nu$ is the measure defined on some space $\mathcal{H}$. In finite dimensions, *e.g.*, $\mathcal{H} = \mathbb{R}^d$, the Gibbs density can be expressed with respect to Lebesgue measure *i.e.*, $d\pi(\mathbf{x}) \propto e^{-U(\mathbf{x})}d\mathbf{x}$. However, due to the absence of an equivalent formulation of Lebesgue measure for Hilbert spaces, the expression in (6) is well-defined only after specifying an appropriate reference measure $\nu$. The following lemma states a condition for the target law to be well defined for arbitrary Hilbert spaces.

**Lemma 1** (Finite Normalization Constant). *Let us assume* $-U(\mathbf{x}) \leq \beta \|\mathbf{x}\|_{\mathcal{H}}^2 - C$ *with constants* $\beta, C > 0$. *Then, for any* $Tr(Q) < \infty$, *choosing* $\nu = \mathcal{N}(0, Q)$ *in* (6) *results in a finite normalization constant* $\mathcal{Z} < \infty$.

From Lemma 1, we have a simple yet effective choice of reference measure $\nu$, the centered Gaussian $\mathcal{N}(0, Q)$ with trace-class covariance $Q$. This ensures that $\mathcal{Z} < \infty$, and, consequently, the target $\pi$ in (6) is well-defined on $\mathcal{H}$.

To sample from (6) on $\mathcal{H}$, it is natural to consider an $\mathcal{H}$-valued stochastic process that follows the gradient flow of $U$ (Da Prato et al., 1996; Suzuki, 2020)—in the same spirit of overdamped Langevin dynamics in $\mathbb{R}^d$. Concretely, one may define the infinite-dimensional dynamics as follows:

$$d\mathbf{X}_t = [\mathcal{A}\mathbf{X}_t - QD_{\mathbf{x}}U(\mathbf{x})]dt + \sigma_t d\mathbf{W}_t^Q, \quad (7)$$

Convergence of the dynamics (7) into the target measure (6) is guaranteed when $\sigma_t := \sigma$, and direct simulation via space-time discretization of the $\mathcal{H}$-valued dynamics is standard. However, the resulting dynamics are of limited practical use as the mixing is only effective near equilibrium *i.e.*, $t \to \infty$, because they rely on asymptotic convergence to the invariant reference measure $\nu_\infty$ in (5) required for the well-defined normalizing constant $\mathcal{Z} < \infty$ by following Lemma 1.

**Non equilibrium sampling (NES)** Hence, to reach the target distribution (6) within a finite time horizon $T < \infty$, we recast our sampling problem as a NES problem. Unlike the Langevin-type processes in (7), which approach the target only asymptotically at equilibrium with reference $\nu_\infty$, we employ an importance-sampling (IS) via change of measure to sample the target efficiently in finite time:

$$d\pi(\mathbf{x}) \propto e^{-U(\mathbf{x})}\frac{d\nu_\infty(\mathbf{x})}{d\nu_T^{\mathbf{x}_0}(\mathbf{x})}d\nu_T^{\mathbf{x}_0}(\mathbf{x})$$

$$= e^{-U(\mathbf{x}) - \log \frac{d\nu_T^{\mathbf{x}_0}(\mathbf{x})}{d\nu_\infty(\mathbf{x})}}d\nu_T^{\mathbf{x}_0}(\mathbf{x}), \quad (8)$$

where $\nu_T^{\mathbf{x}_0}$ is marginal law of the reference dynamics (3) at time $T$. Hence, for the NES in infinite dimensions we need

an extra correction term in the importance weight, given by the Radon–Nikodým derivative (RND). It corrects the mismatch between the two Gaussian measures so that after reweighting, samples drawn from $\nu_T^{\mathbf{x}_0}$ have the desired target $\pi$, while guarantees the $\pi$ to be well-defined. However, compared to finite dimensional cases, the infinite dimensional RND exists only when the two measures are exactly compatible, because of absence of Lesbesgue measure.

When this compatibility condition fails, the two measures are singular. Hence, no density can be written explicitly. Fortunately, under Ornstein–Uhlenbeck semigroup generated by (3), $\nu_T^{\mathbf{x}_0}$ and $\nu_\infty$ are mutually absolutely continuous Gaussian measures (Da Prato and Zabczyk, 2002). Consequently, we can derive the explicit RND, as provided below.

**Theorem 2** (Explicit RND). *Suppose Assumption B.1 holds. Then, for any $0 < t < \infty$ and initial condition $\mathbf{x}_0 \in \mathcal{H}$, $\nu_t^{\mathbf{x}_0} = \mathcal{N}(e^{t\mathcal{A}}\mathbf{x}_0, Q_t)$ and $\nu_\infty = \mathcal{N}(0, Q_\infty)$ are mutually absolutely continuous. Moreover, for any $\mathbf{x} \in \mathcal{H}$, we obtain the explicit RND $q_t(\mathbf{x}_0, \mathbf{x}) := \frac{d\nu_t^{\mathbf{x}_0}}{d\nu_\infty}(\mathbf{x})$ defined as follows:*

$$q_t(\mathbf{x}_0, \mathbf{x}) = det(\Theta_t)^{-\frac{1}{2}} \exp \left[ -\frac{1}{2} \langle \Theta_t^{-1} \mathbf{m}_t^{\mathbf{x}_0}, \mathbf{m}_t^{\mathbf{x}_0} \rangle_\infty \right.$$
$$\left. + \langle \Theta_t^{-1} \mathbf{m}_t^{\mathbf{x}_0}, \mathbf{x} \rangle_\infty - \frac{1}{2} \langle e^{2t\mathcal{A}} \Theta_t^{-1} \mathbf{x}, \mathbf{x} \rangle_\infty \right], \quad (9)$$

*where we define $\mathbf{m}_t^{\mathbf{x}_0} = e^{t\mathcal{A}}\mathbf{x}_0$, $\Theta_t = 1 - e^{2t\mathcal{A}}$ and $Q_\infty$ norm $\langle u, v \rangle_\infty := \langle Q_\infty^{-\frac{1}{2}} u, Q_\infty^{-\frac{1}{2}} v \rangle_\mathcal{H}$.*

Theorem 2 provides an explicit RND, so IS can in principle generate samples from $\pi$ in (6). In practice, unfortunately, the importance weights are computed on a finite domain discretization, and when that discretization is refined the number of samples required grows exponentially proportional to discretization. Therefore, IS might become impractical.

### 3.2. Infinite-Dimensional Stochastic Optimal Control

Inspired by recent advances in diffusion samplers (Zhang and Chen, 2022; Vargas et al., 2023; Berner et al., 2024; Liu et al., 2026; Choi et al., 2026), we reformulate the NES problem as a variational inference problem on path space $\Omega = C([0, T], \mathcal{H})$ to build an efficient sampler by introducing a learnable *control* to steer the process (3) to the $\pi$ in finite horizon $T < \infty$. Let us introduce a target path measure $\mathbb{P}^\star$, which is an extension of target $\pi$ in (6) into $\Omega$:

$$d\mathbb{P}^\star(\mathbf{X}) \propto e^{-U(\mathbf{X}_T) - \log q_T(\mathbf{x}_0, \mathbf{X}_T)} d\mathbb{P}(\mathbf{X}), \quad (10)$$

where $\mathbb{P}$ is path measure induced by the process in (3). Now, we consider variational path measures $\mathbb{P}^\alpha$ induced by infinite-dimensional SDE with *control* (Fabbri et al., 2017):

$$d\mathbf{X}_t^\alpha = \left[ \mathcal{A}\mathbf{X}_t^\alpha + \sigma_t Q^{1/2} \alpha(\mathbf{X}_t^\alpha, t) \right] dt + \sigma_t d\mathbf{W}_t^Q \quad (11)$$

where $\alpha : \mathcal{H} \times [0, T] \to \mathcal{U}$ such that $Q^{1/2}\alpha \in \mathcal{H}_0 \subseteq \mathcal{H}$ is a Markov control functional. The primary reason for studying this path measure formulation is that the Gibbs-type target distribution $\pi$ in (6) can be effectively induced by choosing *optimal* control $\alpha_t^\star$ in (11). By invoking generalized Girsanov theorem (Da Prato and Zabczyk, 2014, Theorem 10.14), such an optimal control can be found by solving the following infinite-dimensional optimization problem (Fuhrman, 2003; Theodorou et al., 2018):

$$\min_\alpha \mathbb{E}_{\mathbb{P}^\alpha} \left[ \int_0^T \frac{1}{2} \|\alpha_s\|_\mathcal{H}^2 \, ds + g(\mathbf{X}_T^\alpha) \right] \text{ s.t. } (11), \quad (12)$$

where $g(\mathbf{x}) = U(\mathbf{x}) + \log q_T(\mathbf{x}_0, \mathbf{x})$. Accordingly, the optimal control $\alpha_t^\star$ which minimize the objective in (11) is equivalent to minimizer of Kullback-Liebler divergence *i.e.*, $\alpha^\star = \min_\alpha D_{KL}(\mathbb{P}^\alpha | \mathbb{P}^\star)$ with target path measure $\mathbb{P}^\star$ in (10). In other word, this formulation enables us to the exact sampling from target $\pi$, namely $\mathbf{X}_T^{\alpha^\star} \sim \pi$ in finite time $T < \infty$. This successfully extends the finite-dimensional control-based NES found in (Tzen and Raginsky, 2019; Zhang and Chen, 2022) into the Hilbert spaces $\mathcal{H}$.

### 3.3. Infinite Dimensional Adjoint Matching

The SOC problem in (12) follows the *least-action* principle in path space, where the goal is to find the *global* optimal control minimizes the terminal cost $g(\mathbf{x})$ while minimizing the action. In practice, solving this global problem by direct path-wise minimization is expensive because every update requires gradients through the full trajectory (Havens et al., 2025). A practical workaround is to move to the dual formulation. *Adjoint Matching* (AM; Domingo-Enrich et al., 2025) offers a scalable route by reformulating the global SOC problem into the local matching problem.

Extending AM to infinite dimensions requires a rigorous definition of the *first-order adjoint process* that appears in the corresponding infinite-dimensional SOC formulation. Below, we will demonstrate that a specific mathematical concept in SOC called *Stochastic Maximum Principle* (SMP; Bensoussan, 1983) provides existence and uniqueness of the adjoint process via coupled *Forward-Backward SDEs* (FBSDEs; Du and Meng, 2013) system and necessary condition.

**Lemma 3** (Stochastic Maximum Principle). *Consider the general infinite-dimensional SOC problem:*

$$\min_{\alpha \in \mathbb{A}} \mathbb{E}_{\mathbb{P}^\alpha} \left[ \int_0^T l(t, \mathbf{X}_t^\alpha, \alpha_t) ds + g(\mathbf{X}_T^\alpha) \right], \quad (13)$$

$$s.t. \, d\mathbf{X}_t^\alpha = [\mathcal{A}\mathbf{X}_t^\alpha + f(t, \mathbf{X}_t^\alpha, \alpha_t)] dt + \sigma(t, \mathbf{X}_t^\alpha) d\mathbf{W}_t^Q.$$

*With fixed initial condition $\mathbf{X}_0^\alpha = \mathbf{x}_0$. Now, let Assumption B.1-B.2 holds and define the Hamiltonian functional $H : [0, T] \times \mathcal{H} \times \mathcal{U} \times \mathcal{H} \times \mathcal{H} \to \mathbb{R}$:*

$$H(t, \mathbf{X}, \alpha, \mathbf{Y}, \mathbf{Z}) := l(t, \mathbf{X}, \alpha) + \langle \mathbf{Y}, f(t, \mathbf{X}, \alpha) \rangle_\mathcal{H} \quad (14)$$
$$+ \langle \mathbf{Z}, \sigma(t, \mathbf{X}) \rangle_\mathcal{H}.$$

*Suppose* $\mathbf{X}^\star$ *is the optimally controlled process driven by optimal control* $\alpha^\star$. *Then, the following first-order adjoint infinite-dimensional backward SDEs has a unique solution:*

$$\begin{aligned}
\mathrm{d}\mathbf{Y}_t^\star = &- \left[\mathcal{A}^\dagger \mathbf{Y}_t^\star + D_\mathbf{x} H(t, \mathbf{X}_t^\star, \alpha_t^\star, \mathbf{Y}_t^\star, \mathbf{Z}_t^\star)\right]\mathrm{d}t \quad (15) \\
&+ \mathbf{Z}_t^\star \mathrm{d}\mathbf{W}_t^Q, \quad \mathbf{Y}_T^\star = D_\mathbf{x} g(\mathbf{X}_T^\star)
\end{aligned}$$

*Moreover, we get necessary condition of optimal control* $\alpha_t^\star = \arg\min_{\alpha \in \mathbb{A}} H(t, \mathbf{X}_t^\star, \alpha, \mathbf{Y}_t^\star, \mathbf{Z}_t^\star)$.

Lemma 3 states that, for the general SOC problem (13), there exists a first-order adjoint pairs $(\mathbf{Y}_t, \mathbf{Z}_t)$ solving a backward SDE in (15). This result provides a *necessary condition* for optimality: any control that violates the backward BSDE in (15) or the point-wise minimization of Hamiltonian for all $t \in [0, T]$ cannot be optimal control.

In other words, SMP provides a *point-wise* optimality condition determined by the corresponding Hamiltonian $H$, allowing optimal control $\alpha_t^\star$ to be computed *locally* rather than *path-wise* computation. This principle can be effective, since solving path-wise minimization in (12) is often computationally prohibitive. With this local characterization in hand, we now derive a proper training objective for the infinite-dimensional SOC problem (12) based on the SMP. By applying Lemma 3 into our SOC problem in (12), the corresponding Hamiltonian $H$ is defined by:

$$\begin{aligned}
H(t, \mathbf{X}_t^{\bar{\alpha}}, \bar{\alpha}_t, \mathbf{Y}_t^{\bar{\alpha}}, \mathbf{Z}_t^{\bar{\alpha}}) := &\tfrac{1}{2}\|\bar{\alpha}_t\|_\mathcal{H}^2 + \langle \mathbf{Y}_t^{\bar{\alpha}}, \sigma_t Q^{1/2}\bar{\alpha}_t \rangle_\mathcal{H} \\
&+ \langle \mathbf{Z}_t^{\bar{\alpha}}, \sigma_t Q^{1/2} \rangle_\mathcal{H}, \quad (16)
\end{aligned}$$

with a fixed state control pair $(\mathbf{X}^{\bar{\alpha}}, \bar{\alpha})$. Note that the Hamiltonian $H$ in (16) is convex with respect to the control $\bar{\alpha}_t$, hence any critical point is global minimum. It implies that if $(\mathbf{X}^{\bar{\alpha}}, \bar{\alpha})$ solves the backward SDEs in (15), then the optimal control is characterized by the first-order stationarity of $H$:

$$\begin{aligned}
\nabla_\alpha H(t, \mathbf{X}_t^{\bar{\alpha}}, \alpha_t, \mathbf{Y}_t^{\bar{\alpha}}, \mathbf{Z}_t^{\bar{\alpha}}) &= \alpha_t + \langle \mathbf{Y}_t^{\bar{\alpha}}, \sigma_t Q^{1/2} \rangle_\mathcal{H} \quad (17) \\
&= \nabla_\alpha \tfrac{1}{2}\left\|\alpha_t + \sigma_t Q^{1/2}\mathbf{Y}_t^{\bar{\alpha}}\right\|_\mathcal{H}^2.
\end{aligned}$$

It follows that, with the state variables $(\mathbf{X}_t^{\bar{\alpha}}, \mathbf{Y}_t^{\bar{\alpha}}, \mathbf{Z}_t^{\bar{\alpha}})$ remain fixed for the chosen control $\bar{\alpha}_t$, the minimization over $\alpha_t$ reduces to the convex quadratic in (17), whose unique minimizer is $\alpha_t = -\sigma_t Q^{1/2}\mathbf{Y}_t^{\bar{\alpha}}$. Leveraging this observation, we now state our main result:

**Proposition 4** (Adjoint Matching in $\mathcal{H}$). *Consider the following infinite-dimensional matching objective:*

$$\mathcal{L}(\theta) = \int_0^T \mathbb{E}_{\mathbb{P}^{\bar{\alpha}}}\left[\tfrac{1}{2}\left\|\alpha^\theta(\mathbf{X}_t^{\bar{\alpha}}, t) + \sigma_t Q^{1/2}\mathbf{Y}_t^{\bar{\alpha}}\right\|_\mathcal{H}^2\right]\mathrm{d}t \quad (18)$$

$$\mathrm{d}\mathbf{Y}_t^{\bar{\alpha}} = -\mathcal{A}^\dagger \mathbf{Y}_t^{\bar{\alpha}}\mathrm{d}t + \mathbf{Z}_t^{\bar{\alpha}}\mathrm{d}\mathbf{W}_t^Q, \ \mathbf{Y}_T^{\bar{\alpha}} = D_\mathbf{x} g(\mathbf{X}_T^{\bar{\alpha}}). \quad (19)$$

*where* $\bar{\alpha} := \mathtt{stopgrad}(\alpha^\theta)$. *Then, the critical point of* $\mathcal{L}$ *is optimal control* $\alpha^\star$ *for the SOC problem in (12).*

Compared to the original SOC problem (12), SMP-based matching objective in (18) is inherently scalable since it can avoids path-wise backpropagation and reduces training to point-wise convex minimizations with $\mathbf{X}_t^{\bar{\alpha}}$ held fixed. This local structure enables straightforward parallelization over time. Note that when $\mathcal{H} := \mathbb{R}^d$ with $Q = \mathbf{I}_d$, then the objective function in (18) recovers the *lean* AM objective in (Domingo-Enrich et al., 2025). In other words, our formulation strictly generalizes AM into the Hilbert spaces.

Yet, for $\mathcal{L}$ in (18), the stochastic term $\mathbf{Z}_t^{\bar{\alpha}}$ can incur a computational burden because this term requires simulating the BSDE in (19) to obtain the solution $\mathbf{Y}_t^{\bar{\alpha}}$, which markedly increases the cost of each optimization step. Since this BSDE is adapted to the filtration generated by the same $Q$-Wiener process $\mathbf{W}_t^Q$, we substitute the adjoint process $\mathbf{Y}_t^{\bar{\alpha}}$ with its conditional expectation $\mathbb{E}_{\mathbb{P}^{\bar{\alpha}}}[\mathbf{Y}_t^{\bar{\alpha}}|\mathbf{X}_t^{\bar{\alpha}}]$.

**Proposition 5** (Unbiased Estimator). *Define the sample-wise adjoint matching objective with conditional expectation* $\tilde{\mathbf{Y}}_t^{\bar{\alpha}} := \mathbb{E}_{\mathbb{P}^{\bar{\alpha}}}[\mathbf{Y}_t^{\bar{\alpha}}|\mathbf{X}_t^{\bar{\alpha}}]$ *for any sample trajectory* $\mathbf{X}_t^{\bar{\alpha}} \sim \mathbb{P}^{\bar{\alpha}}$:

$$\tilde{\mathcal{L}}(\theta) := \int_0^T \tfrac{1}{2}\|\alpha^\theta(\mathbf{X}_t^{\bar{\alpha}}, t) + \sigma_t Q^{1/2}\tilde{\mathbf{Y}}_t^{\bar{\alpha}}\|_\mathcal{H}^2 \mathrm{d}t \quad (20)$$

*Then, we get unbiased gradient estimator* $\frac{\mathrm{d}}{\mathrm{d}\theta}\mathbb{E}_{\mathbb{P}^{\bar{\alpha}}}[\tilde{\mathcal{L}}(\theta)] = \frac{\mathrm{d}}{\mathrm{d}\theta}\mathcal{L}(\theta)$ *with same critical point in (18).*

In practice, we can estimate $\tilde{\mathbf{Y}}_t^{\bar{\alpha}}$ from the trajectory saved during simulation of $\mathbf{X}_T^{\bar{\alpha}}$ for efficient computation. This preserves the same critical point $\alpha^\star$ for the SOC problem in (12) while reducing the computational burden because the substitution yields a closed form solution of the BSDE:

$$\begin{aligned}
\tilde{\mathbf{Y}}_t^{\bar{\alpha}} = \mathbb{E}_{\mathbb{P}^{\bar{\alpha}}}[\mathbf{Y}_T^{\bar{\alpha}}|\mathbf{X}_t^{\bar{\alpha}}] &= e^{-(T-t)\mathcal{A}^\dagger}\mathbb{E}_{\mathbb{P}^{\bar{\alpha}}}[D_\mathbf{x} g(\mathbf{X}_T^{\bar{\alpha}})|\mathbf{X}_t^{\bar{\alpha}}] \\
&\approx e^{-(T-t)\mathcal{A}^\dagger} D_\mathbf{x} g(\mathbf{X}_T^{\bar{\alpha}}). \quad (21)
\end{aligned}$$

Finally, since the optimal path measure $\mathbb{P}^\star$ in (6) admits the endpoint disintegration $\mathbb{P}^\star(\mathbf{X}_t) = \mathbb{P}^\star(\mathbf{X}_T)\mathbb{P}_{t|T}(\mathbf{X}_t \mid \mathbf{X}_T)$; see Appendix B.8 for details. Substituting these relations into (18) yields our proposed training objective for the efficient and scalable sampling in function spaces:

$$\mathcal{L}_{\mathrm{FAS}} = \int_0^T \mathbb{E}_{\mathbb{P}_{t|T}\mathbb{P}_T^{\bar{\alpha}}}\left[\tfrac{1}{2}\left\|\alpha^\theta(\mathbf{X}_t, t) + \mathcal{C}_t D_\mathbf{x} g(\mathbf{X}_T^{\bar{\alpha}})\right\|_\mathcal{H}^2\right]\mathrm{d}t, \quad (22)$$

where $\mathcal{C}_t := \sigma_t Q^{1/2}e^{-(T-t)\mathcal{A}^\dagger}$ Note that the path measure $\mathbb{P}^{\bar{\alpha}}$ induced by current control $\bar{\alpha}$ may not satisfy the endpoint disintegration. In this case, we can project the path measure to a conditional path measure that does satisfy the disintegration (Havens et al., 2025).

### 3.4. Numerical Computation

Because FAS operates on infinite-dimensional path spaces, both training and sampling require appropriate discretization and simulation schemes. This requires specifying the

---

**Algorithm 1** `Galerkin` SDE simulation

> **Input:** Initial condition $\mathbf{x}_0$, basis functions $\{\phi^{(k)}\}_{k=1}^K$.
> Set initial condition $\mathbf{X}_0^\alpha = \mathbf{x}_0$
> **for** time $t$ in range $[0, T]$ **do**
>      Estimate control $\alpha_t := \alpha(\mathbf{X}_t^\alpha, t)$
>      Sample Gaussian noise $\epsilon \sim \mathcal{N}(0, \mathbf{I}_K)$
>      $\{\mathbf{X}_t^{(k)}\}_{k \in K} \leftarrow$ `Galerkin expansion`$(\mathbf{X}_t^\alpha)$
>      $\{\alpha_t^{(k)}\}_{k \in K} \leftarrow$ `Galerkin expansion`$(\alpha_t)$
>      **for** $k = 1, \cdots, K$ **do in parallel**
>          Euler-Step for each $\mathbf{X}_t^{(k)}$ using (24)
>      **end for**
>      $\mathbf{X}_t^\alpha \leftarrow$ `Galerkin truncation`$(\{\mathbf{X}_{t+\mathrm{d}t}^{(k)}\}_{k \in K})$
> **end for**
> **Output:** $\mathbf{X}_T^\alpha \sim \mathbb{P}_T^\alpha$

**Algorithm 2** Functional Adjoint Sampler

> **Input:** Differentiable cost $g(\mathbf{x})$, parametrized $\alpha_\theta(\mathbf{x}, t)$
> Initialize $\alpha_\theta^{(0)} := 0$
> **for** step $m$ in $1, 2, \ldots$ **do**
>      Sample $t \in \mathcal{U}[0, T]$
>      $\bar{\theta} \rightarrow$ `Stopgrad`$(\theta)$
>      $\mathbf{X}_T^{\bar{\alpha}} \leftarrow$ `Galerkin SDE`$(\mathbf{x}_0, \alpha_{\bar{\theta}}, \{\phi^{(k)}\}_{k=1}^K)$
>      ▷ Algorithm 1 for `Galerkin` SDE
>      Compute adjoint $\mathbf{Y}_t^{\bar{\alpha}}$ using (21)
>      Sample $\mathbf{X}_{t|T}^{\bar{\alpha}} \sim \mathbb{P}_{t|T}(\cdot | \mathbf{X}_T^{\bar{\alpha}})$
>      ▷ cf. Appendix C.1 for bridge sampling
>      Update control $\alpha_\theta^{(m)}$ by minimizing (22)
> **end for**
> **Output:** $\alpha_\theta^{(M)} \approx \alpha^\star$

---

operators $\mathcal{A}$ and $Q$, which are also key modeling choices reflecting the domain structure of a given task. For NES, we adopt Assumption B.1, ensuring well-posed reference dynamics and yielding an explicit RND in Theorem 2. Hence, we restrict to a specific class of operator pairs $(\mathcal{A}, Q)$.

**Proposition 6** (Laplacian Operator). *For the negative Laplacian $\mathcal{A} = -\Delta$ on a bounded domain with Dirichlet boundary condition, take $Q = \mathcal{A}^{-s}$ with $s > \frac{d}{2}$. Then, $\mathcal{A}$ and $Q$ satisfy the conditions in Assumption B.1.*

**Boundary condition** Since the Laplacian $\Delta$ admits different realizations depending on the boundary conditions (BCs) (Evans, 2022), we restrict to Dirichlet BCs in this study. While Neumann BCs are common in other domains (Rissanen et al., 2023; Lim et al., 2023b; Park et al., 2024), our goal is conditional path sampling, where Dirichlet BCs are natural because they keep boundary values fixed. For example, in TPS, a transition path between metastable states $\mathbf{A}$ and $\mathbf{B}$ is meaningful when these states are fixed as boundary conditions *i.e.*, $\mathbf{X}_t[0] = \mathbf{A}$, $\mathbf{X}_t[L] = \mathbf{B}$. To do so, we use the Dirichlet sine basis for $k = [1, \ldots, K]$:

$$\phi^{(k)}[u] = \left(\frac{2}{L}\right)^{1/2} \sin\left(\frac{\pi k u}{L}\right), \quad u \in [0, L]. \quad (23)$$

This basis satisfies $\phi^{(k)}[0] = \phi^{(k)}[L] = 0$, and therefore naturally enforces zero boundary values for any path. In our TPS implementation, we write $\mathbf{X}_t = \mathbf{x}_0 + \mathbf{R}_t$, where $\mathbf{x}_0$ is a fixed reference path connecting $\mathbf{A}$ to $\mathbf{B}$ and $\mathbf{R}_t$ is expanded in the above sine-basis in (23). Hence, $\mathbf{R}_t[0] = \mathbf{R}_t[L] = 0$ for all $t$, so the reconstructed path always satisfies $\mathbf{X}_t[0] = \mathbf{A}$ and $\mathbf{X}_t[L] = \mathbf{B}$ for a path $\mathbf{X}_t[\cdot] : \mathbb{R}_{0>} \rightarrow \mathbb{R}^d$ and for any $t \in [0, T]$. We defer the details of enforcing BCs for TPS under our formulation to Appendix C.2.

**Numerical simulation** Simulating infinite-dimensional SDEs is a well-studied problem (Lord et al., 2014). Since it

often requires a finite-dimensional approximation, we use a Galerkin truncation scheme (Shardlow, 1999) by approximating the function with $K$ basis functions $\{\phi^{(k)}\}_{k=1}^K$:

$$\mathbf{X}_t \approx \sum_{k=1}^K \mathbf{X}_t^{(k)} \phi^{(k)}, \quad \text{(Galerkin truncation)}$$

$$\mathbf{X}_t^{(k)} := \langle \mathbf{X}_t, \phi^{(k)} \rangle_{\mathcal{H}} = \int_0^L \mathbf{X}_t[u] \phi^{(k)}[u] \mathrm{d}u. \quad \text{(Galerkin expansion)}$$

Hence, with the finite eigen-system $\{(\lambda^{(k)}, \phi^{(k)}) \in \mathbb{R} \times \mathcal{H} : k \in [1, \cdots, K]\}$ of Dirichlet Laplacian $\Delta$, the $\mathcal{H}$-valued controlled SDE (11) reduces to the $K$-dimensional coefficient SDEs after `Galerkin expansion` (GE):

$$\mathrm{d}\mathbf{X}_t^{(k)} = \left[ -\lambda^{(k)} \mathbf{X}_t^{(k)} + \sigma_t (\lambda^{(k)})^{-s/2} \langle \alpha(\mathbf{X}_t, t), \phi^{(k)} \rangle_{\mathcal{H}} \right] \mathrm{d}t + \sigma_t (\lambda^{(k)})^{-s/2} d\mathbf{W}_t^{(k)}, \quad (24)$$

where $\{\mathbf{W}^{(k)}\}_{k=1}^K$ are independent standard Brownian motions. In particular, since (24) is a finite-dimensional modewise diagonal system, we can simulate each Galerkin mode $\mathbf{X}_t^{(k)}$ independently using a standard numerical SDE integrator, as summarized in Algorithm 1. At each time step, we first expand the current path $\mathbf{X}_t^\alpha$ and control $\alpha$ onto the first $|K|$ basis functions, evolve the resulting coefficient SDEs in parallel, and then reconstruct the path by the corresponding $|K|$-mode `Galerkin truncation`. This `Galerkin` SDE simulation is used in the FAS training procedure, summarized in Algorithm 2. Appendix C.3 provides additional details for the TPS instantiation and corresponding DST-based `Galerkin` SDE simulation[2].

---

[2]Other parameterizations can be applied to different input domains with different BCs, such as image and PDE domains (a rectangular domain $\mathbb{R}^2$) or molecular conformations (a graph domain $\mathcal{G}$), as discussed in Appendix C.5.

# 4. Related Works

**Infinite-dimensional generative models** Many recent works extend finite-dimensional generative models to infinite-dimensional function spaces. Score-based diffusion models (Song et al., 2021) have been extended to function space (Lim et al., 2023a;b; Pidstrigach et al., 2023; Franzese et al., 2024). Later, matching-based generative models follow the same direction. For example, flow matching (Lipman et al., 2023) has been generalized to function space (Kerrigan et al., 2024), and Schrödinger bridge matching (Shi et al., 2024) has been adapted as well (Park et al., 2024). See (Franzese and Michiardi, 2025) for a survey on this topic. However, these algorithms operate in a *sample-to-sample* setting where paired samples from both distributions are available for training. To the best of our knowledge, scalable function-space generative modeling for *sample-to-energy* such as targeting sampling from a Gibbs distribution $\pi(\mathbf{x}) \propto e^{-U(\mathbf{x})}$ in (6), has not been investigated.

**Diffusion sampler** Diffusion samplers address the *sample-to-energy* setting, where the goal is to draw samples from the Gibbs distribution $\pi(\mathbf{x}) \propto e^{-U(\mathbf{x})}$. Based on SOC theory, this line of work reformulates sampling as an optimization problem and methods for learning such samplers have advanced rapidly (Zhang and Chen, 2022; Vargas et al., 2023; Berner et al., 2024; Zhu et al., 2026). Due to the target is specified only through a complex energy functional $U$, optimization tends to be computationally demanding. This challenge motivates adjoint-based samplers (Havens et al., 2025; Liu et al., 2026; Guo et al., 2026), which avoid simulation-based targets and yield a method that scales better. However, existing methods are limited to finite-dimensional settings, and extensions to function space remain unexplored, despite the appeal and practical importance.

**Transition path sampling** TPS aims to sample reactive trajectories connecting metastable states in molecular systems (Dellago et al., 1998; Vanden-Eijnden et al., 2010). Classical TPS and enhanced sampling methods construct Markov chains over trajectories or bias molecular dynamics to increase the probability of rare transitions (Izrailev et al., 1999), but they often require long simulations, carefully chosen collective variables, or expensive accept-reject procedures. Recent learning-based methods mitigate these issues by learning controlled dynamics or variational path distributions for TPS (Holdijk et al., 2022; Seong et al., 2025; Blessing et al., 2026). A related line of work instead optimizes discretized path functionals to find minimum-energy or high-probability transition paths (Petersen and Covino, 2025; Ramakrishnan et al., 2025; Raja et al., 2025). We leave introduction of transition path sampling and additional related work in Appendix A.3. Compared to prior works that operate after fixing a finite-discretization, FAS formulates TPS as sampling from a path-space Gibbs measure in (6) in Hilbert space $\mathcal{H}$. This Hilbert-space formulation separates the target path measure from its numerical discretization, allowing the learned sampler to be reused across different path resolutions.

# 5. Experiments

## 5.1. Experimental Setup

We evaluate FAS on three benchmarks for TPS. We start with a synthetic potential, then move to two molecular systems. For the synthetic case, following (Du et al., 2024), we use the standard TPS benchmark, the Müller–Brown potential. For molecular systems, we consider alanine dipeptide and chignolin, with 22 and 138 atoms in 3D, respectively. All molecular simulations use Cartesian coordinates, and we compute potential energy with OpenMM (Eastman et al., 2023). Further experimental details appear in Appendix C.

**Baselines and evaluation** We compare FAS with: unbiased MD (UMD) at elevated temperature, steered MD (SMD) with forces on selected collective variables, and deep learning methods PIPS (Holdijk et al., 2022), TPS-DPS (Seong et al., 2025), TR-LV (Blessing et al., 2026), and DL (Du et al., 2024). Note that DL uses a different training environments, so we report it only on the synthetic benchmark. We evaluate with metrics that capture reliability and probabilistic fidelity:

- **RMSD**: We first align the heavy atoms of the endpoint $\mathbf{X}[L]$ to the target state $\mathbf{B}$ with the Kabsch algorithm (Kabsch, 1976). Then calculate the root mean square deviation (**RMSD**) between the heavy atom positions of $\mathbf{X}[L]$ and $\mathbf{B}$.

- **THP**: It is the indicator-based success rate of trajectories reaching the target metastable state $\mathbf{B}$. Given endpoints $\{\mathbf{X}^{(i)}[L]\}_{i=1}^N$ from $N$ paths, we define **THP** as:

$$\mathbf{THP} = 100 \times \frac{|\{i : \mathbf{X}^{(i)}[L] \in \mathcal{B}\}|}{N}. \quad (25)$$

where $\mathcal{B} = \{\mathbf{X}[L] | \, \|\mathbf{X}[L] - \mathbf{B}\| < \epsilon\}$ and choose $\epsilon$ following (Seong et al., 2025).

- **ETS**: It measure how well the method find probable transition states when crossing the energy barrier. Given a transition path $\mathbf{X}$ of length $L$ that reaches $\mathcal{B}$, we define

$$\mathbf{ETS}(\mathbf{X}) = \max_{u \in [0,L]} V(\mathbf{X}[u]). \quad (26)$$

On the synthetic potential, we follow (Du et al., 2024) and additionally report the log likelihood (**LLK**) of a transition path distribution (TPD) defined in (27).

*Table 1.* Comparison on synthetic potential. We report the metrics averaged over 64 paths. Best results are highlighted.

| Methods | **THP** (% ↑) | **ETS** (↓) | **LLK** (↑) |
|---|---|---|---|
| MCMC | - | $-13.77_{\pm16.43}$ | - |
| DL (Du et al., 2024) | 82 | $-14.93_{\pm12.61}$ | $10.42_{\pm0.42}$ |
| FAS (**Ours**) | 100 | $-36.70_{\pm3.09}$ | $11.48_{\pm0.05}$ |

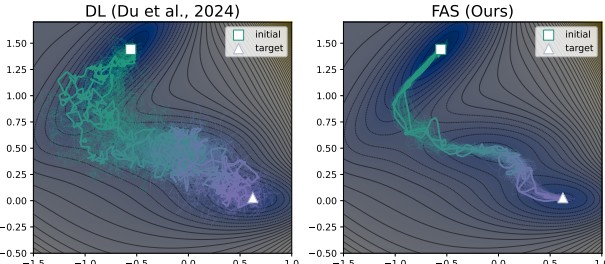

*Figure 1.* **TPS for synthetic potential.** Sampled transiton path on energy landscape.

**Energies** We adopt the TPD described in (Holdijk et al., 2022; Du et al., 2024) for the Müller-Brown potential:

$$\mathbb{P}(\mathbf{X}) = \mathbf{1_A}(\mathbf{X}[0])\mathbf{1_B}(\mathbf{X}[L]) \tag{27}$$

$$\prod_{u=0}^{L-1} \mathcal{N}\left(\mathbf{X}[u+1]\Big|\mathbf{X}[u] - \tfrac{1}{\gamma\mathbf{m}}\nabla_\mathbf{X} V(\mathbf{X}[u])\delta_u, \tfrac{2k_B T}{\gamma\mathbf{m}}\mathbf{I}_{3N}\delta_u\right),$$

where both $\mathbf{1_A}$ and $\mathbf{1_B}$ are indicator functions and $V$ is potential function. For molecular systems, we observed that TPS in (27) led to unstable training because it relies on local potential gradients. We therefore derive a TPD based on the Feynman-Kac formula (Del Moral, 2011), which removes the reliance on local gradients:

$$\mathbb{P}(\mathbf{X}) = \mathbf{1_A}(\mathbf{X}[0])\mathbf{1_B}(\mathbf{X}[L]) \tag{28}$$

$$\prod_{u=0}^{L-1} e^{-\frac{1}{k_B T}V(\mathbf{X}[u])}\mathcal{N}\left(\mathbf{X}[u+1]\big|\mathbf{X}[u], \tfrac{2k_B T}{\gamma\mathbf{m}}\mathbf{I}_{3N}\delta_u\right).$$

We provide further discussions on the choice of TPD in Appendix A.6.

### 5.2. Experimental Results

**Müller Brown potential** The Müller Brown potential is a widely adopted benchmark for TPS and features three local minima. Following the setup in (Du et al., 2024), we aim to sample transition paths connecting the top left state to the bottom right state. Table 1 reports the results on the the Müller Brown potential. FAS attains lower **ETS** and higher **LLK** than baselines, showing it yields reliable paths that remain in low energy valleys as shown in Figure 1.

**Alanine dipeptide** We evaluate FAS on alanine dipeptide, a peptide with two alanine residues, and target transitions

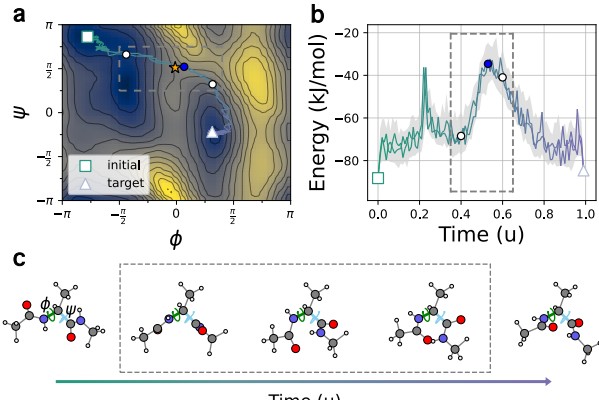

*Figure 2.* **TPS on alanine dipeptide.** (a) Projected transition paths sampled by FAS on the conformational landscape as a function of CVs. The saddle point that separates the two-meta stable states is depicted as ★ (orange star) and the highest energy along paths sampled from FAS depicted as ● (blue circle). (b) Potential energy plot. Grey box highlights the transition-state region where paths reach the **ETS**. (c) Visualization of molecule transition trajectory for given meta-stable states C5 (far left) and C7ax (far right).

between meta-stable states C5 and C7ax. We use torsion angles $\phi$ and $\psi$ as collective variables (CVs) from atomic coordinates. Table 2 summarizes alanine dipeptide results. Notably, our function space approaches achieve perfect **THP** with zero **RMSD** due to the Dirichlet boundary formulation enforces these endpoints by construction. Perfect **THP** implies that all sampled paths are reactive, which maximizes compute use and prevents bias in transition rate and mechanism estimates hence lowers variance of path statistics. Moreover, FAS also attain significantly lower **ETS** than baselines, implying the path generated by FAS is reliable.

Moreover, as shown in Figure 2a, the free energy surface over these CVs shows a high barrier with a saddle point. Reliable reactive paths are expected to cross near this saddle because it is the lowest barrier route. The highest energy along paths sampled from FAS occurs close to the saddle, which supports that the sampled paths are physically meaningful transition paths. Figure 2c provides a visualization of configuration for the molecule transition.

**Chignolin** Finally, we evaluate sampling of the folding dynamics of the small protein Chignolin. As shown in Table 2, FAS achieves the lowest **ETS**. Because of chignolin has a higher dimensional and more rugged landscape with a narrow folding funnel and multiple intermediates basin, baselines show larger **RMSD** and lower **THP** than on alanine dipeptide. Baselines that rely on gradient driven or steered updates often traverse strained conformations. This inflates **RMSD** even after Kabsch alignment and lowers the chance of reaching the folded basin within the time horizon. In contrast, FAS enforces the endpoints throughout and searches

*Table 2.* Comparison between baselines on the molecular transition path sampling of the alanine dipeptide and chignolin. We report the metrics averaged over 64 paths. Best results are highlighted.

| Method | Alanine Dipeptide | | | Chignolin | | |
|---|---|---|---|---|---|---|
| | **RMSD** (Å ↓) | **THP** (% ↑) | **ETS** (kJ/mol ↓) | **RMSD** (Å ↓) | **THP** (% ↑) | **ETS** (kJ/mol ↓) |
| UMD | $1.19_{\pm0.03}$ | 6.25 | $812.47_{\pm148.80}$ | $7.23_{\pm0.93}$ | 1.56 | 388.17 |
| SMD (Izrailev et al., 1999) | $0.56_{\pm0.27}$ | 54.69 | $78.40_{\pm12.76}$ | $0.85_{\pm0.24}$ | 34.38 | $179.52_{\pm138.87}$ |
| PIPS (Holdijk et al., 2022) | $0.66_{\pm0.15}$ | 43.75 | $28.17_{\pm10.86}$ | $4.66_{\pm0.17}$ | 0.00 | - |
| TPS-DPS (Seong et al., 2025) | $0.47_{\pm0.18}$ | $39.58_{\pm28.13}$ | $46.34_{\pm10.16}$ | $1.06_{\pm0.08}$ | $25.00_{\pm10.69}$ | $-189.81_{\pm23.01}$ |
| TR-LV (Blessing et al., 2026) | $0.29_{\pm0.03}$ | $61.25_{\pm4.05}$ | $49.11_{\pm5.84}$ | $0.90_{\pm0.01}$ | $43.95_{\pm5.64}$ | $-303.98_{\pm28.65}$ |
| FAS (**Ours**) | $0.00_{\pm0.00}$ | $100.00_{\pm0.00}$ | $-29.46_{\pm3.87}$ | $0.00_{\pm0.00}$ | $100.00_{\pm0.00}$ | $-360.85_{\pm53.71}$ |

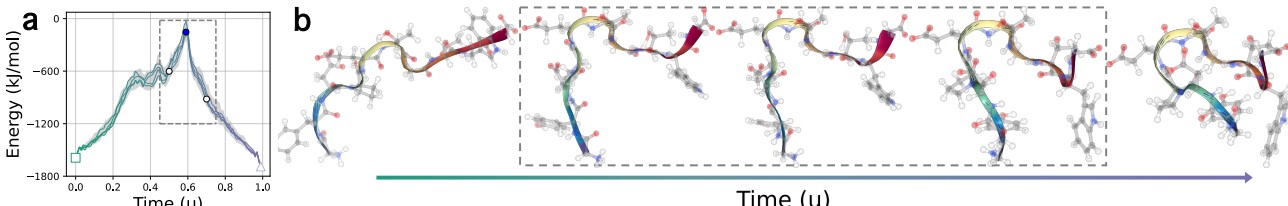

*Figure 3.* **TPS on Chignolin.** (a) Potential energy plot. (b) Visualization of protein folding trajectory from unfolded (far left) to folded (far right). Grey box highlights the transition-state region where trajectories reach the **ETS**.

only within admissible paths. This boundary enforcement removes endpoint drift and avoids late corrective pulls. As a result, transitions are smoother and lead to lower **ETS**. We illustrate visualization of the folding dynamics in Figure 3b.

### 5.3. Discussions

**Discretization invariance sampling** Infinite dimensional formulation enables inference on paths at arbitrary time discretizations. Table 3 reports **ETS** across varying resolutions. Specifically, we train $\alpha^\theta$ with a discretization of implicit time interval $[0,1]$ into 100 uniform time steps (denoted ×1) and evaluate on unseen, more finer time discretization (denoted ×2-100). Across both potentials, FAS maintains nearly unchanged **ETS** on unseen resolutions, which indicate discretization-agnostic behavior. In particular, for the synthetic Müller Brown potential, **ETS** and **LLK** remain stable even when the resolution increases by ×100, where $[0,1]$ is discretized into $10^4$ steps. Compared to synthetic potential, alanine dipeptide exhibits higher **ETS** variability on unseen discretizations because molecular potentials are highly sensitive to configuration. We leave additional discussions on discretization invariance sampling and further ablation studies in Appendix C.8.

## 6. Conclusion

We introduced the Functional Adjoint Sampler (FAS), a scalable diffusion sampler for Gibbs-type distributions on Hilbert spaces. FAS generalizes adjoint matching algorithms (Domingo-Enrich et al., 2025; Havens et al., 2025)

*Table 3.* **ETS** across various unseen discretization levels. **Disc** represents how fine the generated path is, where ×r multiplies the base number of discretization by $r$.

| | Müller Brown | | Alanine Dipeptide | |
|---|---|---|---|---|
| **Disc** | **ETS**(kJ/mol ↓) | **LLK** (↑) | **Disc** | **ETS**(kJ/mol ↓) |
| ×1 | $-36.70_{\pm3.09}$ | $11.48_{\pm0.05}$ | ×1 | $-29.46_{\pm3.87}$ |
| ×10 | $-38.55_{\pm1.41}$ | $11.65_{\pm0.02}$ | ×2 | $-28.72_{\pm41.52}$ |
| ×100 | $-37.83_{\pm3.17}$ | $11.72_{\pm0.09}$ | ×4 | $-33.58_{\pm35.59}$ |

to infinite-dimensional Hilbert spaces, enabling efficient sampling over functional representations such as paths evaluated on a one-dimensional time grid, and it demonstrates superior performance on path sampling tasks. Despite these promising results, we focused on a one dimensional domain. However, FAS readily extends to other tasks such as PDE generation and graph based domains, and we leave a comprehensive study for future work.

## Acknowledgments

The authors would like to thank Benjamin Kurt Miller, Juno Nam, Yuanqi Du, and Ricky T. Q. Chen for the helpful discussions and comments. This work was partly supported by Institute for Information & Communications Technology Planning & Evaluation(IITP) grant funded by the Korea government(MSIT) (No.RS-2019-II190075, Artificial Intelligence Graduate School Program(KAIST), No.RS-2024-00509279, Fundamental Research in Artificial Intelligence, No.RS-2022-II220713, Meta-learning Applicable to Real-world Problems).

## Impact Statement

This paper aims to advance the field of machine learning. Our work may have societal implications, but we do not identify any that warrant specific emphasis here.

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

# A. Overview of Relevant Concepts

We first give a concise overview of core probabilistic notions in infinite dimensional spaces, then describe stochastic differential equations on Hilbert spaces (Da Prato and Zabczyk, 2002; 2014).

### A.1. Gaussian Measure and Generalized Wiener Processes.

Gaussian measures are central because the infinite dimensional setting lacks a Lebesgue reference measure, and a Gaussian with a trace class covariance provides a natural baseline that keeps stochastic integrals square integrable and provides explicit RND, both essential for our framework.

**Gaussian Measure.** Let $(\Omega, \mathcal{F}, \mathbb{Q})$ be a probability space and $(\mathcal{H}, B(\mathcal{H}))$ be a measurable state spaces where $\mathcal{H}$ be a separable Hilbert space with inner product $\langle \cdot, \cdot \rangle_{\mathcal{H}}$ and norm $\|\cdot\|_{\mathcal{H}}$ and $\mathcal{B}(\mathcal{H})$ is Borel measure on $\mathcal{H}$. We consider the $\mathcal{H}$-valued random variable $\mathbf{X} : \Omega \to \mathcal{H}$. Now, define the push-forward measure $\nu := \mathbf{X}_{\#}\mathbb{Q}$ by $\nu(\mathbf{B}) = \mathbb{Q}(\mathbf{X}^{-1}(\mathbf{B}))$ for every Borel set $\mathbf{B} \subset B(\mathcal{H})$. We call $\nu$ Gaussian if every $\mathbb{R}$-valued projection $\langle u, \mathbf{X} \rangle_{\mathcal{H}}$ is a Gaussian random variable for all $u \in \mathcal{H}$. When $\mathcal{H} = \mathbb{R}^d$ with the standard inner product, this definition reproduces the usual Gaussian distribution.

For a Gaussian measure $\nu$, there exists a unique mean function $\mathbf{m} \in \mathcal{H}$ that statisfies:

$$\mathbf{m} = \int_{\mathcal{H}} \mathbf{X} d\nu, \quad \langle \mathbf{m}, u \rangle_{\mathcal{H}} = \mathbb{E}_{\nu} \left[ \langle \mathbf{X}, u \rangle_{\mathcal{H}} \right], \quad \forall u \in \mathcal{H}. \tag{29}$$

Moreover, there exists a unique bounded linear operator $Q : \mathcal{H} \to \mathcal{H}$ such that

$$Qu = \int_{\mathcal{H}} \langle \mathbf{X}, u \rangle_{\mathcal{H}} \mathbf{X} d\nu - \langle \mathbf{m}, u \rangle_{\mathcal{H}} \mathbf{m}, \quad \forall u \in \mathcal{H}. \tag{30}$$

A Gaussian measure is fully specified by its mean and covariance operator. If the mean is zero, so that $\nu = \mathcal{N}(0, Q)$, we say that $\nu$ is *centred*. The Fernique's integrability theorem (Da Prato and Zabczyk, 2014, Theorem 2.7) states that for any centred Gaussian measure $\nu$ on a separable Banach space $\mathcal{B}$ (in particular on $\mathcal{H}$) there exists $r > 0$ such that $\int_{\mathcal{H}} \exp(r \|\mathbf{X}\|_{\mathcal{H}}^2) d\nu < \infty$. Consequently, it implies that every polynomial moment is finite *i.e.*, $\mathbb{E}_{\nu}[\|\mathbf{X}\|_{\mathcal{H}}^2] < \infty$. Using definition (30) and an arbitrary orthonormal basis of $\mathcal{H}$, $\{\phi^{(k)}\}_{k \geq 1}$ for each $\phi^{(k)} \in \mathcal{H}$ for every $k \in \mathbb{N}$:

$$\mathrm{Tr}(Q) = \sum_{k \geq 1} \langle Q\phi^{(k)}, \phi^{(k)} \rangle_{\mathcal{H}} = \sum_{k \geq 1} \mathbb{E}_{\nu} \left[ \langle \mathbf{X}, \phi^{(k)} \rangle_{\mathcal{H}}^2 \right] = \mathbb{E}_{\nu} \left[ \|\mathbf{X}\|_{\mathcal{H}}^2 \right] < \infty. \tag{31}$$

This implies that the symmetric and non-negative covariance operator $Q$ is trace class, *i.e.*, $\mathrm{Tr}(Q) < \infty$, hence implies that $Q$ is compact. Hence, there exists the eigen-system $\{(\lambda^{(k)}, \phi^{(k)}) \in \mathbb{R} \times \mathcal{H} : k \in \mathbb{N}\}$, where $\{\phi^{(k)}\}_{k \geq 1}$ is orthonormal eigen-basis and $\{\lambda^{(k)}\}_{k \geq 1}$ is positive eigenvalues such that

$$Q\phi^{(k)} = \lambda^{(k)}\phi^{(k)}, \quad \text{and} \quad \mathrm{Tr}(Q) = \sum_{k \geq 1} \langle Q\phi^{(k)}, \phi^{(k)} \rangle_{\mathcal{H}} = \sum_{k \geq 1} \lambda^{(k)} < \infty. \tag{32}$$

We now extend the notion of a Wiener process to an infinite dimensional SDE.

**Wiener Processes on Hilbert Spaces.** A natural way to generalize the Wiener process to a Hilbert space $\mathcal{H}$ is through the *cylindrical Wiener process*. It is introduced as the formal series of independent $\mathbb{R}$-valued Wiener processes $\{\mathbf{W}^{(k)}\}_{k \geq 1}$ along an orthonormal basis $\{\phi^{(k)}\}_{k \geq 1}$:

$$\mathbf{W}_t = \sum_{k \geq 1} \mathbf{W}_t^{(k)} \phi^{(k)}, \quad t \geq 0. \tag{33}$$

However, the series $\mathbf{W}_t$ dose not converge in $\mathcal{H}$ because

$$\mathbb{E} \left[ \|\mathbf{W}_t\|_{\mathcal{H}}^2 \right] = \sum_{k \geq 1} \mathbb{E} \left[ (\mathbf{W}_t^{(k)})^2 \right] = t \sum_{k \geq 1} 1 = \infty, \tag{34}$$

thereby $\mathbf{W}_t$ fails to be an $\mathcal{H}$-valued random variable. Hence, it can be used only under additional smoothing to obtain $\mathcal{H}$-valued noise process. This smoothing is done with trace class operator $Q$. Define $\mathbf{W}_t^Q = Q^{1/2}\mathbf{W}_t =$

$\sum_{k\geq 1}\sqrt{\lambda^{(k)}}\mathbf{W}_t^{(k)}\phi^{(k)}$. Then, we have

$$\mathbb{E}\left[\left\|\mathbf{W}_t^Q\right\|_{\mathcal{H}}^2\right] = \sum_{k\geq 1}\mathbb{E}\left[\lambda^{(k)}(\mathbf{W}_t^{(k)})^2\right] = t\sum_{k\geq 1}\lambda^{(k)} < \infty, \tag{35}$$

where $\{(\lambda^{(k)},\phi^{(k)})\in\mathbb{R}\times\mathcal{H}:k\in\mathbb{N}\}$ is eigen-system of $Q$. Hence $\mathbf{W}_t^Q$ is $\mathcal{H}$-valued process. This smoothed process is called a *Q-Wiener process* or *coloured noise* because the noise covariance operator $Q$ introduces a spatial correlation.

## A.2. Stochastic Differential Equations in Hilbert Spaces

With the infinite dimensional Wiener process $\mathbf{W}_t^Q$ well defined $\mathcal{H}$, we then define the infinite-dimensional SDEs as follows:

$$d\mathbf{X}_t = \mathcal{A}\mathbf{X}_t dt + \sigma_t d\mathbf{W}_t^Q, \quad \mathbf{X}_0 = \mathbf{x}_0 \in \mathcal{H}, \tag{36}$$

where $\mathcal{A}:\mathcal{H}\to\mathcal{H}$ is a linear operator and $\sigma_t > 0$. We recall the classical mild formulation driven by a general $Q$-Wiener process on $\mathcal{H}$. We then restrict attention to a cylindrical Wiener process on $\mathcal{H}_0$, introduce an appropriate Hilbert–Schmidt embedding that can be naturally aligned with our framework, and outline the minimal assumptions that still guarantee existence of a solution for the infinite-dimensional SDE under the choice of cylindrical Wiener process.

**Mild solution.** Let a linear operator $\mathcal{A}:\mathcal{H}\to\mathcal{H}$ generate a strongly continuous $C_0$ semi-group $\{e^{t\mathcal{A}}\}_{t\geq 0}$ on $\mathcal{H}$. Assume $\mathbf{W}^Q$ is a centred $Q$-Wiener process with trace-class covariance operator $Q$ and that $e^{t\mathcal{A}}Q^{1/2}$ is Hilbert-Schmidt for any $t>0$. Under these conditions, the infinite-dimensional SDE (or stochastic PDE) in (3) admit the unique mild solution:

$$\mathbf{X}_t = e^{t\mathcal{A}}\mathbf{x}_0 + \int_0^t e^{(t-s)\mathcal{A}}\sigma_s d\mathbf{W}_s^Q, \quad t \geq 0. \tag{37}$$

The resulting stochastic process $\{\mathbf{X}_t\}_{t\geq 0}$ is then $\mathcal{H}$-valued random variable since the stochastic convolution $\int_0^t e^{(t-s)\mathcal{A}}d\mathbf{W}_s^Q = \int_0^t e^{(t-s)\mathcal{A}}Q^{1/2}d\mathbf{W}_s$ is lies on $\mathcal{H}$ because $e^{(t-s)\mathcal{A}}Q^{1/2}$ is Hilbert-Schmidt. We can compute its mean $\mathbf{m}_t$ and covariance operator $Q_t$ for any $t>0$ as follows:

$$\mathbf{m}_t = e^{t\mathcal{A}}\mathbf{x}_0, \quad Q_t = \int_0^t \sigma_s^2 e^{(t-s)\mathcal{A}}Qe^{(t-s)\mathcal{A}^\dagger}ds. \tag{38}$$

Due to the Hille-Yosida Theorem (Da Prato and Zabczyk, 2014, Theorem A.3), $\mathcal{A}$ satisfies $\left\|e^{t\mathcal{A}}\right\|\leq Me^{-wt}$ for some $M\geq 1$, $w\geq 0$. Hence, for every $0\leq s\leq t$, $T_{st}:=e^{(t-s)\mathcal{A}}Qe^{(t-s)\mathcal{A}^\dagger}$ satisfies:

$$\mathrm{Tr}(T_{st}) \leq \left\|e^{(t-s)\mathcal{A}}\right\|\mathrm{Tr}(Q)\left\|e^{(t-s)\mathcal{A}^\dagger}\right\| \leq M^2 e^{-2w(t-s)}\mathrm{Tr}(Q) < \infty. \tag{39}$$

Implies that $T_{st}$ is a trace class operator. Therefore, since the bound is integrable on $[0,t]$,

$$\mathrm{Tr}(Q_t) = \mathrm{Tr}(\sigma_t^2\int_0^t T_{st}ds) \leq \sigma_t^2\int_0^t \mathrm{Tr}(T_{st})ds < \infty. \tag{40}$$

Hence $Q_t$ is a trace-class operator for every $t<\infty$. Hence, we can conform that the $\mathcal{H}$-valued stochastic process $\mathbf{X}_t$ has a Gaussian law $\nu_t = \mathcal{N}(\mathbf{m}_t, Q_t)$.

**Invariant Measure.** Let $\mathcal{A}$ is dissipative (*i.e.*, $\langle\mathcal{A}\mathbf{x},\mathbf{x}\rangle_{\mathcal{H}}\leq -\lambda\|\mathbf{x}\|_{\mathcal{H}}^2$ for some $\lambda>0$), the integral converges as $t\to\infty$ and we define the noise schedule such that $\sup_t\sigma_t<\infty$, $\sigma_t\to\sigma_\infty$. Then, the operator $Q_t$ solves the following differential Lyapunov equation:

$$\dot{Q}_t = \mathcal{A}Q_t + Q_t\mathcal{A}^\dagger + \sigma_t^2 Q, \quad Q_t = e^{t\mathcal{A}}Q_0 e^{t\mathcal{A}^\dagger} + \int_0^t \sigma_{t-r}^2 e^{r\mathcal{A}}Qe^{r\mathcal{A}^\dagger}dr. \tag{41}$$

Because $Q_0 = 0$ since we assume Dirac delta initial condition $\mathbf{x}_0$ in (3), we obtain

$$Q_t \overset{t\to\infty}{\to} Q_\infty := \sigma_\infty^2\int_0^\infty e^{t\mathcal{A}}Qe^{t\mathcal{A}^\dagger}dt \tag{42}$$

Now, let $F(t) = \sigma_\infty^2 e^{t\mathcal{A}} Q e^{t\mathcal{A}^\dagger}$. Then, we get:

$$\tfrac{\mathrm{d}}{\mathrm{d}t} F(t) = \mathcal{A}F(t) + F(t)\mathcal{A}^\dagger = \mathcal{A}\sigma_\infty^2 e^{t\mathcal{A}} Q e^{t\mathcal{A}^\dagger} + \sigma_\infty^2 e^{t\mathcal{A}} Q e^{t\mathcal{A}^\dagger} \mathcal{A}^\dagger. \tag{43}$$

Since $F(t) \overset{t\to\infty}{\to} 0$ and $F(0) = \sigma_\infty^2 Q$, by integrating (43) from 0 to $\infty$, we get:

$$\mathcal{A}Q_\infty + Q_\infty \mathcal{A}^\dagger = \int_0^\infty \tfrac{\mathrm{d}}{\mathrm{d}t} F(t)\mathrm{d}t = \lim_{t\infty} F(t) - F(0) = -\sigma_\infty^2 Q \tag{44}$$

Hence, we get the operator Lyapunov equation (Da Prato and Zabczyk, 2002, Proposition 10.1.4):

$$\mathcal{A}Q_\infty + Q_\infty \mathcal{A}^\dagger + \sigma_\infty^2 Q = 0, \tag{45}$$

where $Q_\infty$ is trace class operator. so $\mathcal{N}(0, \infty)$ is the unique invariant Gaussian measure of the process in (3). Finally, if $\mathcal{A}$ is trace-class, self-adjoint operator (*i.e.*, $\mathcal{A} = \mathcal{A}^\dagger$) and $\mathcal{A}$ and $Q$ are commute (*i.e.*, $Q\mathcal{A} = \mathcal{A}Q$), we get $Q_\infty = -\tfrac{1}{2}\sigma_\infty^2 \mathcal{A}^{-1} Q$ (Da Prato and Zabczyk, 2002, Proposition 10.1.6).

### A.3. Brief Introduction of the Transtiion Path Theory

In this section, we introduce the classic Traditional *transitional path theory* (TPT; Vanden-Eijnden et al., 2010), a method to sample and quantify reactive pathways meta stable states between **A** and **B** in complex system. Here, we breifely introduce two representative methods for TPT, *transition path sampling* (TPS) and *chain of states* (CoS) methods (*e.g.*, NEB, string). Then, we will make explicit connection between these two viewpoints and show how our infinite-dimensional formulation provides a principled bridge between them by using the Feynman-Kac formula (Del Moral, 2011).

Following the notion of (Dellago et al., 1998), let us denote a discrete space-time path $\mathbf{P} := \{\mathbf{p}_0, \mathbf{p}_1, \cdots \mathbf{p}_L\}$, where $\mathbf{p}_u = (\mathbf{r}_u, \mathbf{v}_u) \in \mathbb{R}^{6N}$ is configuration with $\mathbf{r}_u \in \mathbb{R}^{3N}$ the atom-wise positions and $\mathbf{v}_u \in \mathbb{R}^{3N}$ the atom-wise velocities with $N$ number of atom on time interval $\mathbf{L} = [0, L]$. We assume that consecutive configuration states follow a Markov transition $p(\mathbf{p}_u \to \mathbf{p}_{u+1})$, then the path probability is given by product of the transition probabilities as shown in below:

$$\mathbb{P}(\mathbf{P}) = e^{-\frac{1}{k_B T} V(\mathbf{p}_0)} \prod_{u=0}^{L-1} p(\mathbf{p}_u \to \mathbf{p}_{u+1}), \tag{46}$$

where $V : \mathbb{R}^{6N} \to \mathbb{R}$ is the energy function. In this setting, our goal is to sample the transition path bridging boundary constraints $\pi_\mathbf{A}(\mathbf{p}_0)$ and $\pi_\mathbf{B}(\mathbf{p}_L)$ from the transition path distribution:

$$\mathbb{P}^\star(\mathbf{P}) := e^{-S(\mathbf{P})} = \pi_\mathbf{A}(\mathbf{p}_0)\pi_\mathbf{B}(\mathbf{p}_L)\mathbb{P}(\mathbf{P}), \tag{47}$$

where $\pi_\mathbf{A}(\mathbf{p}_0)$ and $\pi_\mathbf{B}(\mathbf{p}_L)$ enforces the path to be started and terminated with prescribed **A** and **B**, respectively. Note that we can choose Markov transition $p(\mathbf{p}_u \to \mathbf{p}_{u+1})$ freely, if it conserves the Gibbs distribution normalized (Dellago et al., 1998). In following, we show how the choice of Markov transition differentiating the TPS from CoS-based *maximum energy path* (MEP) optimization.

### A.4. Transition Path Sampling

Classic TPS is usually formulated for Langevin dynamics in the under-damped regime. However we focus on the over-damped case for its simple form, and the extension to the under-damped case is straightforward. The molecule system governed by over-damped Langevin dynamics is given by:

$$\mathrm{d}\mathbf{p}_t = -\tfrac{1}{\gamma\mathbf{m}}\nabla_\mathbf{p} V(\mathbf{p}_t)\mathrm{d}t + \sqrt{\tfrac{2k_B T}{\gamma\mathbf{m}}}\mathrm{d}\mathbf{W}_t, \tag{48}$$

where we denote $\gamma$ as friction and $\mathbf{m}$ as a mass. In this case, the discrete Markov transition kernel of overdamped dynamics in (48) follows from an Euler-Maruyama discretization, since the increment over $\delta_t$ is Gaussian with mean $\mu_t = \mathbf{p}_t - \tfrac{1}{\gamma\mathbf{m}}\nabla_\mathbf{p} V(\mathbf{p}_t)\delta_t$ and covariance $\Sigma_t = \tfrac{2k_B T}{\gamma\mathbf{m}}\delta_t \mathbf{I}_{3N}$:

$$\mathbf{p}_{u+1} = \mathbf{p}_u - \tfrac{1}{\gamma\mathbf{m}}\nabla_\mathbf{p} V(\mathbf{p}_t)\delta_t + \sqrt{\tfrac{2k_B T}{\gamma\mathbf{m}}\delta_t}\zeta, \quad \zeta \sim \mathcal{N}(0, \mathbf{I}_{3N}). \tag{49}$$

$$\text{Hence, } p(\mathbf{p}_u \to \mathbf{p}_{u+1}) = \mathcal{N}(\mathbf{p}_{u+1}|\mu_t, \Sigma_t). \tag{50}$$

Hence, substituting (49) into (46) yields the TPS transition path distribution, which is widely used in recent literature (Holdijk et al., 2022; Seong et al., 2025; Blessing et al., 2026). Concretely, starting from the overdamped dynamics in (48), this line of work sets up a finite-dimensional SOC problem by biasing the potential $V$ with a neural-network based control $\alpha^\theta$:

$$\mathrm{d}\mathbf{p}_t^\alpha = -\tfrac{1}{\gamma\mathbf{m}}\nabla_\mathbf{P}\left[V(\mathbf{p}_t^\alpha) + \alpha^\theta(t, \mathbf{p}_t^\alpha)\right]\mathrm{d}t + \sqrt{\tfrac{2k_BT}{\gamma\mathbf{m}}}\mathrm{d}\mathbf{W}_t, \quad \mathbf{p}_0^\alpha = \mathbf{A}. \tag{51}$$

Now, the goal of the optimization is to learn $\alpha^\theta$ that steers the prior dynamics start from a given metastable state $\mathbf{A}$ into another target metastable state $\mathbf{B}$ by maximizing the log-likelihood with respect to the TPD in (47).

### A.5. Chain of States Methods

If the given system has smooth energy landscape, TPS provides an adequate description of the transition process. In this case, the main target is the *transition state*, which is a saddle point on the potential energy landscape (Vanden-Eijnden et al., 2010). However, for a complex energy landscape, when the initial and target states are separated by intermediates basins, the MEP that bridges the relevant minima through the sequence of transition states is preferred, which motivates CoS methods. The objective of CoS is to parametrize the geometric path as discrete chain $\mathbf{P}^\theta := \{\mathbf{p}_0 = \mathbf{A}, \mathbf{p}_1^\theta, \cdots, \mathbf{p}_{L-1}^\theta, \mathbf{p}_L = \mathbf{B}\}$ over the grid with fixed boundary and to optimize a geometric functional. For example, the objective of elastic-band method (Jónsson et al., 1998) is given by:

$$S(\mathbf{P}^\theta) = \sum_{u=0}^{L-1}\left[V(\mathbf{p}_u^\theta) + \tfrac{\kappa}{2}\left\|\mathbf{p}_{u+1}^\theta - \mathbf{p}_u^\theta\right\|^2\right], \tag{52}$$

where $\kappa > 0$ is constant. Recent studies (Petersen and Covino, 2025; Ramakrishnan et al., 2025) optimize the neural-network parameterization of continuous path $\mathbf{P}^\theta$ by minimizing objectives of the similar form in (52) to find the MEP. However, CoS formulations lack a probabilistic foundation, so they cannot target the reactive path ensemble or uncertainty without additional modeling.

### A.6. Choice of the Terminal Cost in (12)

Previously, the transition path distribution of TPS is often derived by plugging the Langevin transition kernel (49) into the factorized form (46) as follows:

$$\mathbb{P}(\mathbf{P}) = e^{-\frac{1}{k_BT}V(\mathbf{p}_0)}\prod_{u=0}^{L-1}p(\mathbf{p}_u \to \mathbf{p}_{u+1}), \tag{53a}$$

$$\text{where} \quad p(\mathbf{p}_u \to \mathbf{p}_{u+1}) = \mathcal{N}(\mathbf{p}_{u+1}|\mathbf{p}_u - \tfrac{1}{\gamma\mathbf{m}}\nabla_\mathbf{p}V(\mathbf{p}_u)\delta_u, \tfrac{2k_BT}{\gamma\mathbf{m}}\mathbf{I}_{3N}\delta_u). \tag{53b}$$

Intuitively, (53a) gives the path log-likelihood of a discrete trajectory under the overdamped Langevin dynamics (48) at temperature $T$. Then, to enforce the endpoint condition $\pi_\mathbf{B}(\mathbf{p}_L)$ for TPS:

$$\mathbb{P}_{\mathrm{TPS}}^\star(\mathbf{P}) = e^{-S(\mathbf{P})} = \pi_\mathbf{A}(\mathbf{p}_0)\pi_\mathbf{B}(\mathbf{p}_L)\mathbb{P}(\mathbf{P}), \tag{54}$$

For the terminal cost $U$ in (12), one can take the TPS energy in (54) as a negative log-likelihood (NLL) *i.e.*, $U(\mathbf{X}) = -\log\mathbb{P}_{\mathrm{TPS}}^\star(\mathbf{X})$, and then optimize the control $\alpha^\theta$ to sample transition paths. In practice, however, we observed that this choice led to instability during FAS training. Indeed, the simulation of Langevin dynamics in (48) moves according to the local gradient $-\nabla V$, which helps stabilize trajectories, because this force steers paths toward descending the potential and suppress large uphill moves. In contrast, our infinite-dimensional SDE in (11) injects noise and evolves on a fixed training grid without explicitly aligning the path with geometric of $-\nabla V$, so small perturbations of the path can cause large oscillations in $V$ along the grid, thereby $U$ fluctuates across gradient steps.

Hence, we instead derive a principled path-objective that is well-suited for our path parameterization. Compared to TPS, by adopting well-known reweighting principle of Markov process known as the Feynman-Kac formula (Del Moral, 2011), we can rewrite the path distribution as *tilted* Markov chain:

$$\tilde{\mathbb{P}}(\mathbf{P}) = \prod_{u=0}^{L-1}e^{-\frac{1}{k_BT}V(\mathbf{p}_u)}\tilde{p}(\mathbf{p}_u \to \mathbf{p}_{u+1}), \tag{55a}$$

$$\text{where} \quad \tilde{p}(\mathbf{p}_u \to \mathbf{p}_{u+1}) = \mathcal{N}(\mathbf{p}_{u+1}|\mathbf{p}_u, \tfrac{2k_BT}{\gamma\mathbf{m}}\mathbf{I}_{3N}\delta_u). \tag{55b}$$

Note that, the potential $V(\mathbf{p}_u)$ is carried by the path distribution rather than by the dynamics. Although the goal of (55) is same as (53), which is to concentrate the resulting path measure in high-potential region, this construction differs from (53), the high-potential can be achieved by direct reweighting, not from the local gradient $-\nabla V$. This viewpoint enable us to derive the CoS type objective (52) that is suited to our path-parametrization, leads to the transition path distribution:

$$\mathbb{P}_{\text{CoS}}^{\star}(\mathbf{P}) \propto e^{-\tilde{S}(\mathbf{P})} = \pi_{\mathbf{A}}(\mathbf{p}_0)\pi_{\mathbf{B}}(\mathbf{p}_L)\tilde{\mathbb{P}}(\mathbf{P}), \tag{56}$$

With this transition path distribution, the NLL $U(\mathbf{X}) = -\log \mathbb{P}_{\text{CoS}}^{\star}(\mathbf{X})$ can be written as follows:

$$U(\mathbf{X}) = -\left[\log \pi_{\mathbf{A}}(\mathbf{X}[0]) + \pi_{\mathbf{B}}(\mathbf{X}[L])\right] + \frac{1}{k_B T}\sum_{u=0}^{L-1}\left[V(\mathbf{X}[u]) + \frac{\kappa}{2}\|\mathbf{X}[u+1] - \mathbf{X}[u]\|^2\right] \tag{57}$$

$$\stackrel{(i)}{=} C + \frac{1}{k_B T}\underbrace{\sum_{u=0}^{L-1}\left[V(\mathbf{X}[u]) + \frac{\kappa}{2}\|\mathbf{X}[u+1] - \mathbf{X}[u]\|^2\right]}_{\text{CoS objective described in (52)}} \tag{58}$$

where $\kappa = \frac{1}{2}\frac{\gamma \mathbf{m}}{\delta_u}$ and $(i)$ follows from a fixed endpoints of our methods. Note that the second term on the right-hand side equals the CoS objective in (52). Hence (56) can be viewed as a probabilistic interpretation of classic elastic band methods. Compared to the NLL based on (54), (57) is better aligned with a path parameterization. It pushes every grid point $\mathbf{X}[u]$ for all $u \in \mathbf{L}$ toward low values of $V$ and penalizes rough increments, which damps oscillations of $V$ and yields smoother gradients, thereby training is more stable particular on sharp potentials such as molecular systems.

## B. Proofs and Derivations

**Assumption B.1.** *We assume a linear operator $\mathcal{A}: \mathcal{H} \to \mathcal{H}$ that generates a $C_0$-semigroup and is exponentially stable, dissipative i.e., $\langle \mathcal{A}\mathbf{x}, \mathbf{x}\rangle_{\mathcal{H}} \leq -\lambda\|\mathbf{x}\|_{\mathcal{H}}^2$ for some $\lambda > 0$ and self-adjoint i.e., $\mathcal{A} = \mathcal{A}^{\dagger}$. We also take a Q-Wiener process with a self-adjoint trace-class operator $Q$ that commutes with $\mathcal{A}$ i.e., $Q\mathcal{A} = \mathcal{A}Q$.*

**Assumption B.2.** *For each $(\mathbf{x}, \alpha) \in \mathcal{H} \times \mathcal{U}$, the maps $t \mapsto f(t, \mathbf{x}, \alpha)$, $t \mapsto g(t, \mathbf{x}, \alpha)$, and $t \mapsto l(t, \mathbf{x}, \alpha)$ are predictable. The function $h(\mathbf{x})$ is $\mathcal{F}_T$-measurable. For each $(t, \alpha, \omega) \in [0, T] \times \mathcal{U} \times \Omega$, the functions $f$, $g$, $l$, and $h$ are globally twice Fréchet differentiable with respect to $\mathbf{x}$. The derivatives $D_{\mathbf{x}}f$, $D_{\mathbf{x}}g$, $D_{\mathbf{xx}}f$, $D_{\mathbf{xx}}g$, $D_{\mathbf{xx}}l$, and $D_{\mathbf{xx}}h$ are continuous in $\mathbf{x}$ and uniformly bounded by a constant $K$. Moreover, we assume the growth bounds:*

$$|f(t, \mathbf{x}, \alpha)| + \|g(t, \mathbf{x}, \alpha)\| + \|D_{\mathbf{x}}l(t, \mathbf{x}, \alpha)\| + \|D_{\mathbf{x}}h(\mathbf{x})\| \leq K(1 + \|\mathbf{x}\|_{\mathcal{H}} + |\alpha|_{\mathcal{U}}), \tag{59}$$

$$|l(t, \mathbf{x}, \alpha)| + |h(\mathbf{x})| \leq K(1 + \|\mathbf{x}\|_{\mathcal{H}} + |\alpha|_{\mathcal{U}}^2). \tag{60}$$

**Assumption B.3.** *The function $\mathcal{V}: [0, T] \times \mathcal{H} \to \mathbb{R}$ and its derivatives $D_{\mathbf{x}}\mathcal{V}, D_{\mathbf{xx}}\mathcal{V}, \partial_t\mathcal{V}$ are uniformly continuous on bounded subsets of $[0, T] \times \mathcal{H}$ and $(0, T) \times \mathcal{H}$, respectively. Moreover, for all $(t, \mathbf{x}) \in (0, T) \times \mathcal{H}$, there exists $C_1, C_2 > 0$ such that*

$$|\mathcal{V}(t, \mathbf{x})| + |D_{\mathbf{x}}\mathcal{V}(t, \mathbf{x})| + |\partial_t\mathcal{V}(t, \mathbf{x})| + \|D_{\mathbf{xx}}\mathcal{V}(t, \mathbf{x})\| + |\mathcal{A}^{\dagger}D_{\mathbf{x}}\mathcal{V}(t, \mathbf{x})| \leq C_1(1 + |\mathbf{x}|)^{C_2}. \tag{61}$$

### B.1. Preliminary

We recall two key ingredients. The first is the product rule for Hilbert space semimartingales, also called stochastic integration by parts. We use this rule to derive the adjoint matching objective. The second is a verification theorem that we use to establish the optimality.

**Lemma B.4** (Stochastic integration by parts in $\mathcal{H}$). *Assume $A, C$ are $\mathcal{H}$–valued predictable processes with $\int_0^T \mathbb{E}\|A_s\|^2\,\mathrm{d}s < \infty$ and $\int_0^T \mathbb{E}\|C_s\|^2\,\mathrm{d}s < \infty$ for every $T > 0$ and $B, D$ are predictable processes with values in the Hilbert–Schmidt space $\mathcal{L}_2(\mathcal{U}_Q, \mathcal{H})$ and $\int_0^T \mathbb{E}\|B_s\|_{\mathcal{L}_2}^2\,\mathrm{d}s < \infty$, $\int_0^T \mathbb{E}\|D_s\|_{\mathcal{L}_2}^2\,\mathrm{d}s < \infty$ for every $T > 0$. Then, let us define two stochastic convolutions:*

$$\mathbf{X}_t = \mathbf{X}_0 + \int_0^t A_s\mathrm{d}s + \int_0^t B_s\mathrm{d}\mathbf{W}_s^Q, \quad \mathbf{Y}_t = \mathbf{Y}_0 + \int_0^t C_s\mathrm{d}s + \int_0^t D_s\mathrm{d}\mathbf{W}_s^Q \tag{62}$$

*Then, for all $t \geq 0$, almost surely, we get:*

$$\langle \mathbf{X}_t, \mathbf{Y}_t \rangle_{\mathcal{H}} = \langle \mathbf{X}_0, \mathbf{Y}_0 \rangle_{\mathcal{H}} + \int_0^t \langle \mathbf{Y}_s, A_s \rangle_{\mathcal{H}} \mathrm{d}s + \int_0^t \langle \mathbf{X}_s, C_s \rangle_{\mathcal{H}} \mathrm{d}s \tag{63}$$

$$+ \int_0^t \langle \mathbf{Y}_s, B_s d\mathbf{W}_s^Q \rangle_{\mathcal{H}} + \int_0^t \langle \mathbf{X}_s, D_s d\mathbf{W}_s^Q \rangle_{\mathcal{H}} + \int_0^t Tr\left[ B_s Q D_s^{\dagger} \right] \mathrm{d}s. \tag{64}$$

*Equivalently, in differential form,*

$$\mathrm{d}\langle \mathbf{X}_t, \mathbf{Y}_t \rangle_{\mathcal{H}} = \langle \mathbf{Y}_t, A_t \rangle_{\mathcal{H}} \mathrm{d}t + \langle \mathbf{X}_t, C_t \rangle_{\mathcal{H}} \mathrm{d}t + \langle \mathbf{Y}_t, B_t \mathrm{d}\mathbf{W}_t^Q \rangle_{\mathcal{H}} + \langle \mathbf{X}_t, D_t \mathrm{d}\mathbf{W}_t^Q \rangle_{\mathcal{H}} + Tr\left[ B_t Q D_t^{\dagger} \right] \mathrm{d}t.$$

*Proof.* Let us define $\mathbb{R}$-valued process $\mathbf{Z}_t := \langle \mathbf{X}_t, \mathbf{Y}_t \rangle_{\mathcal{H}}$ and $C^2$ mapping $f : \mathcal{H} \times \mathcal{H} \to \mathbb{R}$ such that $f(\mathbf{x}, \mathbf{y}) = \langle \mathbf{x}, \mathbf{y} \rangle_{\mathcal{H}}$ for any $\mathbf{x}, \mathbf{y} \in \mathcal{H}$. Since

$$D_{\mathbf{x}}f(\mathbf{x}, \mathbf{y}) = \mathbf{y}, D_{\mathbf{y}}f(\mathbf{x}, \mathbf{y}) = \mathbf{x}, D_{\mathbf{xx}}f(\mathbf{x}, \mathbf{y}) = 0, D_{\mathbf{yy}}f(\mathbf{x}, \mathbf{y}) = 0, \tag{65}$$

and $D_{\mathbf{xy}}f(\mathbf{x}, \mathbf{y})$ is the continuous bilinear form $(h, k) \to \langle h, k \rangle_{\mathcal{H}}$. Now, write the semi-martingale decomposition on $\mathcal{H} \times \mathcal{H}$:

$$(\mathbf{X}_t, \mathbf{Y}_t) = (\mathbf{X}_0, \mathbf{Y}_0) + \int_0^t (A_s, B_s) \mathrm{d}s + \int_0^t \tilde{B}_s \mathrm{d}\mathbf{W}_s^Q, \tag{66}$$

where $\tilde{B}_s : \mathcal{U} \to \mathcal{H} \times \mathcal{H}$ such that $\tilde{B}_s u = (B_s u, D_s u)$. By applying the infinite-dimensional Itô formula (Da Prato and Zabczyk, 2014, Chapter 4.4) for $f$ on $\mathcal{H} \times \mathcal{H}$ gives:

$$\mathrm{d}\mathbf{Z}_t = \langle D_{\mathbf{x}}f(\mathbf{X}_t, \mathbf{Y}_t), d\mathbf{X}_t \rangle_{\mathcal{H}} + \langle D_{\mathbf{y}}f(\mathbf{X}_t, \mathbf{Y}_t), d\mathbf{Y}_t \rangle_{\mathcal{H}} + \tfrac{1}{2} Tr\left[ D_{\mathbf{xy}}f(\mathbf{X}_t, \mathbf{Y}_t)(\tilde{B}_t Q \tilde{B}_t^{\dagger}) \right] \mathrm{d}t, \tag{67}$$

where we compute the trace by expanding the function in any orthonormal basis $\{\phi^{(k)}\}_{k \geq 1}$ of $\mathcal{U}$:

$$\tfrac{1}{2} Tr\left[ D_{\mathbf{xy}}f(\mathbf{X}_t, \mathbf{Y}_t)(\tilde{B}_t Q \tilde{B}_t^{\dagger}) \right] = \sum_{k \geq 1} \lambda^{(k)} \left( \langle B_t, \phi^{(k)} \rangle_{\mathcal{H}} + \langle D_t, \phi^{(k)} \rangle_{\mathcal{H}} \right) = Tr\left[ B_t Q D_t^{\dagger} \right], \tag{68}$$

where $\lambda^{(k)}$ is eigen-value corresponding eigen-basis $\phi^{(k)}$ such that $Q\phi^{(k)} = \lambda^{(k)}\phi^{(k)}$. Therefore,

$$\mathrm{d}\langle \mathbf{X}_t, \mathbf{Y}_t \rangle_{\mathcal{H}} = \langle \mathbf{Y}_t, \mathrm{d}\mathbf{X}_t \rangle_{\mathcal{H}} + \langle \mathbf{X}_t, \mathrm{d}\mathbf{Y}_t \rangle_{\mathcal{H}} + Tr\left[ B_t Q D_t^{\dagger} \right] \mathrm{d}t. \tag{69}$$

By plugging $\mathrm{d}\mathbf{X}_t = A_t \mathrm{d}t + B_t \mathrm{d}\mathbf{W}_t^Q$ and $\mathrm{d}\mathbf{Y}_t = C_t \mathrm{d}t + D_t \mathrm{d}\mathbf{W}_t^Q$, we get the desired results. $\qquad \square$

**Lemma B.5** (Verification Theorem). *Let $\mathcal{V}$ be a solution of Hamilton-Jacobi-Bellman (HJB) equation:*

$$\partial_t \mathcal{V}_t + \mathcal{A}\mathcal{V}_t + \inf_{\alpha \in \mathbb{A}} \left[ \langle \alpha, \sigma_t Q^{1/2} D_{\mathbf{x}} \mathcal{V}_t \rangle + \tfrac{1}{2} \|\alpha\|_{\mathcal{H}}^2 \right] = 0, \quad \mathcal{V}(T, \mathbf{x}) = g(\mathbf{x}) \tag{70}$$

*which satisfying the Assumption B.3. Then, we have $\mathcal{V}(0, \mathbf{x}_0) \leq \mathcal{J}(\alpha, \mathbb{P}^{\alpha})$ for every $\alpha \in \mathbb{A}$. Now, let $(\alpha^*, \mathbf{X}^{\star})$ be an admissible pair such that*

$$\alpha_t^*(t, \mathbf{X}_t^{\star}) = \arg\inf_{\alpha \in \mathbb{A}} \left[ \langle \alpha_t, \sigma_t Q^{1/2} D_{\mathbf{x}} \mathcal{V}_t \rangle + \frac{1}{2} \|\alpha_t\|_{\mathcal{H}}^2 \right] = -\sigma_t Q^{1/2} D_{\mathbf{x}} \mathcal{V}(t, \mathbf{X}_t^{\star}) \tag{71}$$

*for almost every $t \in [0, T]$ and $\mathbb{P}$-almost surely. Then $(\alpha^*, \mathbf{X}^{\alpha^*})$ satisfying $\mathcal{V}(0, \mathbf{x}_0) = \mathcal{J}(\alpha^{\star}, \mathbb{P}^{\star})$.*

*Proof.* To start the proof, we formally minimize $F(D_{\mathbf{x}} \mathcal{V})$. Using results from (Da Prato and Zabczyk, 1997), we obtain the candidate minimizer and proceed accordingly:

$$F(\mathbf{x}) = \inf_{\alpha \in \mathbb{A}} \left[ \langle \mathbf{x}, \alpha \rangle + \frac{1}{2} \|\alpha\|_{\mathcal{H}}^2 \right] = -\frac{1}{2} \|\mathbf{x}\|_{\mathcal{H}}^2. \tag{72}$$

Thus, $F(\sigma_t Q^{1/2} D_{\mathbf{x}} \mathcal{V})$ is minimized at at $\alpha^* = -\sigma Q^{1/2} D_{\mathbf{x}} \mathcal{V}$. Next, applying Itô's formula (Da Prato and Zabczyk, 2014, Chapter 4.4) to $\mathcal{V}$ and take the expectation on both sides to obtain

$$\mathbb{E}_{\mathbb{P}^\alpha}\left[\mathcal{V}(T, \mathbf{X}_T^\alpha)\right] = \mathbb{E}_{\mathbb{P}^\alpha}\left[g(\mathbf{X}_T^\alpha)\right] = \tag{73}$$

$$= \mathcal{V}(0, \mathbf{x}_0) + \mathbb{E}_{\mathbb{P}^\alpha}\left[\int_0^T \left(\partial_t \mathcal{V}(t, \mathbf{X}_t^\alpha) + \mathcal{A}\mathcal{V}(t, \mathbf{X}_t^\alpha) + \langle \sigma_t Q^{1/2} D_{\mathbf{x}} \mathcal{V}(t, \mathbf{X}_t^\alpha), \alpha_t \rangle_{\mathcal{H}}\right) dt\right], \tag{74}$$

Using that $\mathcal{V}$ satisfies (71), add $\mathbb{E}_{\mathbb{P}^\alpha}\left[\int_0^T \frac{1}{2}\|\alpha_t\|_{\mathcal{H}}^2 \, dt\right]$ to both sides. Then the LHS of (73) becomes:

$$\mathbb{E}_{\mathbb{P}^\alpha}\left[g(\mathbf{X}_T^\alpha) + \int_0^T \frac{1}{2}\|\alpha_t\|_{\mathcal{H}}^2 \, dt\right] = \mathcal{J}(\alpha, \mathbb{P}^\alpha). \tag{75}$$

Now for the RHS of the equation (73):

$$\mathcal{V}(0, \mathbf{x}_0) + \mathbb{E}_{\mathbb{P}^\alpha}\left[\int_0^T \left(\frac{1}{2}\|\alpha_t\|_{\mathcal{H}}^2 + \partial_t \mathcal{V}(t, \mathbf{X}_t^\alpha) + \mathcal{A}\mathcal{V}(t, \mathbf{X}_t^\alpha) + \langle \sigma_t Q^{1/2} D_{\mathbf{x}} \mathcal{V}(t, \mathbf{X}_t^\alpha), \alpha_t \rangle_{\mathcal{H}}\right) dt\right] \tag{76}$$

$$\overset{(i)}{=} \mathcal{V}(0, \mathbf{x}_0) + \mathbb{E}_{\mathbb{P}^\alpha}\left[\int_0^T \left(\langle \sigma_t Q^{1/2} D_{\mathbf{x}} \mathcal{V}(t, \mathbf{X}_t^\alpha), \alpha_t \rangle_{\mathcal{H}} + \frac{1}{2}\|\alpha_t\|_{\mathcal{H}}^2 - F(\sigma_t Q^{1/2} D_{\mathbf{x}} \mathcal{V}(t, \mathbf{X}_t^\alpha))\right) dt\right]. \tag{77}$$

Here $(i)$ holds because we add and subtract $\mathbb{E}_{\mathbb{P}^\alpha}\left[\int_0^T F(D_{\mathbf{x}} \mathcal{V}(t, \mathbf{X}_t^\alpha))dt\right]$ and use again that $\mathcal{V}$ satisfies (71). Therefore, we obtain the following equation:

$$\mathcal{J}(\alpha, \mathbb{P}^\alpha) = \mathcal{V}(0, \mathbf{x}_0) \tag{78}$$

$$+ \mathbb{E}_{\mathbb{P}^\alpha}\left[\int_0^T \left(\langle \sigma_t Q^{1/2} D_{\mathbf{x}} \mathcal{V}(t, \mathbf{X}_t^\alpha), \alpha_t \rangle_{\mathcal{H}} + \frac{1}{2}\|\alpha_t\|_{\mathcal{H}}^2 - F(\sigma_t Q^{1/2} D_{\mathbf{x}} \mathcal{V}(t, \mathbf{X}_t^\alpha))\right) dt\right]. \tag{79}$$

By definition, $\left[\langle \sigma_t Q^{1/2} D_{\mathbf{x}} \mathcal{V}(t, \mathbf{X}_t^\alpha), \alpha_t \rangle_{\mathcal{H}} + \frac{1}{2}\|\alpha_t\|_{\mathcal{H}}^2\right] - F(D_{\mathbf{x}} \mathcal{V}(t, \mathbf{X}_t^\alpha)) \geq 0$. Thus, taking the infimum over $\alpha \in \mathbb{A}$ on the RHS of (78) yields $\mathcal{J}(\alpha, \mathbb{P}^\alpha) \geq \mathcal{V}(0, \mathbf{x}_0)$. We already verified that $F(\sigma_t Q^{1/2} D_{\mathbf{x}} \mathcal{V})$ attains the infimum at $\alpha_t^\star = -\sigma_t Q^{1/2} D_{\mathbf{x}} \mathcal{V}$, so choose $u = \alpha^\star$. Then,

$$\left[\langle \sigma Q^{1/2} D_{\mathbf{x}} \mathcal{V}(s, \mathbf{X}_s^u), u_s \rangle + \frac{1}{2}\|u_s\|^2\right] - F(\sigma Q^{1/2} D_{\mathbf{x}} \mathcal{V}(s, \mathbf{X}_s^u)) = 0. \tag{80}$$

Thus, we get $\mathcal{J}(u, \mathbb{P}^u) = \mathcal{V}(t, \mathbf{x})$. Combining with (78), we conclude that $(\alpha^\star, \mathbf{X}^{\alpha^\star})$ is optimal for $(t, \mathbf{x}) \in [0, T] \times \mathcal{H}$. This completes the proof. $\qquad\square$

## B.2. Derivation of Divergence Between Path Measures

Here we present an infinite-dimensional generalization of Girsanov theorem (Da Prato and Zabczyk, 2014, Theorem 10.14), which plays a crucial role in estimating the divergence between two path.

**Theorem B.6** (Generalized Girsanov). *Let $\{\zeta_t\}_{0 \leq t \leq T}$ be a $\mathcal{U}$-valued predictable process such that*

$$\mathbb{P}\left(\int_0^T \|\zeta_t\|_{\mathcal{U}}^2 \, dt < \infty\right) = 1, \quad \mathbb{E}\left[\exp\left(\int_0^T \langle \zeta_t, d\mathbf{W}_t^Q \rangle_{\mathcal{U}} - \frac{1}{2}\int_0^T \|\zeta_t\|_{\mathcal{U}}^2 \, dt\right)\right] = 1. \tag{81}$$

*Then the process $\tilde{\mathbf{W}}_t^Q = \mathbf{W}_t^Q - \int_0^t \zeta_s ds$ is a $Q$-Wiener process with respect to the filtration on the probability space $(\Omega, \mathcal{F}, \mathbb{Q})$ where*

$$d\mathbb{Q} = \exp\left(\int_0^T \langle \zeta_t, d\mathbf{W}_t^Q \rangle_{\mathcal{U}} - \frac{1}{2}\int_0^T \|\zeta_t\|_{\mathcal{U}}^2 \, dt\right) d\mathbb{P}. \tag{82}$$

*Or alternatively, by substituting $\mathbf{W}_t^Q = \tilde{\mathbf{W}}_t^Q + \int_0^t \zeta_s ds$ to (87), we obtain*

$$d\mathbb{Q} = \exp\left(\int_0^T \langle \zeta_t, d\tilde{\mathbf{W}}_t^Q \rangle_{\mathcal{U}} + \frac{1}{2}\int_0^T \|\zeta_t\|_{\mathcal{U}}^2 \, dt\right) d\mathbb{P}. \tag{83}$$

*Proof.* Proof can be found in (Da Prato and Zabczyk, 2014, Theorem 10.14) $\qquad\square$

Based on Theorem B.6, we derive the KL-objective in (12). Let $\eta_t = Q^{1/2}\alpha_t \in \mathcal{H}_0 \subseteq \mathcal{H}$ and $\tilde{\mathbf{W}}_t \in \mathcal{U}$ be a $\mathbb{P}^\alpha$-cylindrical Wiener process and set $\mathbf{W}_t := \tilde{\mathbf{W}}_t - \int_0^t \alpha_s \mathrm{d}s$. Then, substitute $\mathbf{W}_t^Q = \tilde{\mathbf{W}}_t^Q - \int_0^t \eta_t \mathrm{d}s$ to a controlled SDEs in $\mathbb{P}^\alpha$:

$$\mathrm{d}\mathbf{X}_t^\alpha = \mathcal{A}\mathbf{X}_t^\alpha \mathrm{d}t + \sigma_t \eta_t \mathrm{d}t + \sigma_t \mathrm{d}\mathbf{W}_t^Q \tag{84}$$

$$= \mathcal{A}\mathbf{X}_t \mathrm{d}t + \sigma_t \eta_t \mathrm{d}t + \sigma_t \left[\mathrm{d}\tilde{\mathbf{W}}_t^Q - \eta_t \mathrm{d}t\right] \tag{85}$$

$$= \mathcal{A}\mathbf{X}_t dt + \sigma_t \mathrm{d}\tilde{\mathbf{W}}_t^Q \tag{86}$$

It implies that $\tilde{\mathbf{W}}_t^Q$ is Q-Wiener process with respect to $\mathbb{P}^\alpha$. Moreover, by Girsanov theorem,

$$\mathrm{d}\mathbb{P} = \exp\left(\int_0^T \langle \alpha_t, \mathrm{d}\tilde{\mathbf{W}}_t \rangle_{\mathcal{U}} - \frac{1}{2}\int_0^T \|\alpha_t\|_{\mathcal{U}}^2 \mathrm{d}t\right) \mathrm{d}\mathbb{P}^\alpha \tag{87}$$

Now, by change of measure in (10), we get the following relation:

$$\mathrm{d}\mathbb{P}^\star = \frac{1}{\tilde{\mathcal{Z}}} e^{-U(\mathbf{X}_T) - \log q_T(\mathbf{X}_T)} \mathrm{d}\mathbb{P} \tag{88}$$

$$= \frac{1}{\tilde{\mathcal{Z}}} e^{-U(\mathbf{X}_T^\alpha) - \log q_T(\mathbf{X}_T^\alpha)} \frac{\mathrm{d}\mathbb{P}}{\mathrm{d}\mathbb{P}^\alpha} \mathrm{d}\mathbb{P}^\alpha, \tag{89}$$

where $\tilde{\mathcal{Z}} = \mathbb{E}\left[\frac{1}{\tilde{\mathcal{Z}}} e^{-U(\mathbf{X}_T) - \log q_T(\mathbf{X}_T)}\right]$ is normalization constant. Combining (87) and (89) and take logarthm on both sides, we get

$$\log \frac{\mathrm{d}\mathbb{P}^\star}{\mathrm{d}\mathbb{P}^\alpha} = \log \frac{1}{\tilde{\mathcal{Z}}} e^{-U(\mathbf{X}_T^\alpha) - \log q_T(\mathbf{X}_T^\alpha)} + \log \frac{\mathrm{d}\mathbb{P}}{\mathrm{d}\mathbb{P}^\alpha} \tag{90}$$

$$= -U(\mathbf{X}_T^\alpha) - \log q_T(\mathbf{X}_T^\alpha) + \int_0^T \langle \alpha_t, \mathrm{d}\tilde{\mathbf{W}}_t \rangle_{\mathcal{U}} - \frac{1}{2}\int_0^T \|\alpha_t\|_{\mathcal{U}}^2 \mathrm{d}t - \log \tilde{\mathcal{Z}}. \tag{91}$$

Since $\tilde{\mathbf{W}}_t$ is $\mathbb{P}^\alpha$-Wiener process, taking expecatation with respect to $\mathbb{P}^\alpha$ on (91) we get

$$\mathbb{E}_{\mathbb{P}^\alpha}\left[\log \frac{\mathrm{d}\mathbb{P}^\alpha}{\mathrm{d}\mathbb{P}^\star}\right] = \mathbb{E}_{\mathbb{P}^\alpha}\left[\frac{1}{2}\int_0^T \|\alpha_t\|_{\mathcal{U}}^2 \mathrm{d}t + (U(\mathbf{X}_T^\alpha) + \log q_T(\mathbf{X}_T^\alpha))\right] + \log \tilde{\mathcal{Z}}. \tag{92}$$

Then, since $\tilde{\mathcal{Z}}$ is constant, we get the desired KL-divergence minimization objective:

$$\min_\alpha D_{\mathrm{KL}}(\mathbb{P}^\alpha | \mathbb{P}^\star) = \min_\alpha \mathbb{E}_{\mathbb{P}^\alpha}\left[\frac{1}{2}\int_0^T \|\alpha_t\|_{\mathcal{U}}^2 \mathrm{d}t + (U(\mathbf{X}_T^\alpha) + \log q_T(\mathbf{X}_T^\alpha))\right]. \tag{93}$$

### B.3. Proof of Lemma 1

**Lemma 1** (Finite Normalization Constant). *Let us assume $-U(\mathbf{x}) \leq \beta \|\mathbf{x}\|_{\mathcal{H}}^2 - C$ with constants $\beta, C > 0$. Then, for any $\mathrm{Tr}(Q) < \infty$, choosing $\nu = \mathcal{N}(0, Q)$ in (6) results in a finite normalization constant $\mathcal{Z} < \infty$.*

*Proof.* The Fernique's integrability theorem (Da Prato and Zabczyk, 2014, Theorem 2.7) states that for any centred Gaussian measure $\nu = \mathcal{N}(0, Q)$ with any $\mathrm{Tr}(Q) < \infty$ on a separable Banach space $\mathcal{B}$ (in particular on $\mathcal{H}$) there exists $\beta > 0$ such that $\int_{\mathcal{H}} e^{\beta \|\mathbf{x}\|_{\mathcal{H}}^2} d\nu(\mathbf{x}) < \infty$. Because the energy satisfies quadratic lower bound $U(\mathbf{x}) \geq -\beta \|\mathbf{x}\|_{\mathcal{H}}^2 + C$, we obtain $e^{-U(\mathbf{x})} \leq \tilde{C} e^{\beta \|\mathbf{x}\|^2}$, where $\tilde{C} = e^{-C}$. Hence, we get:

$$\mathcal{Z} = \int_{\mathcal{H}} e^{-U(\mathbf{x})} d\nu(\mathbf{x}) \leq \tilde{C}\int_{\mathcal{H}} e^{\beta \|\mathbf{x}\|^2} d\nu(\mathbf{x}) < \infty, \tag{94}$$

which confirms that the partition function $\mathcal{Z}$ is finite. $\qquad\square$

## B.4. Proof of Theorem 2

**Theorem 2** (Explicit RND). *Suppose Assumption B.1 holds. Then, for any $0 < t < \infty$ and initial condition $\mathbf{x}_0 \in \mathcal{H}$, $\nu_t^{\mathbf{x}_0} = \mathcal{N}(e^{t\mathcal{A}}\mathbf{x}_0, Q_t)$ and $\nu_\infty = \mathcal{N}(0, Q_\infty)$ are mutually absolutely continuous. Moreover, for any $\mathbf{x} \in \mathcal{H}$, we obtain the explicit RND $q_t(\mathbf{x}_0, \mathbf{x}) := \frac{d\nu_t^{\mathbf{x}_0}}{d\nu_\infty}(\mathbf{x})$ defined as follows:*

$$q_t(\mathbf{x}_0, \mathbf{x}) = det(\Theta_t)^{-\frac{1}{2}} \exp\left[ -\tfrac{1}{2}\langle \Theta_t^{-1}\mathbf{m}_t^{\mathbf{x}_0}, \mathbf{m}_t^{\mathbf{x}_0}\rangle_\infty \right.$$

$$\left. + \langle \Theta_t^{-1}\mathbf{m}_t^{\mathbf{x}_0}, \mathbf{x}\rangle_\infty - \tfrac{1}{2}\langle e^{2t\mathcal{A}}\Theta_t^{-1}\mathbf{x}, \mathbf{x}\rangle_\infty \right], \tag{9}$$

*where we define $\mathbf{m}_t^{\mathbf{x}_0} = e^{t\mathcal{A}}\mathbf{x}_0$, $\Theta_t = 1 - e^{2t\mathcal{A}}$ and $Q_\infty$ norm $\langle u, v\rangle_\infty := \langle Q_\infty^{-\frac{1}{2}}u, Q_\infty^{-\frac{1}{2}}v\rangle_\mathcal{H}$.*

*Proof.* From (41), we get the covariance identity:

$$Q_t = \int_0^t \sigma_{t-r}^2 e^{r\mathcal{A}} Q e^{r\mathcal{A}^\dagger} dr \tag{95}$$

$$= \int_0^t \sigma_\infty^2 e^{r\mathcal{A}} Q e^{r\mathcal{A}^\dagger} dr + \int_0^t (\sigma_{t-r}^2 - \sigma_\infty^2) e^{r\mathcal{A}} Q e^{r\mathcal{A}^\dagger} dr \tag{96}$$

$$= Q_\infty - \int_t^\infty e^{r\mathcal{A}} Q e^{r\mathcal{A}^\dagger} dr + \int_0^t (\sigma_{t-r}^2 - \sigma_\infty^2) e^{r\mathcal{A}} Q e^{r\mathcal{A}^\dagger} dr \tag{97}$$

$$= Q_\infty - e^{t\mathcal{A}} Q_\infty e^{t\mathcal{A}^\dagger} + \int_0^t (\sigma_{t-r}^2 - \sigma_\infty^2) e^{r\mathcal{A}} Q e^{r\mathcal{A}^\dagger} dr. \tag{98}$$

Since, $\int_0^t (\sigma_{t-r}^2 - \sigma_\infty^2) e^{r\mathcal{A}} Q e^{r\mathcal{A}^\dagger} dr \to o(1)$ as we choose $\sigma_t \to \sigma_\infty$, we get the relation:

$$Q_t \approx Q_\infty - e^{t\mathcal{A}} Q_\infty Q^{t\mathcal{A}^\dagger} = Q_\infty^{1/2} \Theta_t Q_\infty^{1/2}. \tag{99}$$

Hence, by applying (Da Prato and Zabczyk, 2002, Proposition 1.3.11), for two centered Gaussians $\mathcal{N}(0, Q_t)$ and $\mathcal{N}(0, Q_\infty)$,

$$q_t(0, \mathbf{x}) = \frac{d\mathcal{N}(0, Q_t)}{d\mathcal{N}(0, Q_\infty)}(\mathbf{x}) = det(\Theta)^{-\frac{1}{2}} \exp\left[ -\tfrac{1}{2}\langle e^{2t\mathcal{A}}\Theta^{-1}\mathbf{x}, \mathbf{x}\rangle_\infty \right]. \tag{100}$$

Now, for the shift from centered case to the general mean function $e^{t\mathcal{A}}\mathbf{x}_0$, we utilize the Cameron-Martin theorem (Da Prato and Zabczyk, 2002, Theorem. 1.3.6). It implies that:

$$\frac{d\mathcal{N}(e^{t\mathcal{A}}\mathbf{x}_0, Q_t)}{d\mathcal{N}(0, Q_t)}(\mathbf{x}) = \exp\left[ \langle Q_t^{-\frac{1}{2}} e^{t\mathcal{A}}\mathbf{x}_0, Q_t^{-\frac{1}{2}}\mathbf{x}\rangle_\mathcal{H} - \tfrac{1}{2}\langle Q_t^{-\frac{1}{2}} e^{t\mathcal{A}}\mathbf{x}_0, Q_t^{-\frac{1}{2}} e^{t\mathcal{A}}\mathbf{x}_0\rangle_\mathcal{H} \right] \tag{101}$$

$$= \exp\left[ \langle Q_\infty^{\frac{1}{2}} Q_t^{-1} e^{t\mathcal{A}}\mathbf{x}_0, Q_\infty^{-\frac{1}{2}}\mathbf{x}\rangle_\mathcal{H} - \tfrac{1}{2}\langle Q_\infty^{\frac{1}{2}} Q_t^{-1} e^{t\mathcal{A}}\mathbf{x}_0, Q_\infty^{-\frac{1}{2}} e^{t\mathcal{A}}\mathbf{x}_0\rangle_\mathcal{H} \right] \tag{102}$$

$$\overset{(i)}{=} \exp\left[ \langle \Theta^{-1} Q_\infty^{-\frac{1}{2}} e^{t\mathcal{A}}\mathbf{x}_0, Q_\infty^{-\frac{1}{2}}\mathbf{x}\rangle_\mathcal{H} - \tfrac{1}{2}\langle \Theta^{-1} Q_\infty^{-\frac{1}{2}} e^{t\mathcal{A}}\mathbf{x}_0, Q_\infty^{-\frac{1}{2}} e^{t\mathcal{A}}\mathbf{x}_0\rangle_\mathcal{H} \right] \tag{103}$$

$$= \exp\left[ \langle \Theta^{-1}\mathbf{m}_t^{\mathbf{x}_0}, \mathbf{x}\rangle_\infty - \tfrac{1}{2}\langle \Theta^{-1}\mathbf{m}_t^{\mathbf{x}_0}, \mathbf{m}_t^{\mathbf{x}_0}\rangle_\infty \right], \tag{104}$$

where (i) follows from (99) such that $\Theta^{-1} Q_\infty^{-1/2} = (Q_t^{-1/2} Q_\infty^{1/2})^\dagger Q_t^{-1/2} = Q_\infty^{1/2} Q_t^{-1}$. Hence, by chain-rule, we get the desired result:

$$q_t(\mathbf{x}_0, \mathbf{x}) := \frac{d\mathcal{N}(e^{t\mathcal{A}}\mathbf{x}_0, Q_t)}{d\mathcal{N}(0, Q_t)} \frac{d\mathcal{N}(0, Q_t)}{d\mathcal{N}(0, Q_\infty)}(\mathbf{x}) \tag{105}$$

$$= det(\Theta)^{-\frac{1}{2}} \exp\left[ -\tfrac{1}{2}\langle \Theta^{-1}\mathbf{m}_t^{\mathbf{x}_0}, \mathbf{m}_t^{\mathbf{x}_0}\rangle_\infty + \langle \Theta^{-1}\mathbf{m}_t^{\mathbf{x}_0}, \mathbf{x}\rangle_\infty - \tfrac{1}{2}\langle e^{2t\mathcal{A}}\Theta^{-1}\mathbf{x}, \mathbf{x}\rangle_\infty \right] \tag{106}$$

It concludes the proof. $\square$

**B.5. Proof of Proposition 4**

**Proposition 4** (Adjoint Matching in $\mathcal{H}$). *Consider the following infinite-dimensional matching objective:*

$$\mathcal{L}(\theta) = \int_0^T \mathbb{E}_{\mathbb{P}^{\bar\alpha}} \left[ \frac{1}{2} \left\| \alpha^\theta(\mathbf{X}_t^{\bar\alpha}, t) + \sigma_t Q^{1/2} \mathbf{Y}_t^{\bar\alpha} \right\|_{\mathcal{H}}^2 \right] dt \tag{18}$$

$$d\mathbf{Y}_t^{\bar\alpha} = -\mathcal{A}^\dagger \mathbf{Y}_t^{\bar\alpha} dt + \mathbf{Z}_t^{\bar\alpha} d\mathbf{W}_t^Q, \ \mathbf{Y}_T^{\bar\alpha} = D_{\mathbf{x}} g(\mathbf{X}_T^{\bar\alpha}). \tag{19}$$

*where $\bar\alpha := \mathtt{stopgrad}(\alpha^\theta)$. Then, the critical point of $\mathcal{L}$ is optimal control $\alpha^\star$ for the SOC problem in (12).*

*Proof.* Let $\mathbf{X}^\theta := \mathbf{X}^{\alpha^\theta}$ be a controlled process with parametrized control policy $\alpha_\theta$ and $\xi_t := \partial_\theta \mathbf{X}_t^\theta$ be the variation process of $\mathbf{X}^\theta$ with respect to $\theta$. Since diffusion coefficient $\sigma_t$ is independent to $\theta$, we get the differential equation for the variation process by differentiate (11) with respect to $\theta$ as follows:

$$d\xi_t = \mathcal{A}\xi_t dt + \sigma_t Q^{1/2} \left( D_{\mathbf{x}}\alpha^\theta(\mathbf{X}_t^\theta, t)\xi_t + \partial_\theta \alpha^\theta(\mathbf{X}_t^\theta, t) \right) dt, \quad \xi_t = 0. \tag{107}$$

Now, let us define the cost functional with $\mathbb{P}^\theta := \mathbb{P}^{\alpha^\theta}$:

$$\mathcal{J}(\alpha^\theta, \mathbb{P}^\theta) = \mathbb{E}_{\mathbb{P}^\theta} \left[ \int_0^T \frac{1}{2} \left\| \alpha^\theta(t, \mathbf{X}_t^\theta) \right\|_{\mathcal{H}}^2 dt + g(\mathbf{X}_T^\theta) \right]. \tag{108}$$

Then, differentiating cost functional (108) with respect to $\theta$ is given by,

$$\frac{d}{d\theta} \mathcal{J}(\alpha^\theta, \mathbb{P}^\theta) = \mathbb{E}_{\mathbb{P}^\theta} \left[ \int_0^T \langle \alpha^\theta(\mathbf{X}_t^\theta, t), D_{\mathbf{x}}\alpha^\theta(\mathbf{X}_t^\theta, t)\xi_t + \partial_\theta \alpha^\theta(\mathbf{X}_t^\theta, t)\rangle_{\mathcal{H}} dt + \langle D_{\mathbf{x}} g(\mathbf{X}_T^\theta), \xi_T \rangle_{\mathcal{H}} \right]. \tag{109}$$

Next, by applying the Lemma B.4 to $\langle \mathbf{Y}_t^\theta, \xi_t \rangle_{\mathcal{H}}$, we obtain:

$$d\langle \mathbf{Y}_t^\theta, \xi_t \rangle_{\mathcal{H}} = \langle d\mathbf{Y}_t^\theta, \xi_t \rangle_{\mathcal{H}} + \langle \mathbf{Y}_t^\theta, d\xi_t \rangle_{\mathcal{H}} \tag{110}$$

$$\stackrel{(i)}{=} \langle -\mathcal{A}^\dagger \mathbf{Y}_t^\theta, \xi_t \rangle_{\mathcal{H}} dt + \langle \mathbf{Z}_t^\theta d\mathbf{W}_t^Q, \xi_t \rangle_{\mathcal{H}} \tag{111}$$

$$+ \langle \mathbf{Y}_t^\theta, \mathcal{A}\xi_t + \sigma_t Q^{1/2} \left( D_{\mathbf{x}}\alpha^\theta(\mathbf{X}_t^\theta, t)\xi_t + \partial_\theta \alpha^\theta(\mathbf{X}_t^\theta, t) \right) \rangle_{\mathcal{H}} \tag{112}$$

$$\stackrel{(ii)}{=} \langle \mathbf{Y}_t^\theta, \sigma_t Q^{1/2} \left( D_{\mathbf{x}}\alpha^\theta(\mathbf{X}_t^\theta, t)\xi_t + \partial_\theta \alpha^\theta(\mathbf{X}_t^\theta, t) \right) \rangle_{\mathcal{H}} dt + \langle \mathbf{Z}_t^\theta d\mathbf{W}_t^Q, \xi_t \rangle_{\mathcal{H}} \tag{113}$$

where $(i)$ follows since the quadratic covariation $[\xi, \mathbf{Y}^\theta]_t$ canceled out because $\xi$ has no martingale term, and $(ii)$ follows from the adjointness of $\mathcal{A}$ and $Q^{1/2}$ such that *i.e.*, $\langle \mathcal{A}^\dagger u, v \rangle_{\mathcal{H}} = \langle u, \mathcal{A}v \rangle_{\mathcal{H}}$. Since $\xi_0 = 0$ and $\mathbf{Y}_T^\theta = D_{\mathbf{x}} g(\mathbf{X}_T^\theta)$, by integrating (110) and takes expectation, we obtain:

$$\mathbb{E}_{\mathbb{P}^\theta} \left[ \langle D_{\mathbf{x}} g(\mathbf{X}_T^\theta), \xi_T \rangle \right] = \mathbb{E}_{\mathbb{P}^\theta} \left[ \int_0^T d\langle \mathbf{Y}_t^\theta, \xi_t \rangle_{\mathcal{H}} \right] \tag{114}$$

$$= \mathbb{E}_{\mathbb{P}^\theta} \left[ \int_0^T \langle \mathbf{Y}_t^\theta, \sigma_t Q^{1/2} \left( D_{\mathbf{x}}\alpha^\theta(\mathbf{X}_t^\theta, t)\xi_t + \partial_\theta \alpha^\theta(\mathbf{X}_t^\theta, t) \right) \rangle_{\mathcal{H}} dt \right]. \tag{115}$$

Finally, substituting (114) into (109), we get:

$$\frac{d}{d\theta} \mathcal{J}(\alpha^\theta, \mathbb{P}^\theta) \stackrel{(i)}{=} \mathbb{E}_{\mathbb{P}^\theta} \left[ \int_0^T \langle \alpha^\theta(\mathbf{X}_t^\theta, t) + \sigma_t Q^{1/2} \mathbf{Y}_t^\theta, D_{\mathbf{x}}\alpha^\theta(\mathbf{X}_t^\theta, t)\xi_t + \partial_\theta \alpha^\theta(\mathbf{X}_t^\theta, t) \rangle_{\mathcal{H}} dt \right], \tag{116}$$

where $(i)$ follows from the adjointness of $Q^{1/2}$ and scalar $\sigma_t$. Since $\bar\alpha = \mathrm{stopgrad}(\alpha^\theta)$ only blocks the gradient, $\bar\alpha(t, \mathbf{x}) =$

$\alpha^\theta(t, \mathbf{x})$ in pointwise manner. Hence, the forward law satisfies $\mathbb{P}^\theta = \mathbb{P}^{\bar\alpha}$. Therefore, from (116), we can define

$$\tfrac{\mathrm{d}}{\mathrm{d}\theta}\mathcal{J}(\alpha^\theta, \mathbb{P}^{\bar\alpha}) \overset{(i)}{=} \mathbb{E}_{\mathbb{P}^{\bar\alpha}}\left[\int_0^T \langle \alpha^\theta(\mathbf{X}_t^{\bar\alpha}, t) + \sigma_t Q^{1/2}\mathbf{Y}_t^{\bar\alpha}, \partial_\theta\alpha^\theta(\mathbf{X}_t^{\bar\alpha}, t)\rangle_{\mathcal{H}} dt\right] \tag{117}$$

$$= \mathbb{E}_{\mathbb{P}^{\bar\alpha}}\left[\int_0^T \langle \alpha^\theta(\mathbf{X}_t^{\bar\alpha}, t), \partial_\theta\alpha^\theta(\mathbf{X}_t^{\bar\alpha}, t)\rangle_{\mathcal{H}} + \langle \sigma_t Q^{1/2}\mathbf{Y}_t^{\bar\alpha}, \partial_\theta\alpha^\theta(\mathbf{X}_t^{\bar\alpha}, t)\rangle_{\mathcal{H}} dt\right] \tag{118}$$

$$= \mathbb{E}_{\mathbb{P}^{\bar\alpha}}\left[\int_0^T \tfrac{1}{2}\partial_\theta\left\|\alpha^\theta(\mathbf{X}_t^{\bar\alpha}, t) + \sigma_t Q^{1/2}\mathbf{Y}_t^{\bar\alpha}\right\|_{\mathcal{H}}^2 dt\right] \tag{119}$$

$$\overset{(ii)}{=} \partial_\theta \int_0^T \mathbb{E}_{\mathbb{P}^{\bar\alpha}}\left[\tfrac{1}{2}\left\|\alpha^\theta(\mathbf{X}_t^{\bar\alpha}, t) + \sigma_t Q^{1/2}\mathbf{Y}_t^{\bar\alpha}\right\|_{\mathcal{H}}^2\right] dt \tag{120}$$

$$= \tfrac{\mathrm{d}}{\mathrm{d}\theta}\mathcal{L}(\theta), \tag{121}$$

where $(i)$ follows by substituting $\mathbb{P}^{\bar\alpha}$ into (116) while $D_{\mathbf{x}}\alpha^\theta\xi_t$ disappears due to stopgrad operation blocks the gradient through $\mathbf{X}_t^{\bar\alpha}$ in (107) (*i.e.*, $\xi_t = 0$) and $(ii)$ follows from dominate convergence. Therefore, from the SMP stationarity:

$$\alpha^\theta(\mathbf{X}_t^\theta, t) + \sigma_t Q^{1/2}\mathbf{Y}_t^\theta = \alpha^\theta(\mathbf{X}_t^{\bar\alpha}, t) + \sigma_t Q^{1/2}\mathbf{Y}_t^{\bar\alpha} = 0, \quad t \in [0, T], \tag{122}$$

we obtain $\frac{\mathrm{d}}{\mathrm{d}\theta}\mathcal{J}(\alpha^\theta, \mathbb{P}^{\bar\alpha}) = \frac{\mathrm{d}}{\mathrm{d}\theta}\mathcal{L}(\theta) = 0$. This implies that the SOC objective in (12) and matching objective in (18) share the same critical points.

Now, let us verify that the critical point $\alpha^\theta(\mathbf{X}_t^{\bar\alpha}, t) = -\sigma_t Q^{1/2}\mathbf{Y}_t^{\bar\alpha}$ in (122) is optimal control $\alpha^\star$ for the SOC problem in (12). To do so, let us assume that there exists $\mathcal{V}$ be a solution of HJB equation (70) satisfying Assumption B.3. Then, by applying Itô formula, we have:

$$d\mathcal{V}(t, \mathbf{X}_t^\theta) = \big[\partial_t\mathcal{V}_t + \langle \mathcal{A}\mathbf{X}_t^\theta, D_{\mathbf{x}}\mathcal{V}_t\rangle_{\mathcal{H}} \tag{123}$$

$$+ \tfrac{1}{2}\mathrm{Tr}(\sigma_t^2 Q D_{\mathbf{xx}}\mathcal{V}_t) + \langle \sigma_t Q^{1/2}\alpha_t^\theta, D_{\mathbf{x}}\mathcal{V}_t\rangle_{\mathcal{H}}\big] dt + \langle D_{\mathbf{x}}\mathcal{V}, \sigma_t d\mathbf{W}_t^Q\rangle_{\mathcal{H}} \tag{124}$$

$$\overset{(i)}{=} \big[\langle \alpha_t^\theta, \sigma_t Q^{1/2}D_{\mathbf{x}}\mathcal{V}_t\rangle_{\mathcal{H}} - F(\sigma_t Q^{1/2}D_{\mathbf{x}}\mathcal{V}_t)\big] dt + \langle D_{\mathbf{x}}\mathcal{V}, \sigma_t d\mathbf{W}_t^Q\rangle_{\mathcal{H}}, \tag{125}$$

where $(i)$ follows from the definition of HJB equation (70) and we define $F(\sigma_t Q^{1/2}D_{\mathbf{x}}\mathcal{V}_t) := \inf_{\alpha\in\mathbb{A}}\big[\langle \alpha_t, \sigma_t Q^{1/2}D_{\mathbf{x}}\mathcal{V}_t\rangle_{\mathcal{H}} + \tfrac{1}{2}\|\alpha_t\|_{\mathcal{H}}^2\big]$. Now, assume $\mathbf{Y}_t^\theta := D_{\mathbf{x}}\mathcal{V}_t$ and $\mathbf{Z}_t^\theta := D_{\mathbf{xx}}\mathcal{V}_t$ and apply the Itô formula, then we get

$$d\mathbf{Y}_t^\theta = \big[\partial_t D_{\mathbf{x}}\mathcal{V}_t + \langle \mathcal{A}\mathbf{X}_t^\theta, D_{\mathbf{xx}}\mathcal{V}_t\rangle_{\mathcal{H}} \tag{126}$$

$$+ \tfrac{1}{2}\mathrm{Tr}(\sigma_t Q D_{\mathbf{xx}}(D_{\mathbf{x}}\mathcal{V}_t)) + \langle \sigma_t Q^{1/2}\alpha_t^\theta, D_{\mathbf{xx}}\mathcal{V}_t\rangle_{\mathcal{H}}\big] + \sigma_t\mathbf{Z}_t^\theta d\mathbf{W}_t^Q. \tag{127}$$

Here, by differentiate the HJB in (70) with respect to $\mathbf{x}$, we get:

$$\partial_t D_{\mathbf{x}}\mathcal{V}_t + \mathcal{A}^\dagger D_{\mathbf{x}}\mathcal{V} + \langle \mathcal{A}\mathbf{X}_t^\theta, D_{\mathbf{xx}}\mathcal{V}_t\rangle_{\mathcal{H}} \tag{128}$$

$$+ \tfrac{1}{2}\mathrm{Tr}(\sigma_t^2 Q D_{\mathbf{xx}}(D_{\mathbf{x}}\mathcal{V}_t)) + D_{\mathbf{xx}}\mathcal{V}_t\sigma_t Q^{1/2}D_{\mathbf{x}}F(\sigma_t Q^{1/2}D_{\mathbf{x}}\mathcal{V}) = 0. \tag{129}$$

Now, by substituting (128) into (126), we get the following:

$$d\mathbf{Y}_t^\theta = \Big[-\mathcal{A}^\dagger\mathbf{Y}_t + \langle \sigma_t Q^{1/2}\alpha_t^\theta - \sigma_t Q^{1/2}D_{\mathbf{x}}F(\sigma_t Q^{1/2}\mathbf{Y}_t^\theta), D_{\mathbf{xx}}\mathcal{V}\rangle\Big] dt + \sigma_t\mathbf{Z}_t^\theta d\mathbf{W}_t^Q \tag{130}$$

$$\overset{(i)}{=} \Big[-\mathcal{A}^\dagger\mathbf{Y}_t^\theta + \langle \sigma_t Q^{1/2}\alpha_t^\theta + \sigma_t^2 Q\mathbf{Y}_t^\theta, D_{\mathbf{xx}}\mathcal{V}\rangle\Big] dt + \sigma_t\mathbf{Z}_t^\theta d\mathbf{W}_t^Q \tag{131}$$

$$\overset{(ii)}{=} -\mathcal{A}^\dagger\mathbf{Y}_t^\theta dt + \sigma_t\mathbf{Z}_t^\theta d\mathbf{W}_t^Q, \tag{132}$$

where $(i)$ follows from $F(\sigma_t Q^{1/2}\mathbf{Y}_t^\theta) = -\tfrac{1}{2}\left\|\sigma_t Q^{1/2}\mathbf{Y}_t^\theta\right\|_{\mathcal{H}}^2$ by definition of $F$ and $(ii)$ follows from $\alpha_t^\theta = -\sigma_t Q^{1/2}\mathbf{Y}_t^\theta$. Since the stopgrad operation does not affect any step in the derivation, therefore, if $\mathbf{Y}_t^\theta = D_{\mathbf{x}}\mathcal{V}(t, \mathbf{X}_t^\theta)$, then $\mathbf{Y}_t^\theta$ is the solution of (19) and the critical point in (122) satisfies

$$\alpha_t^\theta = -\sigma_t Q^{1/2}\mathbf{Y}_t^{\bar\alpha} = -\sigma_t Q^{1/2}D_{\mathbf{x}}\mathcal{V}_t. \tag{133}$$

Hence, by Lemma B.5, this control satisfies the HJB and is optimal, hence $\alpha^\theta = \alpha^\star$. Conversely, if we assume that the critical point $\alpha_t^\theta = -\sigma_t Q^{1/2} \mathbf{Y}_t^{\bar{\alpha}}$ is optimal where $\mathbf{Y}_t^{\bar{\alpha}}$ is the solution of (19), then the same verification immediately yields $\mathbf{Y}_t^{\bar{\alpha}} = D_{\mathbf{x}} \mathcal{V}(t, \mathbf{X}_t^{\bar{\alpha}})$. This shows that the critical point in (122) is the optimal control $\alpha^\star$ for the SOC problem in (12). This completes the proof. $\qquad\square$

**Sufficient condition**   Indeed, in the special case *i.e.*, when $g$ is convex, the converse becomes a sufficient condition. One may ask whether solving the adjoint equation and minimizing the Hamiltonian ensures optimality. In general, SOC sampling needs non convex $g$, so this condition often fails. We therefore do not apply it in our study. Instead, we formulate and prove a sufficient optimality condition tailored to specific SOC problem in (12).

**Proposition B.7** (Stochastic Maximum Principle). *Assume the terminal cost $g$ is convex. Let us fix an admissible control $\bar{\alpha} \in \mathbb{A}$ and consider the corresponding infinite dimensional coupled FBSDEs:*

$$d\mathbf{X}_t^{\bar{\alpha}} = \left[ \mathcal{A} \mathbf{X}_t^{\bar{\alpha}} + \sigma_t Q^{1/2} \bar{\alpha}_t \right] dt + \sigma_t d\mathbf{W}_t^Q, \quad \mathbf{X}_0^{\bar{\alpha}} = \mathbf{x}_0, \tag{134}$$

$$d\mathbf{Y}_t^{\bar{\alpha}} = -\left[ \mathcal{A}^\dagger \mathbf{Y}_t^{\bar{\alpha}} + D_{\mathbf{x}} H(t, \mathbf{X}_t^{\bar{\alpha}}, \bar{\alpha}_t, \mathbf{Y}_t^{\bar{\alpha}}, \mathbf{Z}_t^{\bar{\alpha}}) \right] dt + \mathbf{Z}_t^{\bar{\alpha}} d\mathbf{W}_t^Q, \quad \mathbf{Y}_T^{\bar{\alpha}} = D_{\mathbf{x}} g(\mathbf{X}_T^{\bar{\alpha}}). \tag{135}$$

*Then, under the state variables is fixed $(\mathbf{X}_t^{\bar{\alpha}}, \mathbf{Y}_t^{\bar{\alpha}}, \mathbf{Z}_t^{\bar{\alpha}})$ for a choosen control $\bar{\alpha}$, if following holds:*

$$H(t, \mathbf{X}_t^{\bar{\alpha}}, \alpha_t^\star, \mathbf{Y}_t^{\bar{\alpha}}, \mathbf{Z}_t^{\bar{\alpha}}) = \min_{\alpha \in \mathbb{A}} H(t, \mathbf{X}_t^{\bar{\alpha}}, \alpha, \mathbf{Y}_t^{\bar{\alpha}}, \mathbf{Z}_t^{\bar{\alpha}}), \quad \forall t \in [0, T], \tag{136}$$

*the resulting sequence of controls $\{\alpha_t^\star\}_{t \in [0,T]}$ is the optimal control $\alpha^\star$ for a given SOC problem (12).*

*Proof.* We adapt the proof of the sufficient condition in (Carmona, 2016, Theorem 4.14) into our infinite-dimensional control problem. Fix $\beta \in \mathcal{U}$ be a admissible control, let us write $\mathbf{X}_t^\star := \mathbf{X}_t^{\alpha^\star}$ and define $\Delta \mathbf{X}_t := \mathbf{X}_t^\beta - \mathbf{X}_t^\star$. By convexity of $g$, we get

$$g(\mathbf{X}_T^\beta) - g(\mathbf{X}_T^\star) \geq \langle D_{\mathbf{x}} g(\mathbf{X}_T^\beta), \mathbf{X}_T^\beta - \mathbf{X}_T^\star \rangle_{\mathcal{H}} = \langle \mathbf{Y}_T^\star, \Delta \mathbf{X}_T \rangle_{\mathcal{H}}. \tag{137}$$

Now, apply the Hilbert space product rule B.4 to $\langle \mathbf{Y}_T^\star, \Delta \mathbf{X}_T \rangle_{\mathcal{H}}$. Then, by using the dynamics of $\mathbf{X}^\beta$ and $\mathbf{X}^\star$ and the adjoint equation for $\mathbf{Y}^\star$, we get:

$$\mathbb{E}\left[ \langle \mathbf{Y}_T^\star, \Delta \mathbf{X}_T \rangle_{\mathcal{H}} \right] = \mathbb{E} \int_0^T \langle -\mathcal{A}^\dagger \mathbf{Y}_t^\star - D_{\mathbf{x}} H(t, \mathbf{X}_t^\star, \alpha^\star, \mathbf{Y}_t^\star, \mathbf{Z}_t^\star, \Delta \mathbf{X}_t \rangle_{\mathcal{H}} dt \tag{138}$$

$$+ \mathbb{E} \int_0^T \langle \mathbf{Y}_t^\star, \mathcal{A} \Delta \mathbf{X}_t + \sigma_t Q^{1/2} \beta_t - \sigma_t Q^{1/2} \alpha_t^\star \rangle_{\mathcal{H}} dt \tag{139}$$

$$\overset{(i)}{=} -\mathbb{E} \int_0^T \langle D_{\mathbf{x}} H(t, \mathbf{X}_t^\star, \alpha^\star, \mathbf{Y}_t^\star, \mathbf{Z}_t^\star, \Delta \mathbf{X}_t \rangle_{\mathcal{H}} dt \tag{140}$$

$$+ \mathbb{E} \int_0^T \langle \mathbf{Y}_t^\star, \sigma_t Q^{1/2} \beta_t - \sigma_t Q^{1/2} \alpha_t^\star \rangle_{\mathcal{H}} dt, \tag{141}$$

where (i) follows from the adjointness of $\mathcal{A}$. On the other hand, using the definition of $\mathcal{J}$ in (108):

$$\mathcal{J}(\beta) - \mathcal{J}(\alpha^\star) = \mathbb{E}\left[ \int_0^T \left( \tfrac{1}{2} \|\beta_t\|_{\mathcal{H}}^2 - \tfrac{1}{2} \|\alpha_t^\star\|_{\mathcal{H}}^2 \right) dt + g(\mathbf{X}_T^\beta) - g(\mathbf{X}_T^\star) \right] \tag{142}$$

$$\geq \mathbb{E}\left[ \int_0^T \left( \tfrac{1}{2} \|\beta_t\|_{\mathcal{H}}^2 - \tfrac{1}{2} \|\alpha_t^\star\|_{\mathcal{H}}^2 \right) + \langle \mathbf{Y}_t^\star, \sigma_t Q^{1/2} \beta_t - \sigma_t Q^{1/2} \alpha_t^\star \rangle_{\mathcal{H}} dt \right] \tag{143}$$

$$- \mathbb{E} \int_0^T \langle D_{\mathbf{x}} H(t, \mathbf{X}_t^\star, \alpha^\star, \mathbf{Y}_t^\star, \mathbf{Z}_t^\star, \Delta \mathbf{X}_t \rangle_{\mathcal{H}} dt \tag{144}$$

$$\overset{(i)}{=} \mathbb{E} \int_0^T \left[ H(t, \mathbf{X}_t^\beta, \beta_t, \mathbf{Y}_t^\star, \mathbf{Z}_t^\star) - H(t, \mathbf{X}_t^\star, \alpha^\star, \mathbf{Y}_t^\star, \mathbf{Z}_t^\star) - \langle D_{\mathbf{x}} H(t, \mathbf{X}_t^\star, \alpha^\star, \mathbf{Y}_t^\star, \mathbf{Z}_t^\star, \Delta \mathbf{X}_t \rangle_{\mathcal{H}} \right] dt, \tag{145}$$

where $(i)$ follows by the definition of Hamiltonian (16). By convexity of $\mathcal{H}$ with respect to $(\mathbf{x}, \alpha)$,

$$H(t, \mathbf{X}_t^\beta, \beta_t, \mathbf{Y}_t^\star, \mathbf{Z}_t^\star) \geq H(t, \mathbf{X}_t^\star, \alpha^\star, \mathbf{Y}_t^\star, \mathbf{Z}_t^\star) \tag{146}$$
$$+ \langle D_\mathbf{x} H(t, \mathbf{X}_t^\star, \alpha^\star, \mathbf{Y}_t^\star, \mathbf{Z}_t^\star, \Delta\mathbf{X}_t\rangle_\mathcal{H} + \langle D_\alpha H(t, \mathbf{X}_t^\star, \alpha^\star, \mathbf{Y}_t^\star, \mathbf{Z}_t^\star, \beta_t - \alpha_t^\star\rangle_\mathcal{H}. \tag{147}$$

Then, the point-wise minimality of $\alpha^\star$ implies that $\langle D_\alpha H(t, \mathbf{X}_t^\star, \alpha^\star, \mathbf{Y}_t^\star, \mathbf{Z}_t^\star, \beta_t - \alpha_t^\star\rangle_\mathcal{H} \geq 0$ for any admissible control $\beta$. Therefore, it implies that $\mathcal{J}(\beta) \geq \mathcal{J}(\alpha^\star)$ because of (136). Finally, since we choose $\beta$ arbitrary, resulting $\alpha^\star$ is optimal control for all $t \in [0, T]$. This concludes the proof. $\qquad\square$

## B.6. Proof of Proposition 5

**Proposition 5** (Unbiased Estimator). *Define the sample-wise adjoint matching objective with conditional expectation $\tilde{\mathbf{Y}}_t^{\bar\alpha} := \mathbb{E}_{\mathbb{P}^{\bar\alpha}}[\mathbf{Y}_t^{\bar\alpha}|\mathbf{X}_t^{\bar\alpha}]$ for any sample trajectory $\mathbf{X}_t^{\bar\alpha} \sim \mathbb{P}^{\bar\alpha}$:*

$$\tilde{\mathcal{L}}(\theta) := \int_0^T \tfrac{1}{2}||\alpha^\theta(\mathbf{X}_t^{\bar\alpha}, t) + \sigma_t Q^{1/2}\tilde{\mathbf{Y}}_t^{\bar\alpha}||_\mathcal{H}^2 dt \tag{20}$$

*Then, we get unbiased gradient estimator $\frac{\mathrm{d}}{\mathrm{d}\theta}\mathbb{E}_{\mathbb{P}^{\bar\alpha}}[\tilde{\mathcal{L}}(\theta)] = \frac{\mathrm{d}}{\mathrm{d}\theta}\mathcal{L}(\theta)$ with same critical point in (18).*

*Proof.* Under the stopgrad operation, the forward law $\mathbb{P}^{\bar\alpha}$ does not depend on $\theta$. By the dominated convergence theorem we pass the derivative through the expectation and the integral:

$$\frac{\mathrm{d}}{\mathrm{d}\theta}\mathbb{E}_{\mathbb{P}^{\bar\alpha}}\left[\tilde{\mathcal{L}}(\theta)\right] = \mathbb{E}_{\mathbb{P}^{\bar\alpha}}\left[\frac{\mathrm{d}}{\mathrm{d}\theta}\int_0^T \tfrac{1}{2}||\alpha^\theta(\mathbf{X}_t^{\bar\alpha}, t) + \sigma_t Q^{1/2}\tilde{\mathbf{Y}}_t^{\bar\alpha}||_\mathcal{H}^2 dt\right] \tag{148}$$

$$= \mathbb{E}_{\mathbb{P}^{\bar\alpha}}\left[\int_0^T \langle\alpha^\theta(\mathbf{X}_t^{\bar\alpha}, t) + \sigma_t Q^{1/2}\tilde{\mathbf{Y}}_t^{\bar\alpha}, \partial_\theta\alpha^\theta(\mathbf{X}_t^{\bar\alpha}, t)\rangle_\mathcal{H} dt\right] \tag{149}$$

$$= \int_0^T \left[\mathbb{E}_{\mathbb{P}^{\bar\alpha}}\left[\langle\alpha^\theta(\mathbf{X}_t^{\bar\alpha}, t), \partial_\theta\alpha^\theta(\mathbf{X}_t^{\bar\alpha}, t)\rangle_\mathcal{H}\right] + \mathbb{E}_{\mathbb{P}^{\bar\alpha}}\left[\langle\sigma_t Q^{1/2}\tilde{\mathbf{Y}}_t^{\bar\alpha}, \partial_\theta\alpha^\theta(\mathbf{X}_t^{\bar\alpha}, t)\rangle_\mathcal{H}\right]\right] dt \tag{150}$$

$$= \int_0^T \left[\mathbb{E}_{\mathbb{P}^{\bar\alpha}}\left[\langle\alpha^\theta(\mathbf{X}_t^{\bar\alpha}, t), \partial_\theta\alpha^\theta(\mathbf{X}_t^{\bar\alpha}, t)\rangle_\mathcal{H}\right] + \mathbb{E}_{\mathbb{P}^{\bar\alpha}}\left[\langle\sigma_t Q^{1/2}\mathbb{E}_{\mathbb{P}^{\bar\alpha}}\left[\mathbf{Y}_t^{\bar\alpha}|\mathbf{X}_t^{\bar\alpha}\right], \partial_\theta\alpha^\theta(\mathbf{X}_t^{\bar\alpha}, t)\rangle_\mathcal{H}\right]\right] dt \tag{151}$$

$$\overset{(i)}{=} \int_0^T \left[\mathbb{E}_{\mathbb{P}^{\bar\alpha}}\left[\langle\alpha^\theta(\mathbf{X}_t^{\bar\alpha}, t), \partial_\theta\alpha^\theta(\mathbf{X}_t^{\bar\alpha}, t)\rangle_\mathcal{H}\right] + \mathbb{E}_{\mathbb{P}^{\bar\alpha}}\left[\langle\sigma_t Q^{1/2}\mathbf{Y}_t^{\bar\alpha}, \partial_\theta\alpha^\theta(\mathbf{X}_t^{\bar\alpha}, t)\rangle_\mathcal{H}\right]\right] dt \tag{152}$$

$$= \mathbb{E}_{\mathbb{P}^{\bar\alpha}}\left[\int_0^T \langle\alpha^\theta(\mathbf{X}_t^{\bar\alpha}, t) + \sigma_t Q^{1/2}\mathbf{Y}_t^{\bar\alpha}, \partial_\theta\alpha^\theta(\mathbf{X}_t^{\bar\alpha}, t)\rangle_\mathcal{H} dt\right] \tag{153}$$

$$\overset{(ii)}{=} \frac{\mathrm{d}}{\mathrm{d}\theta}\mathcal{L}(\theta), \tag{154}$$

$$\tag{155}$$

where $(i)$ follows from the tower property and $(ii)$ follows from (117). Hence, we get $\frac{\mathrm{d}}{\mathrm{d}\theta}\mathbb{E}_{\mathbb{P}^{\bar\alpha}}\left[\tilde{\mathcal{L}}(\theta)\right] = \frac{\mathrm{d}}{\mathrm{d}\theta}\mathcal{L}(\theta) = 0$ at $\alpha^\theta(\mathbf{X}_t^{\bar\alpha}, t) = -\sigma_t Q^{1/2}\mathbf{Y}_t^{\bar\alpha}$. It shows that the critical point remains unchanged to $\frac{\mathrm{d}}{\mathrm{d}\theta}\mathcal{L}(\theta)$. It concludes the proof. $\qquad\square$

## B.7. Proof of Proposition 6

**Proposition 6** (Laplacian Operator). *For the negative Laplacian $\mathcal{A} = -\Delta$ on a bounded domain with Dirichlet boundary condition, take $Q = \mathcal{A}^{-s}$ with $s > \frac{d}{2}$. Then, $\mathcal{A}$ and $Q$ satisfy the conditions in Assumption B.1.*

*Proof.* **(A) Trace-class** Let $\{\phi^{(k)}\}_{k\geq 1}$ be an eigen-basis with $\mathcal{A}\phi^{(k)} = -\lambda^{(k)}\phi^{(k)}$. Then, we have

$$Q\mathbf{x} = \sum_{k\geq 1}(\lambda^{(k)})^{-s}\mathbf{x}^{(k)}\phi^{(k)}. \tag{156}$$

Hence the eigenvalues of $Q$ are $(\lambda^{(k)})^{-s}$ and it implies that $Tr(Q) = \sum_{k\geq 1}(\lambda^{(k)})^{-s}$. By the Weyl law $\lambda^{(k)} \asymp k^{\frac{2}{d}}$ on a bounded $C^\infty$ domain in $\mathbb{R}^d$, therefore $Tr(Q) < \infty$ if $s > \frac{d}{2}$.

**(B) Self-adjoint** Since Laplacian $\Delta$ is self-adjoint, we get adjointness of $\mathcal{A}$ directly, and for $Q$:

$$Q^\dagger = ((-\mathcal{A})^{-s})^\dagger = ((-\mathcal{A})^{-s}) = Q \tag{157}$$

**(C) Dissipative** Since $0 < \lambda_1 < \lambda_2, \cdots < \lambda_k < \cdots$ for $k \in \mathbb{N}$, we obtain:

$$\langle \mathcal{A}\mathbf{x}, \mathbf{x} \rangle_\mathcal{H} = -\sum_{k \geq 1} \lambda^{(k)} \left\| \langle \mathbf{x}, \phi^{(k)} \rangle_\mathcal{H} \right\|^2 \leq -\lambda^{(1)} \sum_{k \geq 1} \left\| \langle \mathbf{x}, \phi^{(k)} \rangle_\mathcal{H} \right\|^2 = -\lambda^{(1)} \|\mathbf{x}\|_\mathcal{H}^2. \tag{158}$$

**(D) Commute** For any $\mathbf{x} \in \mathcal{H}$, we get:

$$\mathcal{A}Q\mathbf{x} = \mathcal{A} \left( \sum_{k \geq 1} (\lambda^{(k)})^{-s} \mathbf{x}^{(k)} \phi^{(k)} \right) = \sum_{k \geq 1} (\lambda^{(k)})^{-s} \mathbf{x}^{(k)} (\mathcal{A}\phi^{(k)}) \tag{159}$$

$$= \sum_{k \geq 1} (\lambda^{(k)})^{-s} \mathbf{x}^{(k)} (-\lambda^{(k)} \phi^{(k)}) = -\sum_{k \geq 1} (\lambda^{(k)})^{1-s} \mathbf{x}^{(k)} \phi^{(k)} \tag{160}$$

and

$$Q\mathcal{A}\mathbf{x} = -Q \left( \sum_{k \geq 1} \lambda^{(k)} \mathbf{x}^{(k)} \phi^{(k)} \right) = -\sum_{k \geq 1} \lambda^{(k)} \mathbf{x}^{(k)} (Q\phi^{(k)}) \tag{161}$$

$$= \sum_{k \geq 1} \lambda^{(k)} \mathbf{x}^{(k)} ((\lambda^{(k)})^{-s} \phi^{(k)}) = -\sum_{k \geq 1} (\lambda^{(k)})^{1-s} \mathbf{x}^{(k)} \phi^{(k)}. \tag{162}$$

Therefore, we get $\mathcal{A}Q^{1/2}\mathbf{x} = Q^{1/2}\mathcal{A}\mathbf{x}$. It completes the proof. $\qquad\square$

### B.8. Endpoint Disintegration

From (10), let us denote $\mathbf{X} := \{\mathbf{X}_t\}_{t \in [0,T]} \in \Omega$ and define the IS as follows:

$$w(\mathbf{X}) = e^{-U(\mathbf{X}_T) - \log q_T(\mathbf{x}_0, \mathbf{X}_T)}. \tag{163}$$

Then, the RND for $\mathbb{P}^\star$ with respect to reference measure $\mathbb{P}$ is given by:

$$\frac{d\mathbb{P}^\star}{d\mathbb{P}}(\mathbf{X}) = \frac{w(\mathbf{X})}{\mathbb{E}_\mathbb{P}[w(\mathbf{X})]}. \tag{164}$$

Then, for any bounded functional $F : \Omega \to \mathbb{R}$, we have the expectation:

$$\mathbb{E}_{\mathbb{P}^\star}[F(\mathbf{X}^\star)] = \frac{\mathbb{E}_\mathbb{P}[F(\mathbf{X})\, w(\mathbf{X})]}{\mathbb{E}_\mathbb{P}[w(\mathbf{X})]}. \tag{165}$$

Since terminal law of $\mathbb{P}$ admits the density $q_T$ with respect to $\nu_\infty$ based on Theorem 2, we can disintegrate $\mathbb{P}$ with respect to $\mathbf{X}_T$ to obtain:

$$\mathbb{E}_\mathbb{P}[F(\mathbf{X})w(\mathbf{x})] = \int_\mathcal{H} \left( \int_\mathcal{H} F(\mathbf{X})\mathbb{P}_{\cdot|T}(d\mathbf{X}|\mathbf{X}_T = \mathbf{y}) \right) e^{-U(\mathbf{y}) - \log q_T(\mathbf{x}_0, \mathbf{x}_T)} q_T(\mathbf{x}_0, \mathbf{y}) d\nu_\infty(\mathbf{y}) \tag{166}$$

$$= \int_\mathcal{H} \left( \int_\mathcal{H} F(\mathbf{X})\mathbb{P}_{\cdot|T}(d\mathbf{X}|\mathbf{X}_T = \mathbf{y}) \right) e^{-U(\mathbf{y})} d\nu_\infty(\mathbf{y}) \tag{167}$$

$$\stackrel{(i)}{=} \int_\mathcal{H} \left( \int_\mathcal{H} F(\mathbf{X})\mathbb{P}_{\cdot|T}(d\mathbf{X}|\mathbf{X}_T = \mathbf{y}) \right) \mathbb{E}_\mathbb{P}[w(\mathbf{X})]\, d\pi(\mathbf{y}), \tag{168}$$

where $(i)$ follows from the definition of target distribution $\pi$ in (6). Hence, we obtain that

$$\mathbb{E}_{\mathbb{P}^\star}[F(\mathbf{X}^\star)] = \int \left( \int F(\mathbf{X})\mathbb{P}_{\cdot|T}(d\mathbf{X}|\mathbf{X}_T = \mathbf{y}) \right) \pi(d\mathbf{y}) \tag{169}$$

$$= \mathbb{E}_{\mathbb{P}_{\cdot|T}\mathbb{P}_T^\star}[F(\mathbf{X})]. \tag{170}$$

## C. Experimental Details

### C.1. Bridge sampling

Because the solution of the reference process in (3) defines an Ornstein-Uhlenbeck semigroup, the conditional law at time $t$ given time $T$ is:

$$\mathbb{P}_{t|T}(\mathbf{X}_t|\mathbf{x}_T) = \mathcal{N}(\mathbf{m}_{t|T}, Q_{t|T}), \tag{171}$$

where the conditional mean function and covariance operator is given by:

$$\mathbf{m}_{t|T} = \mathbf{x}_0 + Q_{t|T}Q_T^{-1}(\mathbf{x}_T - \mathbf{x}_0), \quad Q_{t|T} = Q_T - Q_{t|T}Q_T^{-1}Q_{T|t} \tag{172}$$

$$Q_t = \int_0^t e^{(t-s)\mathcal{A}}\sigma_s^2 Q e^{(t-s)\mathcal{A}^\dagger}\mathrm{d}s, \tag{173}$$

$$Q_{t|T} = \int_0^t e^{(t-s)\mathcal{A}}\sigma_s^2 Q e^{(T-s)\mathcal{A}^\dagger}\mathrm{d}s = Q_t e^{(T-t)\mathcal{A}^\dagger}, \quad Q_{T|t} = (Q_{t|T})^\dagger. \tag{174}$$

Moreover, the coordinate process $\mathbf{X}_t^{(k)} = \langle\mathbf{X}_t, \phi^{(k)}\rangle$ of the Ornstein-Uhlenbeck semigroup in (3) is also Gaussian and the coordinates are independent. Therefore we can compute coordinate-wise:

$$\mathbb{P}_{t|T}(\mathbf{X}_t^{(k)}|\mathbf{x}_T^{(k)}) = \mathcal{N}(\mathbf{m}_{t|T}^{(k)}, q_{t|T}^{(k)}) \tag{175}$$

where we denote $q_t^{(k)} = \langle Q_t, \phi^{(k)}\rangle$ for any $t \in [0, T]$ and

$$\mathbf{m}_{t|T}^{(k)} = \mathbf{x}_0^{(k)} + \frac{q_t^{(k)}}{q_\infty^{(k)}}e^{-(T-t)\lambda^{(k)}}(\mathbf{x}_T^{(k)} - \mathbf{x}_0^{(k)}), \tag{176}$$

$$q_{t|T}^{(k)} = q_t^{(k)} - \frac{(q_t^{(k)})^2}{q_\infty^{(k)}}e^{-2(T-t)\lambda^{(k)}} \tag{177}$$

$$q_t^{(k)} = \int_0^t \sigma_s^2 e^{-2(t-s)\lambda^{(k)}}(\lambda^{(k)})^{-s}\mathrm{d}s, \quad \forall t \in [0, T]. \tag{178}$$

### C.2. Enforcing boundary condition

To properly define a transition path between two metastable states, say $\mathbf{A}$ and $\mathbf{B}$, we must fix these states as the boundary conditions for a desired path. Our method achieves this by decomposing the path $\mathbf{X}_t$ into two parts: a fixed reference path $\mathbf{x}_0$ and a fluctuating residual path $\mathbf{R}_t$. Then, the total path is given by $\mathbf{X}_t = \mathbf{x}_0 + \mathbf{R}_t$, where $\mathbf{x}_0$ lifting the residual path (boundary values are zero) to the path that have desired boundary values. The reference path $\mathbf{x}_0$ provides the direct connection from $\mathbf{A}$ to $\mathbf{B}$, while the residual path $\mathbf{R}_t$ is forced to be zero at the boundaries. This construction guarantees that the full path $\mathbf{X}_t$ always begins at $\mathbf{A}$ and ends at $\mathbf{B}$.

For a residual path $\mathbf{R}_t$, we enforce the sample-path endpoints to be a absorbed states *i.e.*, $\mathbf{R}_t[0] = 0, \mathbf{R}_t[L] = 0$ for a function $\mathbf{R}_t : \mathbb{R}_{>0} \to \mathbb{R}^d$ by choosing Dirichlet eigen-basis:

$$\phi^{(k)}[u] = (\tfrac{2}{L})^{1/2}\sin\left(\tfrac{\pi k u}{L}\right), \ \forall k \in \mathbb{N}, \ u \in \mathbf{L} := [0, L] \tag{179}$$

$$\mathbf{R}_t = \sum_{k \geq 1}\mathbf{R}_t^{(k)}\phi^{(k)}, \ \text{where } \mathbf{R}_t^{(k)} := \langle\mathbf{R}_t, \phi^{(k)}\rangle_{\mathcal{H}} = \int_{\mathbf{L}}\mathbf{R}_t[u]\phi^{(k)}[u]\mathrm{d}u. \tag{180}$$

Therefore, we get $\mathbf{R}_t[0] = \mathbf{R}_t[L] = 0$ for all $t \in [0, T]$ since $\phi^{(k)}[0] = \phi^{(k)}[L] = 0$ from (180). For a reference path $\mathbf{x}_0$, we define the as a continuous function $\mathbf{x}_0 : \mathbf{L} \to \mathbb{R}^d$:

$$\mathbf{x}_0[u] = c_0[u]\mathbf{A} + c_1[u]\mathbf{B}, \ \text{with} \begin{cases} c_0[0] = 1, c_0[L] = 0 \\ c_1[0] = 0, c_1[L] = 1 \end{cases} \tag{181}$$

Here, the residual process $\mathbf{R}_t$ on the Dirichlet space is given by:

$$\mathrm{d}\mathbf{R}_t = \left[\mathcal{A}\mathbf{R}_t + \sigma_t Q^{1/2}\alpha(t, \mathbf{x}_0 + \mathbf{R}_t)\right]\mathrm{d}t + \sigma_t\mathrm{d}\mathbf{W}_t^Q, \quad \mathbf{R}_0 = 0. \tag{182}$$

It results that since $\mathbf{X}_t = \mathbf{x}_0 + \mathbf{R}_t$, we can deduce the following $\mathcal{H}$-valued SDE:

$$\mathrm{d}\mathbf{X}_t = \left[\mathcal{A}(\mathbf{X}_t - \mathbf{x}_0) + \sigma_t Q^{1/2}\alpha(t, \mathbf{X}_t)\right]\mathrm{d}t + \sigma_t\mathrm{d}\mathbf{W}_t^Q, \quad \mathbf{X}_0 = \mathbf{x}_0. \tag{183}$$

## C.3. DST-based Galerkin SDE simulation

Given Dirichlet eigen-basis in (179), we can express the infinite-dimensional SDEs into infinite system of real-valued SDEs for each coordinate $k$:

$$d\mathbf{R}_t^{(k)} = \left[-\lambda^{(k)}\mathbf{R}_t^{(k)} + \sigma_t\langle\alpha(t, \mathbf{x}_0 + \mathbf{R}_t), (Q^{1/2})^\dagger\phi^{(k)}\rangle\right]dt + \sigma_t(\lambda^{(k)})^{-\frac{s}{2}}d\beta_t^{(k)}, \quad \mathbf{R}_t^{(k)} = 0, \tag{184}$$

$$= \left[-\lambda^{(k)}\mathbf{R}_t^{(k)} + \sigma_t(\lambda^{(k)})^{-\frac{s}{2}}\langle\alpha(t, \mathbf{X}_t), \phi^{(k)}\rangle\right]dt + \sigma_t(\lambda^{(k)})^{-\frac{s}{2}}d\beta_t^{(k)} \tag{185}$$

where $\{\beta_t^{(k)}\}_{k\geq 1}$ are standard Wiener processes. Note that, to compute $(Q^{1/2})^\dagger\phi^{(k)}$, we choose $\mathcal{U} = \mathcal{H}$, means $\mathbf{W}_t = \sum_{k\geq 1}\phi_k\beta_t^{(k)}$. Even though $\mathbf{W}_t$ is not an $\mathcal{H}$-valued random variable, it is a cylindrical process indexed by $\mathcal{H}$, applying Hilbert-Schmidt $Q^{1/2}$ before stochastic integral yields the resulting process remain square integrable.

If we treat the finite path evaluations as samples on a uniform grid $\mathbf{L} = [0, L]$, we project onto the sine basis with the discrete sine transform (DST). Since a finite-resolution path has a natural cutoff frequency $K$ as a number of evaluation points, projection to and from the sine basis is exact. Concretely, approximate the Laplacian by $\Delta \triangleq \mathbf{E}\mathbf{D}\mathbf{E}^\top$, where $\mathbf{E}^\top$ is the DST projection consists of eigen functions $\{\phi^{(k)}\}_{k=1}^K$ so that $\tilde{\mathbf{R}}_t = \mathbf{E}^\top\mathbf{R}_t$ and $\mathbf{D}$ is a diagonal matrix containing the eigen values $\{\lambda^{(k)}\}_{k=1}^K$. Hence, stochastic evolution in (182) is described by the finite-dimensional model (configuration to spectral space):

$$d\tilde{\mathbf{R}}_t = \left[-\mathbf{D}\tilde{\mathbf{R}}_t + \sigma_t\mathbf{D}^{-\frac{s}{2}}\mathbf{E}^\top\alpha(t, \mathbf{X}_t)\right]dt + \sigma_t\mathbf{D}^{-\frac{s}{2}}d\tilde{\mathbf{W}}_t. \tag{186}$$

The simulation algorithm for FAS is provided in detail in Algorithm 3. Additionally, we introduce a spectral precision scale $\kappa$ on the Dirichlet Laplacian, replacing $\mathbf{D}$ by $\kappa^2\mathbf{D}$ to control stiffness and smoothness of the reference dynamics.

## C.4. overNumerical Computation

**Correction RND** From Proposition 6, we can calculate the RND in Theorem 2 as follows:

$$\log q_t(\mathbf{x}_0, \mathbf{x}) = -\frac{1}{2}\sum_{k\geq 1}\frac{1}{q_\infty^{(k)}(e^{2t\lambda_k} - 1)}\left[\left\|\mathbf{x}_0^{(k)}\right\|^2 - 2e^{-t\lambda_k}\langle\mathbf{x}_0^{(k)}, \mathbf{x}^{(k)}\rangle + \left\|\mathbf{x}^{(k)}\right\|^2\right], \tag{187}$$

where $q_\infty^{(k)}$ is diagonal component of $Q_\infty$ such that $Q_\infty\phi^{(k)} = q_\infty^{(k)}\phi^{(k)}$. From (45), we get $Q_\infty = -\frac{1}{2}\sigma_\infty^2\mathcal{A}^{-1}Q$, therefore we get $q_\infty^{(k)} = \frac{1}{2}\sigma_\infty^2(\lambda^{(k)})^{(1+s)}$.

**Noise schedule $\sigma_t$** The closed-form simulation may be infeasible because computing $q_t^{(k)}$ in (178) requires numerical integration for arbitrary noise schedule $\sigma_t$. In this case, the two noise schedules studied for adjoint sampling in (Havens et al., 2025; Liu et al., 2026) can be adapted to our setting.

- The constant noise schedule sets $\sigma(t) := \sigma$. Then, we can can compute the integral in (178):

$$q_t^{(k)} = (\lambda^{(k)})^{-s}\frac{(1 - e^{-2t\lambda_k})}{2\lambda_k}. \tag{188}$$

- The geometric noise schedule (Song et al., 2021; Karras et al., 2022) decreases monotonically by predefined $\beta_{\min}$ and $\beta_{\max}$:

$$\sigma_t = \begin{cases} \beta_{\min}\left(\frac{\beta_{\max}}{\beta_{\min}}\right)^{T-t}\sqrt{2\log\frac{\beta_{\max}}{\beta_{\min}}}, & t < T \\ \beta_{\min}\sqrt{2\log\frac{\beta_{\max}}{\beta_{\min}}}, & t \geq T. \end{cases} \tag{189}$$

Because choosing the noise schedule $\sigma_t = \beta e^{\alpha t}$ gives a closed-form for the integral in (178), we rearrange it as follows:

$$\sigma_t = \beta_{\min}\left(\frac{\beta_{\max}}{\beta_{\min}}\right)^{T-t}\sqrt{2\log\frac{\beta_{\max}}{\beta_{\min}}} = \beta_{\min}\left(\frac{\beta_{\max}}{\beta_{\min}}\right)^T\sqrt{2\log\frac{\beta_{\max}}{\beta_{\min}}}e^{-\log\left(\frac{\beta_{\max}}{\beta_{\min}}\right)t}. \tag{190}$$

*Table 4.* Training Hyper-parameters

|  | Synthetic potential | Alanine dipeptide | Chignolin |
|---|---|---|---|
| $\beta_{\min}$ | 0.1 | 0.01 | 0.01 |
| $\beta_{\max}$ | 10 | 15 | 5 |
| $\kappa$ | $10^{-2}$ | $10^{-5}$ | $10^{-5}$ |
| $M$ | $10^3$ | $10^4$ | $10^4$ |
| $L$ | 100 | 100 | 100 |
| $N$ | 512 | 16 | 16 |
| $|\mathcal{B}|$ | $10^4$ | $10^3$ | $10^3$ |
| $\alpha_{\max}$ | 100 | $10^4$ | $10^4$ |
| $|\mathbf{U}|$ | 100 | 100 | 100 |
| Learning rate | $10^{-4}$ | $10^{-5}$ | $10^{-4}$ |
| Base channel | 32 | 64 | 128 |
| Base mode | 8 | 8 | 8 |
| Embed dim | 128 | 256 | 512 |

Therefore, we get $\beta = \beta_{\min}(\frac{\beta_{\max}}{\beta_{\min}})^T \sqrt{2 \log \frac{\beta_{\max}}{\beta_{\min}}}$, $\alpha = -\log\left(\frac{\beta_{\max}}{\beta_{\min}}\right)$. This choice yields that:

$$q_t^{(k)} = \beta^2 (\lambda^{(k)})^{-s} \frac{e^{2\alpha t} - e^{-2\lambda_k t}}{2(\alpha + \lambda_k)} \qquad (191)$$

**Replay buffer** $\mathcal{B}$    Following previous diffusion samplers (Havens et al., 2025; Liu et al., 2026; Choi et al., 2026), we maintain a replay buffer $\mathcal{B}$ for the computation of the objective in (22). In this case, the expectation under $\mathbb{P}^\alpha$ is approximated by an average over $\mathcal{B}$, which holds the most recent $|\mathcal{B}|$ samples. We refresh the buffer $\mathcal{B}$ by inserting $N$ new samples after every $L$ gradient steps.

**Parametrization fo control network**    We parameterize the control $\alpha^\theta$ with a neural operator to preserve its functional form. We parametrize the control as $\alpha^\theta(t, \mathbf{x}) := \sigma_t Q^{1/2} e^{-(T-t)\mathcal{A}^\dagger} u^\theta(t, \mathbf{x})$, which absorbs the noise schedule $\sigma_t$ and Laplacian contraction $Q^{1/2} e^{-(T-t)\mathcal{A}^\dagger}$ into the control and removes its explicit appearance from the matching objective in (22), yielding more stable training. For all experiment, we take $u^\theta$ to be a modified U-shaped Neural Operator (UNO) (Rahman et al., 2023) tailored to diffusion-style conditioning. Specifically, the control takes a 1D sequence $\mathbf{x} \in \mathbb{R}^{|\mathbf{L}| \times \mathbb{R}^d}$ and a time input $t$. We embed $t$ with a sinusoidal MLP and build simple Fourier grid features, then compress these into a single conditioning vector. Each encoder and decoder block applies a spectral convolution on low Fourier modes with a point-wise path, then uses adaptive groupnorm modulated by conditioning vector. Conditioning enters only through normalization, which keeps the operator discretization agnostic and suitable for diffusion training. We set a base channel dimension $\mathbf{c}$ and set channel multiplier $[1, 2, 4]$. Each stage uses two spectral residual blocks. Conversely, we set a base Fourier modes and set channel multiplier $[1, 0.5, 0.25]$. The timestep embedding and the conditioning vector use dimension equal to the highest channel.

**Clipping** $\alpha_{\max}$    We clip the energy gradient so its maximum norm does not exceed $\alpha_{\max}$

**Algorithms**    The overall algorithm of the implementation of FAS is summarized in the Algorithm 4.

### C.5. Possible parameterization strategie for other domain

**Rectangular domain**    Let $\mathbf{X} : \mathbb{R}^2 \to \mathbb{R}^d$ be a field on a rectangular subset of $\mathbb{R}^2$ with zero Neumann BCs, cosine modes from the natural basis. Because real data are sampled on a finite grid, the spectrum already contains a built-in cut-off frequency. We can therefore write the operator with a finite eigen decomposition via the discrete cosine transform (DCT) (Rissanen et al., 2023; Lim et al., 2023b; Park et al., 2024). Concretely, we can approximate the Laplacian operator as $\mathcal{A} \triangleq \mathbf{E}\mathbf{D}\mathbf{E}^\top$ with cosine projection matric $\mathbf{E}^\top$ and diagonal eigen values $\{\lambda^{(k)}\}_{k=1}^K$.

**Graph domain**    For a function $\mathbf{X} : \mathcal{G} \to \mathbb{R}^3$ of a discrete graph $\mathcal{G} = \langle \mathbf{V}, \mathcal{E} \rangle$, where $\mathbf{V} = \{v_i\}_{i=1}^n$ is the set of vertices and $\mathcal{E} = \{e_{ij} \mid (i, j) \subseteq |\mathbf{V}| \times |\mathbf{V}|\}$ is the set of edges, we can choose the eigenvectors of the symmetric graph Laplacian $\mathcal{A}_{\mathrm{sym}}$

---

**Algorithm 3** FAS Sampling with path lifting

---

**Require:** Initial condition $\mathbf{x}_0$, linear operator $\mathcal{A}$, trace-class operator $Q$, control $\alpha$, discrete and inverse sine transforms
$\quad$ DST, iDST, set of evaluation points $\mathbf{U} = \{u_i\}_{i=1}^{K}$.

1: Set initial condition $\mathbf{X}_0^{\alpha} = \mathbf{x}_0$ and $\mathbf{R}_0^{\alpha} = \mathbf{0}$
2: **for** time $t$ **in range** $[0, T]$ **do**
3: $\quad$ Estimate control $\alpha_t = \alpha(\mathbf{X}_t^{\alpha}, t)$
4: $\quad$ Sample Gaussian noise $\varepsilon \sim \mathcal{N}(0, \mathbf{I}_K)$
5: $\quad$ Project state, control $\{\tilde{\mathbf{R}}_t^{(k)}\}_{k \in \mathbf{U}} = \text{DST}(\mathbf{R}_t^{\alpha}), \{\tilde{\alpha}_t^{(k)}\}_{k \in \mathbf{U}} = \text{DST}(\alpha_t)$
6: $\quad$ **for** $k = 1, \cdots, |\mathbf{U}|$ **do in parallel**
7: $\quad\quad$ $\tilde{\mathbf{R}}_{t+\delta_t}^{(k)} = \left[ -\lambda^{(k)} \tilde{\mathbf{R}}_t^{(k)} + \sigma_t (\lambda^{(k)})^{-\frac{s}{2}} \tilde{\alpha}_t^{(k)} \right] \delta_t + \sigma_t (\lambda^{(k)})^{-\frac{s}{2}} \varepsilon \sqrt{\delta_t}$
8: $\quad$ **end for**
9: $\quad$ Get residual $\mathbf{R}_{t+\delta_t}^{\alpha} = \text{iDST}(\{\tilde{\mathbf{R}}_{t+\delta_t}^{(k)}\}_{k \in \mathbf{U}})$
10: $\quad$ Update $\mathbf{R}_t^{\alpha} = \mathbf{R}_{t+\delta_t}^{\alpha}$
11: $\quad$ Get path $\mathbf{X}_t^{\alpha} = \mathbf{x}_0 + \mathbf{R}_t^{\alpha}$
12: **end for**

13: **return** $\mathbf{X}_T^{\alpha} \sim \mathbb{P}_T^{\alpha}$

---

as the spectral basis. Each graph signal is already finite-dimensional, with $|\mathbf{V}|$ modes acting as an inherent spectral cut-off. The operator admits a graph Fourier decomposition $\mathcal{A}_{\text{sym}} = \mathbf{U}\mathbf{\Lambda}\mathbf{U}^{\top}$, where $\mathbf{U}^{\top}$ holds the Laplacian eigenvectors and $\mathbf{\Lambda}$ is diagonal with eigen values $\{\lambda^{(k)}\}_{k=1}^{|\mathbf{V}|}$. Specifically, in molecule conformal generation, $\mathbf{X} : \mathcal{G} \to \mathbb{R}^3$ represents a conformal field (Wang et al., 2024). Note that unlike the rectangular or time-series domain, where every function shares a common basis, each molecule $\mathcal{G}_i$ has its own graph, so we should computed the corresponding eigen-system for each training sample.

### C.6. Potential Functions and Initial conditions

For consistency and fair comparison, we follow the respective experimental setups established by baseline methods. For the Müller Brown potential, we follow the experimental setup and reported results from (Du et al., 2024). for the molecule potentials, we follow (Seong et al., 2025) and reported results from (Seong et al., 2025; Blessing et al., 2026).

- The synthetic Müller–Brown potential $V : \mathbb{R}^2 \to \mathbb{R}$ used in our experiments is given by

$$V(x, y) = -200 \cdot \exp(-(x_1 - 1)^2 - 10y^2) - 100 \cdot \exp(-x^2 - 10 \cdot (y - 0.5)^2) \tag{192}$$

$$- 170 \cdot \exp(-6.5 \cdot (x + 0.5)^2 + 11 \cdot (x + 0.5) \cdot (y - 1.5) - 6.5 \cdot (y - 1.5)^2) \tag{193}$$

$$+ 15 \cdot \exp(0.7 \cdot (x + 1)^2 + 0.6 \cdot (x + 1) \cdot (y - 1) + 0.7 \cdot (y - 1)^2). \tag{194}$$

For synthetic potential, we set initial condition $\mathbf{x}_0$ as linear interpolation between two local minimums $\mathbf{A} = (-0.558, 1.442)$ and $\mathbf{B} = (0.624, 0.028)$.

- For real-world molecule system, we implement the potential with OpenMM library (Eastman et al., 2023). Since OpenMM library dose not provide automatic differentiation, we wrap it in PyTorch (Paszke et al., 2019) and expose batched energy and force calls efficiently.

  We use amber99sbildn (Lindorff-Larsen et al., 2010) for alanine dipeptide in vacuum and set $\mathbf{A} = C5$ and $\mathbf{B} = C7ax$. For Chignolin we use ff14SBonlysc (Maier et al., 2015) with the gbn2 solvent (Nguyen et al., 2013) and we set $\mathbf{A}$ as unfolded protein and $\mathbf{B}$ as folded protein by following the setting in (Seong et al., 2025).

### C.7. Transition Path Sampling on Molecule Potential

**Path Initialization** Due to the rugged and stiff nature of molecular potentials, linear interpolation alone can produce clashing and high energy images along the path, which might produces sub-optimal solutions. To avoid this problem, we introduce additional refinement strategy for molecule potentials.

Given an initial condition $\mathbf{x}_0$, we linearly interpolate Cartesian coordinates between $\mathbf{A}$ and $\mathbf{B}$, then refine the path using the *image dependent pair potential* (IDPP) (Smidstrup et al., 2014) before training. Let $\mathbf{C} = (\mathbf{r}_1, \ldots, \mathbf{r}_N)$ denote a configuration

---

**Algorithm 4** Function Space Adjoint Sampler (FAS)

---

**Require:** Initial condition $\mathbf{x}_0$, terminal cost $g(x)$, control network $\alpha^\theta(\mathbf{x}, t)$, replay buffer $\mathcal{B}$, total epoch $M$, resample sizes $N$, gradient steps $L$, maximum energy norm $\alpha_{\max}$.

1: **for** epoch m **in** $1, 2, \ldots, M$ **do**
2:     Sample $N$ paths $\mathbf{X}_T^{\bar{\alpha}^m}$ from Algorithm 3 with $\bar{\alpha}^m = \mathrm{stopgrad}(\alpha_\theta^{\theta_m})$
3:     Compute adjoint $\mathbf{Y}_T^{\bar{\alpha}^m} = \mathrm{clip}\left(\nabla g(\mathbf{X}_T^{\bar{\alpha}^m}), \alpha_{\max}\right)$
4:     Update buffer $\mathcal{B} \leftarrow \mathcal{B} \cup (\mathbf{X}_T^{\bar{\alpha}^m}, \mathbf{Y}_T^{\bar{\alpha}^m})$
5:     **for** step **in** $1, 2, \ldots, L$ **do**
6:         Sample $t \sim \mathcal{U}[0, T]$, $(\mathbf{X}_T, \mathbf{Y}_T) \sim \mathcal{B}$ and $\mathbf{X}_t \sim \mathbb{P}_{t|T}(\cdot | \mathbf{X}_T)$
7:         Compute the training objective:

$$\mathcal{L}_{\mathrm{FAS}}(\theta) = \mathbb{E}_{t, (\mathbf{X}_T, \mathbf{Y}_T), \mathbf{x}_t}\left[\left\|\alpha^{\theta_s}(\mathbf{X}_t, t) + \sigma_t Q^{1/2} e^{-(T-t)\mathcal{A}^\dagger} \mathbf{Y}_T\right\|_{\mathcal{H}}^2\right]$$

8:         Update $\theta_m^{l+1}$ with $\nabla_\theta \mathcal{L}_{\mathrm{FAS}}(\theta_m^l)$.
9:     **end for**
10: **end for**
11: **return** Approximated optimal control $\alpha^\star \approx \alpha^{\theta_M}$

---

**Algorithm 5** IDPP initialization

---

**Require:** Transition states $\mathbf{A}$ and $\mathbf{B}$, discretization $\mathbf{U} = \{u_i\}_{i=1}^K$ with $0 = u_1 < \cdots < u_K = 1$.

1: Set initial condition $\mathbf{x}_0 = (1-t)\mathbf{A} + t\mathbf{B}$
2: **for** implicit time u **in range** $[u_1, u_K]$ **do**
3:     Compute target distance $\tilde{d}_{ij}[u]$ in (195) for all $(i, j) \in N \times N$.
4:     Take $L$ gradient steps $\nabla_{\mathbf{x}_0[u]} \mathbf{E}_{\mathrm{IDPP}}(\mathbf{x}_0[u])$ w.r.t on IDPP energy (196).
5: **end for**
6: **return** Refined initial condition $\tilde{\mathbf{x}}_0$

---

with atom positions $\mathbf{r}_i \in \mathbb{R}^3$. Given endpoints $\mathbf{A}$ and $\mathbf{B}$, and a path with images $\mathbf{x}[u] = \mathbf{0}_{\mathbf{N} \times 3}$ and the target distance for pair $(i, j)$ at time $u$ as linear interpolation between pair-wise distance for a given endpoints:

$$\tilde{d}_{ij}[u] = (1-u)d_{ij}(\mathbf{A}) + u d_{ij}(\mathbf{B}), \quad \text{where} \quad d_{ij}(\mathbf{C}) = |\mathbf{r}_i - \mathbf{r}_j|. \tag{195}$$

Then, the IDPP minimizes the mismatch between current and target distances via

$$\mathbf{E}_{\mathrm{IDPP}}(\mathbf{x}[u]) = \sum_{i<j} w_{ij}^{(u)}\left(d_{ij}(\mathbf{x}[u]) - \tilde{d}_{ij}[u]\right)^2, \tag{196}$$

where $\mathbf{w}_{ij}^{(u)} = 1/\left(\tilde{d}_{ij}[u]\right)^4$ is the weight over all images $u \in \mathbf{U}$. This initialization yields a path near a minimum-energy path with negligible time cost $e.g.$, less than a minute. Algorithm 5 summarizes the IDPP initialization. For chignolin, we take additional gradient flow step on energy (28) with small step size $\epsilon = 10^{-6}$ to make the initial path more stable after IDPP initialization:

$$\mathbf{x}_{t+\epsilon} = \mathbf{x}_t - \nabla U(\mathbf{X}_t)\epsilon. \tag{197}$$

**Regularization** The IDPP improves the starting path by producing realistic images for both alanine dipeptide and chignolin. However, for alanine dipeptide, diffusion in Cartesian coordinates may move the path into energy troughs that stretch or compress bonds beyond physical ranges. We therefore add a regularization term to the terminal cost to limit bond-length deviations:

$$\tilde{U}(\mathbf{X}) = U(\mathbf{x}) + \lambda_{\mathrm{reg}} \sum_{u \in \mathbf{L}} \mathbf{E}_{\mathrm{IDPP}}(\mathbf{x}[u]), \tag{198}$$

where $\lambda_{\mathrm{reg}} > 0$ is constant. This regularization cost penalizes deviations of pairwise distances from the IDPP reference at each image and suppresses short-range clashes and over-stretched bonds. We set $\lambda_{\mathrm{reg}} = 1$ for alanine dipeptide.

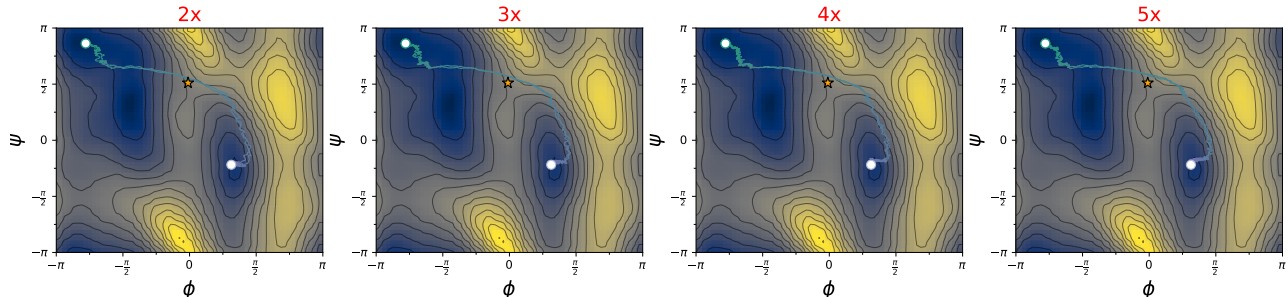

*Figure 4.* Sampled transition path on alanine dipeptide over various discretization steps.

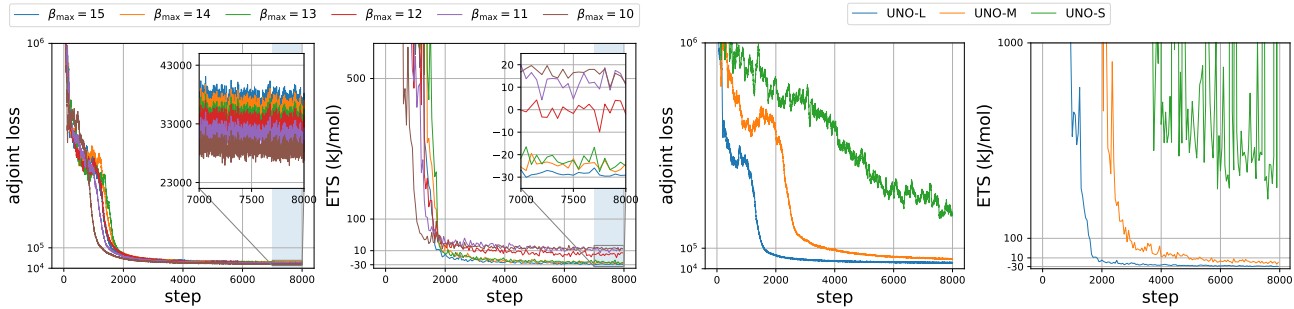

*Figure 6.* **Ablation studies on alanine dipeptide.** (Left) Ablation study on $\beta_{\max}$. (Right) Ablation study on model Scaling.

## C.8. Additional Experiments

**Discretization invariance sampling** Beyond effective boundary enforcement, the functional formulation offers a key benefit: discretization invariance. Once the model is trained on a fixed and usually coarse grid, it can sample paths at arbitrary resolution. This capability has increased interest in functional formulations across scientific domains.

In this section, we evaluate the discretization invariance of FAS on TPS. For TPS, this property is valuable when more fine configuration image near saddle points is required. Classical Langevin dynamics based samplers can refine a path by reducing the step size. However, such an refinement requires additional force and energy evaluations that grow with the number of steps, which increases computational cost and inference time. In contrast, FAS is trained once on a small discretization. At inference it generates target paths on any chosen refined grid without extra energy evaluations because the energy cost is amortized in the learned control.

Figure 5 shows zero shot TPS on a synthetic potential at extremely fine resolutions. Specifically, We train on 100 number of uniform discretization over implicit time interval $[0, 1]$, then evaluate at $\times 10$ and $\times 100$ finer grids with $10^3$ and $10^4$ discretization steps. Despite zero shot inference, FAS generates reliable paths consistently. Note that sampling on high resolution requires a larger diffusion scale than originally chosen $(\beta_{\min}, \beta_{\max})$ for desired performance, so we scale $(\beta_{\min}, \beta_{\max})$ with the refinement factor. Concretely, if the grid is refined by factor $r$, we use $\hat{\beta}_{\min} = (1 + 2\log_{10}(r))\beta_{\min}$ and $\hat{\beta}_{\max} = (1 + 2\log_{10}(r))\beta_{\max}$ for high-resolution sampling.

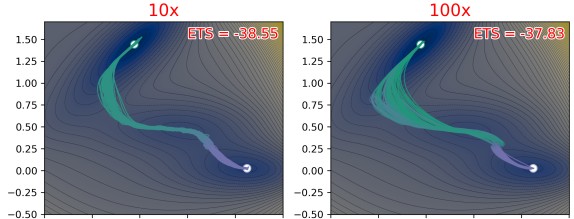

*Figure 5.* Sampled transition path on synthetic potential over various discretization steps.

Figure 4 shows the same behavior on alanine dipeptide. Due to the complex potential landscape of molecule system, we observed that a few evaluation points can show short spikes. We remove these artifacts by applying 10 number of post processing gradient flow steps as in (197) with $\epsilon = 10^{-6}$.

**Ablation studies** For FAS training, the matching objective in (22) depend on sufficient exploration of the path-energy landscape $g$. Moreover, broad exploration in early stage is key to locating the critical point of $\mathcal{L}_{\text{FAS}}$ because higher early

diffusion schedule $\sigma_t$ enlarges the support of sampled paths in $g$ effectively. We therefore ablate the effect of diffusion magnitude $\sigma_t$ helps the performance by increasing $\beta_{\max} \in [10, 15]$ with fixed $\beta_{\min} = 0.01$. As shown in Figure 6, we observed that resulting **ETS** consistently decreases as $\beta_{\max}$ increases.

Additionally, we ablate how FAS scales with model size. Compared to the original path-wise SOC objective in (12), the point-wise computation nature of matching-type objective in (22) can scales better without heavy computation. We vary network size by setting the base dimension to $32$ (UNO-S), $48$ (UNO-M), and $64$ (UNO-L). Note that although the number of parameters increases, memory usage remains nearly unchanged. Figure 6 shows that larger models yield better **ETS**, which demonstrates the FAS scales with model size.

