# OpenReview forum: "Functional Adjoint Sampler: Scalable Sampling on Infinite Dimensional Spaces"
_ICML.cc/2026/Conference — ICML 2026 regular_

### Official Review · Reviewer_eUDQ · 2026-02-20

**Soundness:** 3
**Presentation:** 2
**Significance:** 3
**Originality:** 3
**Overall Recommendation:** 5
**Confidence:** 4

**Summary:**

This paper extends the Adjoint Sampler into functional space, generalizing the original stochastic control problem to an infinite-dimensional one. The aim is to sample from Gibbs distributions on function/path spaces. The authors leverage the stochastic maximum principle to yield a more general adjoint matching training objective. Experiments on real and synthetic molecular systems are constructed and evaluated.

**Compliance With Llm Reviewing Policy:**

Affirmed.

**Key Questions For Authors:**

1. Can you provide some intuition behind Y and Z in the SMP? Y is akin to $ \nabla_x V(x,t) $, $ V $ being the value function. How is $ Z $ initialized and/or propagated? Is it also related to the finite-dimensional analog of $ \nabla^2 V(t,x) \cdot \sigma_t $?

2. Can you provide some intuition on the necessity of the stopgrad in the adjoint matching loss? Specifically, under the stopgrad operation, the forward law $ \mathbb{P}^{\bar \alpha} $ does not depend on $ \theta $ so in effect we are not actually sampling trajectories from the current, most accurate measure though the critical point remains unchanged.

3. Can you clarify the evaluation metrics used? Please see my comments in the Weakness section.

4. Please see all other points in the Weakness section.

5. What happens if some of the idealistic assumptions (e.g. trace class operators) are not satisfied?

I am happy to engage in a discussion with the authors and am willing to revise my score once my understanding has improved.

**Limitations:**

The authors mention they focused on a one-dimensional domain in this work.
Please see the Key Questions/Weakness parts above.

**Strengths And Weaknesses:**

**Strengths**:
* Interesting use of stochastic maximum principle for this infinite-dimensional SOC problem.
* Theory is well-motivated.
* A *tractable* generalization of the adjoint sampler work has been formalized.
* Experiments on alanine dipeptide and chignolin demonstrate the algorithm's performance; though, I have some doubts -- see the below Weakness section.
* The sampling is shown to be invariant to discretization, an advantage of the functional formulation rather than usual sampling in $ \mathbb{R}^d $.

**Weaknesses**:
* THP experiments may not be a good indicator of performance, since the boundary formulation enforces the endpoints (i.e. by construction) (I may have misunderstood the construction here).
* Another evaluation metric, ETS rewards minimum-energy behavior heavily (which makes sense as the formulation is a minimum energy control problem). However, does this necessarily guarantee correct path-distribution sampling?
* The authors themselves mention this, but only a 1D domain is considered. A minimal, small-scale demonstration for the other domains they claim FAS extends to, would be interesting to see (especially given that the SMP is a complex system computationally).
* How about systems with periodic boundaries/Neumann boundaries, etc.? I think only Dirichlet boundaries are considered.
* For molecular systems, how are you ensuring the technical conditions of SMP are satisfied (the smoothness, etc. conditions are fairly strong).


**Typos**:
I will tag English/grammar/sentence structure errors as simply "English" with any accompanying comments. Please try to fix them for readability.
* "In practice, we take a particular **interets**
* Lines 088-093: "...,choice of a basis" English -error in sentence
* Line 096: "which based on" English
* Line 085-086, 2nd col: "Let (\Omega, etc.) and (\mathcal{H}, etc) be A measurable state space in which H-valued random variable" English
* Line 105-106, 2nd col: ", formal series " English
* Line 108, 2nd col: "the set of independent Brownian motion." English
* Line 147, 2nd col: "recast the our sampling problem" English
* Line 168-173: "Outside that narrow compatibility the measures are singular, " English, "are become mutually absolutely continuous" English
* Line 175: "Let the Assumption B.1 holds" English
* Line 195: "Therefore, IS might becomes impractical" Eng
* Line 210: "Now, we consider a variational path measures" Eng
* Line 216: "is Markov control functional" eng
***
Apologies, I cannot continue this as it seems there are far too many. Please read through the paper and enhance the grammar/sentence structure for readability.

---

> ### Author Rebuttal · Authors · 2026-03-31
>
> We thank the reviewer for the constructive feedback, and for recognizing the theoretical motivation and technical contribution of our work. Below, we address the reviewer’s concerns and suggestions below.
>
> ----
> **1. Boundary condition, assumptions.**
>
> > How about systems with periodic boundaries/Neumann boundaries?
>
> * Dirichlet boundaries are natural for TPS with fixed initial and terminal states, since they impose endpoints directly in path space. Other applications may favor Neumann or periodic boundaries, and Appendix C.2 outlines how the same function-space sampling recipe extends to such settings (e.g., Neumann boundary). We will also revise Proposition 6 so that it is not read as restricting the formulation to Dirichlet boundaries.
>
> > For molecular systems, how are you ensuring the technical conditions of SMP are satisfied…
>
> * In our formulation, the state is a path $X:[0,L]\to\mathbb{R}^d$, so the relevant object is a functional on a path-space Hilbert space $\mathcal{H}:=L^2([0,L],\mathbb{R}^d)$, rather than a pointwise energy defined on molecular configurations in $\mathbb{R}^d$.
> * Accordingly, the SMP assumptions are imposed on path-space functionals in our formulation, rather than on arbitrary raw molecular force fields. We agree that fully verifying these assumptions for arbitrary raw molecular force fields is nontrivial, and we do not claim that level of generality here.
>
> >What happens if some of the idealistic assumptions (e.g. trace class operators) are not satisfied?
>
> * The assumptions in our analysis, including the trace-class condition, are structural rather than cosmetic, since the present theory is built on the standard infinite-dimensional SDE framework [1].  If these assumptions are not satisfied, then the current theoretical justification does not directly apply.
>
> * In our formulation, Proposition 6 provides a sufficient condition under which these requirements hold. More specifically, it identifies a Laplacian-based operator class for which the assumptions used in the present analysis are satisfied.
>
> * Extending the formulation to more general settings, such as irregular domains, nonlocal operators, or state spaces without a natural Laplacian generator, is an interesting direction for future work.
>
> **2. Evaluation metrics.**
>
> * We agree that THP is not, by itself, a certificate of correct path-distribution sampling under our Dirichlet formulation. Instead, we use THP as a transition-feasibility metric. It also highlights a practical advantage of our formulation in TPS: unlike many finite-dimensional baselines, it does not require an additional terminal-state penalty, which can be difficult to optimize in high-dimensional settings.
>
> * Likewise, a low ETS score alone does not guarantee correct path-distribution sampling. We report ETS mainly because it is used in [2], enabling direct comparison with prior baselines. More generally, correct path-distribution sampling should be evaluated through multiple complementary criteria rather than any single metric alone.
>
> * In settings such as ALDP, where the true transition state is more explicitly characterized, one can further assess sample quality by checking whether the transition states of generated paths are close to the true transition state, as illustrated in Figure 2(a).
>
> **3. Discussion on Adjoint matching loss.**
>
> > Can you provide some intuition behind Y and Z in the SMP?
>
> * Both $Y$ and $Z$ play the same roles as in finite-dimensional SMP. In practice, however, we do not explicitly initialize or propagate $Z$, as this would make the adjoint estimation much less tractable.
>
> * Instead, Proposition 5 shows that replacing the pathwise adjoint with the conditional expectation $\tilde{Y}_t=\mathbb E[Y_t\mid X_t]$ yields a closed-form solution of adjoint process and an unbiased gradient estimator, allowing efficient adjoint estimation without explicitly modeling $Z$.
>
> > Can you provide some intuition on the necessity of the stopgrad in the adjoint matching loss?
>
> * We use stopgrad because adjoint matching is a self-consistency objective, so both the target and its expectation depend on the current control. Without stopgrad, optimization becomes a moving-target problem that requires differentiating through both the sampling law and the target itself.
> * A useful intuition comes from off-policy RL. One often learns from stale targets or replay distributions rather than requiring fully on-policy samples at every update. Stopgrad plays the same role here by freezing the target branch, which stabilizes training while preserving the same fixed point.
>
> **4. Typos.**
>
> * We sincerely appreciate the feedback regarding readability. We will conduct a thorough proofreading pass to address any syntactic errors, enhance sentence flow, and improve overall clarity.
>
> ----
>     [1] Da Prato and Zabczyk, Stochastic Equations in Infinite Dimensions.
>     [2] Seong et al., Transition path sampling with improved off-policy training of diffusion path samplers.

---

> > ### Author Rebuttal · Reviewer_eUDQ · 2026-04-01
> >
> > I believe the authors have clarified my questions and I have a better understanding of this novel work, and can better place it among other "neural samplers" works. Thank you.

---

### Official Review · Reviewer_3DRF · 2026-03-11

**Soundness:** 4
**Presentation:** 4
**Significance:** 3
**Originality:** 4
**Overall Recommendation:** 5
**Confidence:** 4

**Summary:**

This paper extends Adjoint Sampling (Havens et al )  to
infinite-dimensional function spaces, i.e. where the Gibbs distribution is defined on a  trajectory space H (a Hilbert space).
It relies on  the following construction. The Gibbs distribution and its potential U are defined with respect to a valid reference measure number that is a centered Gaussian measure in H with covariance operator Q. They consider an H-valued stochastic process following the gradient flow of U (similarly to a Langevin dynamic in R^d). However, the mixing of this process is effective near equilibrium since it relies on the asymptotic convergence of the process to the invariant measure nu_infinity.
Hence, they consider a stochastic optimal control problem, to sample exactly from the target at a finite time horizon T.
The latter can be solved through adjoint matching, i.e. a scheme relying on backward on « adjoint pairs » (Yt, Zt) solving a backward SDE (BSDE). The latter generalizes the approach of Doming-Enrich et al 2025 (developed on R^d) to H.
 The claim is that it is more efficient than an important sampling (IS) scheme reweighing the current dynamics law with respect to nu_infinity, that would require a high computational cost/fine discretization in space.

Yet solving the original adjoint matching objective requires simulating a BSDE (to obtain Yt). Hence the authors propose to substitute the adjoint process with its conditional expectation, leading to a novel objective with the same critical point than the original one, while yielding a closed-form solution of the BSDE.
Regarding the numerical simulation, they adopt a standard Galerkin approach that uses K-truncated eigendecompositions of the function-(H) valued stochastic processes. They evaluate their methods on three benchmarks for transition path sampling.

**Compliance With Llm Reviewing Policy:**

Affirmed.

**Key Questions For Authors:**

Sec 5.3 discusses the time-discretization influence. But the method was originally motivated to be sample efficient, regarding space discretization. Can the authors comment on the scaling of their method with respect to others? and the influence of space discretization on their results?

What is the computational cost of the other methods (eg the ones of Table 2) ? The claim is that yours achieve pi in finite time  (hence more reasonable time in practice than infinite time). Are alternative methods comparable in their design?  And in their practical running cost?

Figure 1/Table 1: what is the performance of MCMC? Do you have a plot? Doesn’t achieve the target in a reasonable time and this is why THP is absent from table 1?

Minor questions:
->  g in 12 and 13 is not the same function right? This can be confusing.

**Limitations:**

The limitation is clearly discussed in the conclusion (relatively low dimension)

**Strengths And Weaknesses:**

Strengths:
- The paper is generally very well written.
- The method and idea is very novel. As explained (eg in intro or related work) most works generalizing « sampling from a probability distribution » in function spaces are restricted to the generative modeling setting, where samples of the target distribution are available. In contrast, this paper considers a sampling setting, where the unnormalized density form of the target (its potential) is known but no samples are available.
- The experiments are convincing

Weaknesses:
- Yet very technical and clearly not easily accessible to a wide ICML audience.
- the competing methods (eg MD, or the ones in Table 2) are not clearly introduced and discussed
- the numerical experiments are developed on examples where the paths are evaluated on a one dimensional time-grid

---

> ### Author Rebuttal · Authors · 2026-03-31
>
> Thank you for recognizing the contributions of our work and for the careful, thoughtful review. We address every comment in detail in the responses provided below.
>
> ----
>
> **1. Space-discretization scaling.**
>
> * Thank you for this question. In our setting, discretizing the path domain $u \in [0,L]$ is exactly the relevant space discretization in the function-space sense, and this is the quantity studied in Section 5.3. We referred to it as “time discretization” only because, in TPS, the path domain coincides with physical time. We will revise Section 5.3 to avoid the ambiguity in terminology.
>
> * Regarding scaling, FAS and finite-dimensional baselines do not scale along exactly the same computational axis, the comparison with these baselines should be interpreted carefully,
>
> * In FAS, we can separate (i) _the path-space discretization_, which controls how densely the trajectory is represented on $u \in [0,L]$, from (ii) _the internal SDE-solver discretization_, which controls how many Euler-Maruyama steps on $t \in [0,T]$ are used during train/inference.
>
> * In contrast, in baselines such as TPS-DPS, the trajectory discretization is tied to the solver discretization itself, so refining the discretization also increases the number of solver steps and corresponding score/force evaluations.
>
> * To isolate the effect of space discretization in FAS, we fix the internal SDE discretization at $200$ and vary only the test path discretization. As shown in table below,  increasing the test discretization from $100$ to $1000$ increases inference time from $1.76$ (s) to $8.69$ (s) and memory from $942$ (MB) to $7452$ (MB). This shows that, in FAS, finer space discretization mainly increases the cost of carrying a higher-resolution path-valued state at each fixed solver step, rather than the cost of taking more solver steps.
>
> * Compared with TPS-DPS, FAS remains substantially faster in wall-clock time even at test discretization $1000$ ($8.69$ (s) vs. $60.25$ (s) in our implementation), but it uses more memory because the full path-valued state is processed at every solver step.
>
> * We will clarify in the revision that this is the main scaling tradeoff: FAS decouples path resolution from solver resolution, which gives favorable runtime scaling, while memory grows with the path discretization.
>
>     | Method  | Train disc. | Test disc. | SDE solver disc. | Inference time (s) | Memory (MB) |
>     |---|-------:|-------:|-----:|----------:|-----:|
>     | TPS-DPS | 1000   | 1000 | 1000 | 60.25 | 35 |
>     | FAS     | 100 | 100  | 200 | 1.76 | 942 |
>     | FAS     | 100 | 500 | 200 | 4.64 | 3841 |
>     | FAS     | 100 | 1000 | 200 | 8.69 | 7452 |
>
> **2. Clarification of baseline method.**
>
> > the competing methods (eg MD, or the ones in Table 2) are not clearly introduced and discussed.
>
> * We agree that the baselines are not introduced clearly enough. We will revise the paper to better explain these baselines in the related work and experimental sections.
>
>
> > What is the computational cost of the other methods (eg the ones of Table 2) ? …
>
> * The relevant infinite-time counterpart of FAS is functional MCMC on path space [2], not the practical TPS baselines in Table 2. Our finite-time claim is made relative to that asymptotic function-space sampling view. By contrast, the Table 2 methods are comparable mainly because they target the same final law $\pi$, but not because they share the same design principle or objective. We therefore intend Table 2 as an empirical comparison of different practical approaches to the same target distribution.
>
> > Figure 1/Table 1: what is the performance of MCMC? Do you have a plot? Doesn’t achieve the target in a reasonable time and this is why THP is absent from table 1?
>
> * Thank you for this question. The absence of MCMC THP from Table 1 was not because MCMC failed to reach the target within a reasonable time. In [1], MCMC was used as a proxy ground truth and attains 100% THP under the same setting. We did not emphasize it in Table 1 because the main comparison there was against DL. For completeness, we will include these results in the revision.
>
> **3. Typo.**
>
> * Thank you for pointing this out. The two uses of $g$ denote different objects, and we will introduce distinct notation to avoid the ambiguity.
>
> ----
>
>     [1] Du et al., Doob’s lagrangian: A sample-efficient variational approach to transition path sampling.
>     [2] Beskos et al., Mcmc Methods for Diffusion Bridges.

---

> > ### Author Rebuttal · Reviewer_3DRF · 2026-03-31
> >
> > Thank you for the answer and precisions on numerical results. I have also read other's reviews and answers and while the paper is involved, I remain enthousiastic about it. Yet I need to precise that while I know well sampling methods in finite-dimensional spaces, I am not familiar with its infinite counterpart and the complementary references brought by other reviewers (eg Xfzg).

---

### Official Review · Reviewer_mGjZ · 2026-03-12

**Soundness:** 4
**Presentation:** 4
**Significance:** 4
**Originality:** 4
**Overall Recommendation:** 5
**Confidence:** 3

**Summary:**

The paper introduces functional adjoint sampler(FAS), a novel sampling algorithm that can sample Gibbs measure in infinite dimensional space. Most existing method can only sample Gibbs measure from finite dimensional spaces, thus FAS is invented to expand the versatility of this class of algorithms.

The authors start by giving a brief review of Gaussian measure, Hilbert space, and related materials in probability theory. Then they start considering a Gibbs measure with unnormalized energy U through the scope of stochastic differential equations. Afterwards, a change of measure is used so that the target distribution can be reached with in finite time horizon. The authors prove the first theorem about the Radon-Nikodym derivative for the two measures.

Then the paper further enhances its theoretical richness by considering the problem from the scope of stochastic optimal control, results related to adjoint matching in Hilbert space and the unbiased property of the estimator are proved.

Finally, the author set up a bridge of numerical computation and gave the experiments methodology/results on 3 different benchmarks.

**Compliance With Llm Reviewing Policy:**

Affirmed.

**Final Justification:**

I stand by my original judgement of this work and I believe I can recommend it to appear at the venue.

**Key Questions For Authors:**

1. Though it is impressive to go from finite dimensional to infinite dimensional cases, the paper assumes Dirichlet boundary conditions. That is an incredibly strong boundary condition that could potentially limit the versatility of the paper. Can it be relaxed? If not, what is the main blocker and is exploring a more relaxed boundary condition a feasible next-step problem? Or, if it cannot be relaxed, can a more general Gibbs measure sampling problem be potentially reduced to the case with Dirichlet boundary condition so the algorithm can cover a wider range of measures?

2. If you want to present this paper to general ML researcher who rarely even look at random variables through the lens of measure theory, how would you present the results to them? And are you able to present the results so that the general audience can learn something from your research?

3. Both algorithm 1 and algorithm 2 are included in appendix. For a reader who focused on the first 8 pages, how can you help that reader fill the gap between heavy stochastic differential equation to executable and computable algorithms? Can you give more introduction on that regard given the nature of this conference?

**Limitations:**

Yes

**Strengths And Weaknesses:**

Strength: Strong novelty as the leap between sampling in finite dimensional space and infinite dimensional space is highly nontrivial. Exceptional mathematical rigor and theoretical depth. Though very tough to read (more on that in the next part), the paper actually has excellent presentation, like a dense Rudin of the research topic they want to cover.

Weakness: My main concern and confusion is the obvious, and, honestly, huge misalignment between the literature and the venue. To fully understand this paper the reader needs to be a proper probabilist, a mathematician, not just a generic ML researcher or computer scientist. Or let me put this way: if one hands this paper to a math professor in probability whose research focus happens to be in discrete probability, that professor will need to sit down, open Le Gall (GTM 274) for quite some time, and refresh themselves before they are able to comprehend the paper.

Then this paper condenses all the math into 8 pages due to conference format restriction and are forced to break up the flow of math between the main paper and the appendix. This makes the reading of this already highly complex paper even harder.

While the contribution and the quality of this paper is indisputably high, I would strongly suggest the authors to consider submitting this paper to a proper top tier journal in probability, such as Stochastic Processes and their Applications (SPA). It is never a good idea to force a 8-page limit on a paper like this and I highly doubt ICML has the right reviewers and enough reviewing time window to properly evaluate the math in this paper line by line.

Therefore, since this paper is an extreme outlier, I will give a weak accept score due to it being rock solid and horribly misaligned at the same time. I sincerely wish the authors can consider submitting this to a probability journal and I have faith that the full, mathematician-oriented version of this paper, has a very good chance to be accepted.

---

> ### Author Rebuttal · Authors · 2026-03-31
>
> We thank the reviewer for the careful and encouraging assessment, and especially for recognizing the novelty, rigor, and significance of our work. We address the reviewer’s questions and suggestions below.
>
> ----
>
> **1. Clarification of  Boundary Condition**
>
> * We chose Dirichlet boundary conditions because they are the most natural fit for the TPS setting considered in this paper, not because our formulation is inherently restricted to them. At the level of the underlying formulation, the approach is not tied to a specific boundary condition.
>
> * Depending on the target sampling domain, other standard choices such as Neumann or periodic boundary conditions can also be accommodated, as these are common assumptions in the ML literature on image and PDE generation (e.g., [1]).
>
> * In this regard, we will revise Proposition 6, as its current statement may be misread as implying that the formulation is restricted to Dirichlet boundary conditions.
>
> **2. Main Takeaway.**
>
> * Many sampling and generative modeling methods are formulated in finite-dimensional spaces, while many objects of interest in ML are more naturally modeled as functions, such as trajectories, fields, or PDE solutions.
>
> * Our paper shows that the key idea of previously studied sampling methods can be lifted to this function-space setting in a principled way. From this viewpoint, the main message is not the measure-theoretic formalism itself, but rather that controlled sampling ideas familiar in modern ML can still be made rigorous and practical beyond finite-dimensional vectors.
>
> * The main takeaway for a general audience is that moving from vectors to functions is not merely a change of notation. It requires identifying which parts of standard sampling formulations genuinely extend and which parts must be reformulated in Hilbert space.
>
> * We believe this is useful even to readers who do not work directly with measure theory, because it clarifies how modern generative-modeling ideas can be transferred to structured infinite-dimensional domains.
>
>
>
> **3. Clearer Theory-to-Algorithm Introduction**
>
> * We appreciate this suggestion and agree that, for a broader ML audience, the connection between the SDE-based formulation and the executable algorithms should be made clearer.
>
> * At a high level, FAS is implemented by first truncating the functional objective into a finite basis representation, then simulating controlled trajectories using a neural network parameterized control, computing the closed-form adjoint target and the corresponding matching loss on those trajectories, and finally training the control with a stopgrad trick. At inference time, we simulate only the learned controlled dynamics and decode the functional representation back into a path representation.
>
> * In revision, we will add a concise algorithm block in the main text that allows readers to understand how FAS works without going through the full mathematical derivation.
>
> ----
>
>     [1] Lim et al., Score-based Generative Modeling through Stochastic Evolution Equations in Hilbert Spaces

---

> > ### Author Rebuttal · Reviewer_mGjZ · 2026-03-31
> >
> > Thank the authors for addressing my questions. Now the paper has a clearer presentation. I am happy to raise my score to accept.
> >
> > Still, I think publishing at ICML could be an undersell. While I am perfectly fine and happy to see this paper to appear in ICML, I do believe it can appear in top tier journals that are more mathematically focused. I hope the authors can consider my recommendation.

---

### Official Review · Reviewer_Xfzg · 2026-03-13

**Soundness:** 3
**Presentation:** 3
**Significance:** 2
**Originality:** 3
**Overall Recommendation:** 4
**Confidence:** 3

**Summary:**

This paper introduces the Functional Adjoint Sampler, a diffusion-based sampling method designed for Gibbs-type distributions defined over infinite-dimensional function spaces. The work extends recent adjoint-based diffusion samplers, which formulate sampling as a stochastic optimal control problem, to the setting of infinite-dimensional Hilbert spaces. Using the stochastic maximum principle, the optimal control problem can be reformulated as a local matching objective similar to adjoint matching in finite-dimensional diffusion samplers.

Empirically, the paper evaluates their method on several TPS benchmarks, including the Müller–Brown potential and two molecular systems (alanine dipeptide and chignolin). The reported results show improvements over several TPS baselines in terms of transition success rates and energy metrics. The paper argues that the proposed infinite-dimensional formulation provides a scalable and principled approach to path-space sampling tasks.

**Compliance With Llm Reviewing Policy:**

Affirmed.

**Final Justification:**

The paper provides strong theory and good experiments on TPS, but it still remains open as to whether their generalized formulation is uniquely applicable to any ML-adjacent applications, since the TPS problem is separable among path waypoints and does not require functional space SDE's.

**Key Questions For Authors:**

- How does the proposed framework relate to recent work such as the Onsager–Machlup Functional approach (Raja et al.)?

- The experiments focus on TPS tasks. Can the authors provide evidence that the proposed more general functional framework is beneficial for other infinite-dimensional sampling problems (e.g., functional diffusion models, PDE-based generative modeling), or clarify what advantages it provides beyond TPS applications?

- Some claims, such as "exact sampling in finite time", appear to apply to the idealized optimal control formulation rather than the practical algorithm. Could the authors clarify which theoretical guarantees carry over to the implemented method with its approximations and truncations?

**Limitations:**

yes

**Strengths And Weaknesses:**

Strengths:
  - The paper presents a principled formulation of path-space sampling as a stochastic optimal control problem in Hilbert spaces, which is conceptually sound and builds on established theory in stochastic control and infinite-dimensional SDEs.

  - The derivation of the adjoint matching objective using the stochastic maximum principle (SMP) is elegant and connects naturally to recent adjoint-based diffusion samplers.

  - The experiments demonstrate that the method can be implemented in practice and applied to TPS problems, including the Müller–Brown potential, alanine dipeptide, and chignolin, with improvements in transition success rates and energy metrics.

Weaknesses:
  - The related work section omits some relevant TPS literature. In particular, work such as [Raja et al., Action-Minimization Meets Generative Modeling: Efficient Transition Path Sampling with the Onsager–Machlup Functional](https://arxiv.org/abs/2504.18506) directly addresses TPS via path-functional optimization and appears closely related.

  - In particular, TPS can also be addressed using Onsager–Machlup action minimization approaches, where the OM action is separable along the path and does not require formulating a general infinite-dimensional SDE. This raises questions about whether the full Hilbert-space diffusion machinery is necessary when the target application admits a simpler path-local structure.

  - The final training objective relies on multiple approximations (e.g., replacing a conditional expectation with a sample estimate of the terminal gradient), and the paper does not provide a theoretical analysis of the error introduced by these approximations These end up being fine (e.g. fixed endpoints), but more explicitly pointing these out may be helpful for future readers (unless I missed something).
  - The empirical evaluation focuses almost entirely on TPS benchmarks, and it remains unclear how broadly the proposed framework applies beyond this domain. If it’s framed as a TPS paper, it should be benchmarked against more generative model-based TPS methods, and if it’s more general, it should be applied to more general settings

---

> ### Author Rebuttal · Authors · 2026-03-31
>
> We thank the reviewer for the thoughtful and comprehensive feedback, and for recognizing the soundness and practical promise of our work. We address the reviewer’s comments and suggestions below.
>
> ----
>
> **1. Relation to Raja et al. [1].**
> * We sincerely thank the reviewer for pointing us to this closely related work. We agree that [1] is highly relevant and should have been discussed explicitly. In the revision, we will add a paragraph to Section 4 clarifying its relation to our method together with other TPS-related works.
> * Our view is that [1] and FAS are complementary because they address different settings and objectives. [1] approaches TPS through OM optimization of a discretized path objective, using a score extracted from a pre-trained diffusion or flow model, which is useful when explicit energy is unavailable. By contrast, FAS assumes access to the energy functional, which is natural in the TPS settings studied here, and learns a reusable sampler for the continuum path-space Gibbs measure. In settings where only trajectory data are available, a similar pre-training stage could also benefit FAS.
> * In discretize-then-learn approaches, the learned object is tied to the chosen path resolution, since discretization is part of the target problem itself. In our case, the target is the underlying continuum path-space measure, while discretization is only a numerical representation. This is why we adopt the Hilbert-space formulation: to define and learn a sampler for the continuum target rather than one fixed discretization. This supports cross-discretization transfer, as shown in Section 5.3.
> * We therefore do not claim that Hilbert-space machinery is necessary for every TPS method. Rather, our claim is narrower: it is a natural formulation for the amortized sampler-learning problem considered here.
>
> **2. Advantage of the infinite-dimensional formulation and generality.**
> * We agree that the empirical section is focused on TPS. The formulation itself, however, is not TPS-specific. The mathematical object we study is a Gibbs or posterior measure on a function space, and TPS is one important instance of this broader class.
> * From this perspective, the main advantage is not limited to transition paths. The infinite-dimensional formulation separates the continuum target from its numerical discretization, so the learned sampler is not tied to a single grid or path resolution. This is particularly relevant in settings such as Bayesian inverse problems and PDE-related posterior sampling, where discretization robustness is important [2,3]. Appendix C.2 already outlines how the same construction extends once the target is specified as a Gibbs or posterior measure on a Hilbert space.
> * Direct empirical validation beyond TPS would strengthen this point, and we will make this limitation clearer in the revision. We also hope to clarify that the paper makes a broader methodological contribution, with TPS serving as the main application domain used for validation.
>
> **3. Practical approximation and theoretical guarantee.**
> * The statement _exact sampling in finite time_ refers to the idealized infinite-dimensional optimal control problem. In practice, learning the control involves numerical approximation, as in both finite- and infinite-dimensional settings.
> * Our formulation introduces two main approximations: a sample-based replacement of the conditional expectation term, and the Galerkin truncation used to realize the infinite-dimensional objective numerically. The first preserves the critical point and yields the infinite-dimensional analogue of the adjoint-matching loss, while the second is the standard numerical realization of a function-space formulation.
> * Accordingly, the guarantee should be understood in a limiting sense: as the adjoint-matching loss approaches zero and the truncation level increases, the implemented method approaches the idealized control solution. We will revise the wording to make this distinction explicit.
>
> **4. Comparison with TPS methods and experimental scope.**
> * We focused the empirical section on TPS as the primary application in this paper, and used it to evaluate the proposed sampler in a representative function-space sampling problem.
> * Within this setting, we selected PIPS, TPS-DPS, and DL as the most directly comparable baselines. We agree that additional validation beyond TPS would be valuable, and will clarify more explicitly that our present claim is methodological breadth of the formulation, with TPS as the main validation domain in this submission.
>
> ----
>
>      [1] Raja et al., Action-Minimization Meets Generative Modeling: Efficient Transition Path Sampling with the Onsager-Machlup Functional.
>      [2] Baker et al., Supervised Guidance Training for Infinite-Dimensional Diffusion Models.
>      [3] Yao et al., Guided Diffusion Sampling on Function Spaces with Applications to PDEs.

---

> > ### Author Rebuttal · Reviewer_Xfzg · 2026-04-03
> >
> > Thank you for your response. I would like to lead with the fact that I enjoyed reading the paper, and I think it makes a meaningful contribution to the academic community.
> >
> > However, my main remaining reservation is still that there are no provided ML experiments that explicitly require the more general formulation that you all provided (since TPS is resolvable by point-wise SDE's and doesn't necessarily require functional space SDE's). Thus it is not up to question whether the paper is good or not, but whether this submission is a good fit for ICML.
> >
> > All other points are resolved though, so I will move to a weak accept. Without the additional ML environment though, I will keep my score at a weak accept.
> >
> > Thank you again to the authors for their response.

---

### Decision · Program_Chairs · 2026-04-30

**Decision:**

Accept (regular)

**Comment:**

All reviewers (experts in the relevant area) view the paper as theoretically very strong and technically sound. At the same time, three of the four reviewers note that, despite being well written, the paper may be challenging for a broader ICML audience. The authors’ clarifications during the discussion phase have somewhat alleviated this concern and made the reviewers more optimistic. Overall, all reviewers recommend acceptance, and I support that recommendation.